# Efficient Frameworks for Generalized Low-Rank Matrix Bandit Problems

**Yue Kang**
Department of Statistics
University of California, Davis
Davis, CA 95616
`yuekang@ucdavis.edu`

**Cho-Jui Hsieh**
Department of Computer Science
University of California, Los Angeles
Los Angeles, CA 90095
`chohsieh@cs.ucla.edu`

**Thomas C. M. Lee**
Department of Statistics
University of California, Davis
Davis, CA 95616
`tcmlee@ucdavis.edu`

## Abstract

In the stochastic contextual low-rank matrix bandit problem, the expected reward of an action is given by the inner product between the action's feature matrix and some fixed, but initially unknown $d_1$ by $d_2$ matrix $\Theta^*$ with rank $r \ll \{d_1, d_2\}$, and an agent sequentially takes actions based on past experience to maximize the cumulative reward. In this paper, we study the generalized low-rank matrix bandit problem, which has been recently proposed in [26] under the Generalized Linear Model (GLM) framework. To overcome the computational infeasibility and theoretical restrain of existing algorithms on this problem, we first propose the G-ESTT framework that modifies the idea from [17] by using Stein's method on the subspace estimation and then leverage the estimated subspaces via a regularization idea. Furthermore, we remarkably improve the efficiency of G-ESTT by using a novel exclusion idea on the estimated subspace instead, and propose the G-ESTS framework. We also show that both of our methods are the first algorithm to achieve the optimal $\tilde{O}((d_1 + d_2)r\sqrt{T})$ bound of regret presented in [26] up to logarithm terms under some mild conditions, which improves upon the current regret of $\tilde{O}((d_1 + d_2)^{3/2}\sqrt{rT})$ [26]. For completeness, we conduct experiments to illustrate that our proposed algorithms, especially G-ESTS, are also computationally tractable and consistently outperform other state-of-the-art (generalized) linear matrix bandit methods based on a suite of simulations.

## 1 Introduction

The contextual bandit has proven to be a powerful framework for sequential decision-making problems, with great applications to clinical trials [36], recommendation system [22], and personalized medicine [4]. This class of problems evaluates how an agent should choose an action from the potential action set at each round based on an updating policy on-the-fly so as to maximize the cumulative reward or minimize the overall regret. With high dimensional sparse data becoming ubiquitous in various fields nowadays, the most fundamental (generalized) linear bandit framework, although has been extensively studied, becomes inefficient in practice. This fact consequently leads to a line of work on stochastic high dimensional bandit problems with low dimensional structures [16, 24], such as the LASSO bandit and low-rank matrix bandit.

36th Conference on Neural Information Processing Systems (NeurIPS 2022).

Table 1: Comparison with other low-rank matrix bandit algorithms in rounds $T$. ($d = \max\{d_1, d_2\}$, $D_{rr}$ is the $r$-th largest singular value of $\Theta^*$ that is free of $d$)

| METHOD | REGRET BOUND; ORDER OF $d$ | COMMENT |
|---|---|---|
| UCB-GLM (2017) | $\tilde{O}(d_1 d_2 \sqrt{T})$; 2 | |
| ESTR (2019) | $\tilde{O}(\sqrt{d_1 d_2 d r T}/D_{rr})$; 1.5 | BILINEAR |
| $\epsilon$-FALB (2021) | $\tilde{O}(\sqrt{d_1 d_2 d T})$; 1.5 | BILINEAR |
| Low(G)LOC (2021) | $\tilde{O}(\sqrt{d_1 d_2 d r T})$; 1.5 | |
| LowESTR (2021) | $\tilde{O}(\sqrt{d_1 d_2 d r T}/D_{rr})$; 1.5 | |
| G-ESTT (OURS) | $\tilde{O}(d\sqrt{rT}/D_{rr})$; 1 | |
| G-ESTS (OURS) | $\tilde{O}(d\sqrt{rT}/D_{rr})$; 1 | |

In this work, we investigate on the generalized low-rank matrix bandit problem firstly studied in [26]: at round $t = 1, \ldots, T$, the algorithm selects an action represented by a $d_1$ by $d_2$ matrix $X_t$ from the admissible action set $\mathcal{X}_t$ ($\mathcal{X}_t$ may be fixed), and receives its associated noisy reward $y_t = \mu(\langle \Theta^*, X_t \rangle) + \eta_t$ where $\Theta^* \in \mathbb{R}^{d_1 \times d_2}$ is some unknown low-rank matrix with rank $r \ll \{d_1, d_2\}$ and $\mu(\cdot)$ is the inverse link function. More details about this setting are deferred to Section 3. This problem has vast applicability in real world applications. On the one hand, matrix inputs are appropriate when dealing with paired contexts which are omnipresent in practice. For instance, to design a personalized movie recommendation system, we can formulate each user as $m$ $d_1$-dimensional feature vectors ($x_1, \ldots, x_m \in \mathbb{R}^{d_1}$) and each movie as $m$ $d_2$-dimensional feature vectors ($y_1, \ldots, y_m \in \mathbb{R}^{d_2}$). A user-item pair can then be naturally represented by a feature matrix defined as the summation of the outer products $\sum_{k=1}^{m} x_k y_k^\top \in \mathbb{R}^{d_1 \times d_2}$, which will become the contextual feature observed by the bandit algorithm. Other applications involve interaction features between two groups, such as flight-hotel bundles [26] and dating service [17] can also be similarly established. Besides, low-rank models have gained tremendous success in various areas [5]. In particular, our problem can be regarded as an extension of the inductive matrix factorization problem [14, 38], which estimates low-rank matrices with contextual information, under the online learning scenario.

Our study is inspired by a line of work on stochastic contextual low-rank matrix bandit [15, 17, 26]. To design an algorithm for matrix bandit problems, a naïve approach is to flatten the $d_1$ by $d_2$ feature matrices into vectors and then apply any (generalized) linear bandit algorithms, which, however, would be inefficient when $d_1 d_2$ is large. To take advantage of the low-rank structure, [17] have introduced the bilinear low-rank bandit problem and proposed a two-stage algorithm named ESTR which could achieve a regret bound of $\tilde{O}((d_1 + d_2)^{3/2}\sqrt{rT}/D_{rr})$[1]. Subsequently, [15] constructed a new algorithm called $\epsilon$-FALB for bilinear bandits and achieved a better regret of $\tilde{O}(\sqrt{d_1 d_2 (d_1 + d_2)T})$. However, they only studied the linear reward framework and also restricted the feature matrix as a rank-one matrix. As a follow-up work, [26] further released the rank-one restriction on the action feature matrices, and they introduced an algorithm LowGLOC based on the online-to-confidence-set conversion [2] for generalized low-rank matrix bandits with $\tilde{O}(\sqrt{(d_1 + d_2)^3 rT})$ regret bound. However, this regret bound is still loose compared with the optimal rate $\tilde{O}((d_1 + d_2)r\sqrt{T})$ deduced in [15], and the arm set is still assumed fixed at each round. This algorithm is also computationally prohibitive since it requires to calculate the weights of a self-constructed covering of the admissible parameter space at each iteration. And how to find this covering for low-rank matrices is also unclear.

In this work, we propose two efficient methods called G-ESTT and G-ESTS for this problem by modifying two stages of ESTR appropriately from different perspectives. To the best of our knowledge, the proposed methods are the first two generalized (contextual) low-rank bandit algorithms that are computationally feasible, and achieve an improved regret bound of $\tilde{O}((d_1 + d_2)r\sqrt{T})$ upon all existing works on (bilinear) low-rank bandits. The main contributions of this paper can be summarized as: **1)** we propose two novel two-stage frameworks G-ESTT and G-ESTS under some mild assumptions. Compared with ESTR in [17], $\epsilon$-FALB in [15] and LowESTR in [26], our algorithms are proposed for the nonlinear reward framework with arbitrary action matrices. Compared with LowGLOC in [26], our algorithms not only achieve a better regret bound in theory, but also are computationally feasible in practice. **2)** For G-ESTT, we extend the GLM-UCB algorithms [11] via a novel regularization technique. **3)** Our proposed G-ESTS is simple and could be easily implemented based on any

---

[1] $\tilde{O}$ ignores the polylogarithmic factors.

state-of-the-art generalized linear bandit algorithms. In particular, when we combine G-ESTS with some efficient algorithms (e.g. SGD-TS [9]), the total time complexity after a warm-up stage scales as $O(Tr(d_1 + d_2))$. **4)** We verify that G-ESTT and G-ESTS are the first two algorithms to attain the $\tilde{O}((d_1 + d_2)r\sqrt{T})$ optimal regret bound of low-rank matrix bandit problems up to logarithmic terms. And their practical superiority is also validated in experiments.

**Notations:** For a vector $x \in \mathbb{R}^n$, we use $\|x\|_p$ to denote the $l_p$-norm of the vector $x$ and $\|x\|_H = \sqrt{x^\top H x}$ to denote its weighted $2-$norm with regard to a positive definite matrix $H \in \mathbb{R}^{n \times n}$. For matrices $X, Y \in \mathbb{R}^{n_1 \times n_2}$, we use $\|X\|_{\text{op}}$, $\|X\|_{\text{nuc}}$ and $\|X\|_F$ to define the operator norm, nuclear norm and Frobenious norm of matrix $X$ respectively, and we denote $\langle X, Y \rangle := \textbf{trace}(X^\top Y)$ as the inner product between $X$ and $Y$. We write $f(n) \asymp g(n)$ if $f(n) = O(g(n))$ and $g(n) = O(f(n))$.

## 2 Related Work

In this section, we briefly discuss some previous algorithms on low-rank matrix bandit problems. Besides the works we have discussed in the former section, [18, 33] considered the rank-one bandit problems where the expected reward forms a rank-one matrix and the player selects an element from this matrix as the expected reward at each round. In addition, [19] also studied the rank-one matrix bandit via an elimination-based algorithm. Alternatively, [12, 20, 25] considered the general low-rank matrix bandit, and furthermore [13] considered a stochastic low-rank tensor bandit. However, for all these works the feature matrix of an action could be flattened into a one-hot basis vector, and our work yields a more general structure.

Additionally, [24] extended some previous works [16] and presented a unified algorithm based on a greedy search for high-dimensional bandit problems. But it's nontrivial to extend the framework to the matrix bandit problem. For example, they assume that the minimum eigenvalue of the covariance matrix could be strictly lower bounded, but this lower bound would mostly depend on the size of feature matrices, and hence would affect the regret bound consequently.

## 3 Preliminaries

In this section we review our problem setting and introduce the assumptions for our theoretical analysis. Let $T$ be the total number of rounds and $\mathcal{X}_t$ be the action set ($\mathcal{X}_t$ could be fixed or not). Throughout this paper, we denote the action set $\mathcal{X}_t = \mathcal{X}$ as fixed for notation simplicity, while our frameworks also work with the same regret bound when $\mathcal{X}_t$ varies over time (see Appendix H.3 for more details.) Algorithms along with theory could be identically obtained when the action set varies (Appendix H.3). At each round $t \in [T]$, The agent selects an action $X_t \in \mathcal{X}_t$ and gets the payoff $y_t$ which is conditionally independent of the past payoffs and choices. For the generalized low-rank matrix bandits, we assume the payoff $y_t$ follows a canonical exponential family such that:

$$p_{\Theta^*}(y_t | X_t) = \exp\left(\frac{y_t \beta - b(\beta)}{\phi} + c(y_t, \phi)\right), \text{ where } \beta = \text{vec}(X_t)^\top \text{vec}(\Theta^*) := \langle X_t, \Theta^* \rangle, \quad (1)$$

$$\mathbb{E}_{\Theta^*}(y_t | X_t) = b'(\langle X_t, \Theta^* \rangle) := \mu(\langle X_t, \Theta^* \rangle),$$

where $\Theta^* \subseteq \Theta$ is a fixed but unknown matrix with rank $r \ll \{d_1, d_2\}$ and $\Theta$ is some admissible compact subset of $\mathbb{R}^{d_1 \times d_2}$ (w.l.o.g. $d_1 = \Theta(d_2)$). We also call $\mu(\langle X_t, \Theta^* \rangle)$ the reward of action $X_t$.

In addition, one can represent model (1) in the following Eqn. (2). Note that if we relax the definition of $\mu(\cdot)$ to any real univariate function with some centered exogenous random noise $\eta_t$, the model shown in Eqn. (2) generalizes our problem setting to a single index model (SIM) matrix bandit, and the generalized low-rank matrix bandit problem is a special case of this model.

$$y_t = \mu(\langle X_t, \Theta^* \rangle) + \eta_t. \quad (2)$$

Here, $\eta_t$ follows the sub-Gaussian property with some constant parameter $\sigma_0$ conditional on the filtration $\mathcal{F}_t = \{X_t, X_{t-1}, \eta_{t-1}, \ldots, X_1, \eta_1\}$. We also denote $d = \max\{d_1, d_2\}$. And it is natural to evaluate the agent's strategy based on the regret [3], defined as the difference between the total reward of optimal policy and the agent's total reward in practice:

$$Regret_t = \sum_{i=1}^{t} \max_{X \in \mathcal{X}} \mu(\langle X, \Theta^* \rangle) - \mu(\langle X_i, \Theta^* \rangle).$$

We also present the following two definitions to facilitate further analysis via Stein's method:

**Definition 3.1.** Let $p : \mathbb{R} \to \mathbb{R}$ be a univariate probability density function defined on $\mathbb{R}$. The score function $S^p : \mathbb{R} \to \mathbb{R}$ regarding density $p(\cdot)$ is defined as:

$$S^p(x) = -\nabla_x log(p(x)) = -\nabla_x p(x)/p(x), \quad x \in \mathbb{R}.$$

In particular, for a random matrix with its entrywise probability density $\mathbf{p} = (p_{ij}) : \mathbb{R}^{d_1 \times d_2} \to \mathbb{R}^{d_1 \times d_2}$, we define its score function $S^{\mathbf{p}} = (S_{ij}^{\mathbf{p}}) : \mathbb{R}^{d_1 \times d_2} \to \mathbb{R}^{d_1 \times d_2}$ as $S_{ij}^{\mathbf{p}}(x) = S^{p_{ij}}(x)$ by applying the univariate score function to each entry of $\mathbf{p}$ independently.

**Definition 3.2.** (Fact 2.6, [27]) Given a rectangular matrix $A \in \mathbb{R}^{d_1 \times d_2}$, the (Hermitian) dilation $\mathcal{H} : \mathbb{R}^{d_1 \times d_2} \to \mathbb{R}^{(d_1+d_2) \times (d_1+d_2)}$ is defined as:

$$\mathcal{H}(A) = \begin{pmatrix} 0 & A \\ A^\top & 0 \end{pmatrix}.$$

We would omit the subscript $x$ of $\nabla$ and the superscript $p$ of $S$ when the underlying distribution is clear. With these definitions, we make the following mild assumptions:

**Assumption 3.3.** (Finite second-moment score) There exists a sampling distribution $\mathcal{D}$ over $\mathcal{X}$ such that for the random matrix $X$ drawn from $\mathcal{D}$ with its associated density $\mathbf{p} : \mathbb{R}^{d_1 \times d_2} \to \mathbb{R}^{d_1 \times d_2}$, we have $\mathbb{E}[(S^{\mathbf{p}}(X))_{ij}^2] \leq M, \forall i, j$.

**Assumption 3.4.** The norm of true parameter $\Theta^*$ and feature matrices in $\mathcal{X}$ is bounded: there exists $S \in \mathbb{R}^+$ such that for all arms $X \in \mathcal{X}, \|X\|_F, \|\Theta^*\|_F \leq S_0$.

**Assumption 3.5.** The inverse link function $\mu(\cdot)$ in GLM is continuously differentiable and there exist two constants $c_\mu, k_\mu$ such that $0 < c_\mu \leq \mu'(x) \leq k_\mu$ for all $|x| \leq S_0$.

Assumption 3.3 is commonly used in Stein's method [7], and easily satisfied by a wide range of distributions that are non-zero-mean or even non sub-Gaussian thereby allowing us to work with cases not previously possible. For example, to find $\mathcal{D}$ we only need the convex hull of $\mathcal{X}$ contains a ball with radius $R$, and then we can use $p_{ij}$ as centered normal p.d.f. with variance $R^2/(d_i d_j)$. This choice works well in our experiments and please refer to Appendix I for more details. Furthermore, Assumption 3.4 and 3.5 are also standard in contextual generalized bandit literature, and they explicitly imply that we have an upper bound as $|\mu(\langle X, \Theta \rangle)| \leq |\mu(0)| + \max\{|c_\mu|, |k_\mu|\} S_0 := S_f$.

# 4 Main Results

In this section, we present our novel two-stage frameworks, named Generalized Explore Subspace Then Transform (G-ESTT) and Generalized Explore Subspace Then Subtract (G-ESTS) respectively. These two algorithms are inspired by the two-stage algorithm ESTR proposed in [17]. ESTR estimates the row and column subspaces for the true parameter $\Theta^*$ in stage 1. In stage 2, it exploits the estimated subspaces and transforms the original matrix bandits into linear bandits with sparsity, and then invoke a penalized approach called LowOFUL.

To extend their work into the nonlinear reward setting, we firstly adapt stage 1 into the GLM framework by estimating $\Theta^*$ via the following quadratic optimization problem in Eqn. (6) inspired by a line of work in nonlinear signal estimation [29, 37]. And then we propose two different methods based on the estimated subspaces to deal with GLM bandits for stage 2: for G-ESTT we adapt the idea of LowOFUL into the GLM framework and propose an improved algorithm with regularization. For G-ESTS we innovatively drop all negligible entries to get a low-dimensional bandit, which could efficiently be solved by any modern generalized linear bandit algorithm with much less computation.

## 4.1 Stage 1: Subspace Exploration

For any real-value function $f(\cdot)$ defined on $\mathbb{R}$, and symmetric matrix $A \in \mathbb{R}^{d \times d}$ with its SVD decomposition as $A = UDU^\top$, we define $f(A) := U \operatorname{diag}(f(D_{11}), \ldots, f(D_{dd})) U^\top$. To explore the valid subspace of the parameter matrix $\Theta^*$, we firstly define a function $\psi : \mathbb{R} \to \mathbb{R}$ [27] in Eqn. (5) and subsequently we define $\tilde{\psi}_\nu : \mathbb{R}^{d_1 \times d_2} \to \mathbb{R}^{d_1 \times d_2}$ as $\tilde{\psi}_\nu(A) = \psi(\nu \mathcal{H}(A))_{1:d_1,(d_1+1):(d_1+d_2)}/\nu$ for some parameter $\nu \in \mathbb{R}^+$.

$$\psi(x) = \begin{cases} \log(1 + x + x^2/2), & x \geq 0; \\ -\log(1 - x + x^2/2), & x < 0. \end{cases} \tag{5}$$

---

**Algorithm 1** Generalized Explore Subspace Then Transform (G-ESTT)

---

**Input:** $\mathcal{X}, T, T_1, \mathcal{D}$, the probability rate $\delta$, parameters for Stage 2: $\lambda, \lambda_\perp$.

**Stage 1: Subspace Estimation**

1: **for** $t = 1$ **to** $T_1$ **do**
2:     Pull arm $X_t \in \mathcal{X}$ according to $\mathcal{D}$, observe payoff $y_t$.
3: **end for**
4: Obtain $\widehat{\Theta}$ based on Eqn. (6).
5: Obtain the full SVD of $\widehat{\Theta} = [\widehat{U}, \widehat{U}_\perp] \widehat{D} [\widehat{V}, \widehat{V}_\perp]^\top$ where $\widehat{U} \in \mathbb{R}^{d_1 \times r}, \widehat{V} \in \mathbb{R}^{d_2 \times r}$.

**Stage 2: Sparse Generalized Linear Bandits**

6: Rotate the arm feature set: $\mathcal{X}' := [\widehat{U}, \widehat{U}_\perp]^\top \mathcal{X} [\widehat{V}, \widehat{V}_\perp]$ and the admissible parameter space: $\Theta' := [\widehat{U}, \widehat{U}_\perp]^\top \Theta [\widehat{V}, \widehat{V}_\perp]$.
7: Define the vectorized arm set so that the last $(d_1 - r) \cdot (d_2 - r)$ components are negligible:

$$\mathcal{X}_0 := \{\text{vec}(\mathcal{X}'_{1:r,1:r}), \text{vec}(\mathcal{X}'_{r+1:d_1,1:r}), \text{vec}(\mathcal{X}'_{1:r,r+1:d_2}), \text{vec}(\mathcal{X}'_{r+1:d_1,r+1:d_2})\}, \qquad (3)$$

and similarly define the parameter set:

$$\Theta_0 := \{\text{vec}(\Theta'_{1:r,1:r}), \text{vec}(\Theta'_{r+1:d_1,1:r}), \text{vec}(\Theta'_{1:r,r+1:d_2}), \text{vec}(\Theta'_{r+1:d_1,r+1:d_2})\}. \qquad (4)$$

8: For $T_2 = T - T_1$ rounds, invoke (P)LowGLM-UCB with $\mathcal{X}_0, \Theta_0, k = (d_1 + d_2)r - r^2, (\lambda_0, \lambda_\perp)$.

---

We consider the following well-defined regularized minimization problem with nuclear norm penalty:

$$\widehat{\Theta} = \arg\min_{\Theta \in \mathbb{R}^{d_1 \times d_2}} L_{T_1}(\Theta) + \lambda_{T_1} \|\Theta\|_{\text{nuc}}, \quad L_{T_1}(\Theta) = \langle \Theta, \Theta \rangle - \frac{2}{T_1} \sum_{i=1}^{T_1} \langle \tilde{\psi}_\nu(y_i \cdot S(X_i)), \Theta \rangle. \quad (6)$$

An interesting fact is that our estimator is invariant under different choices of function $\mu(\cdot)$, and we could present the following oracle inequality regarding the estimation error $\left\|\widehat{\Theta} - \mu^* \Theta^*\right\|_F$ for some nonzero constant $\mu^*$ by adapting generalized Stein's Method [7].

**Theorem 4.1.** (Bounds for GLM) *For any low-rank generalized linear model with samples $X_1 \ldots, X_{T_1}$ drawn from $\mathcal{X}$ according to $\mathcal{D}$ in Assumption 3.3, and assume Assumption 3.4 and 3.5 hold, then for the optimal solution to the nuclear norm regularization problem* (6) *with $\nu = \sqrt{2 \log(2(d_1 + d_2)/\delta)/((4\sigma_0^2 + S_f^2)MT_1 d_1 d_2)}$ and*

$$\lambda_{T_1} = 4\sqrt{\frac{2(4\sigma_0^2 + S_f^2)M d_1 d_2 \log(2(d_1 + d_2)/\delta)}{T_1}},$$

*with probability at least $1 - \delta$ it holds that:*

$$\left\|\widehat{\Theta} - \mu^* \Theta^*\right\|_F^2 \leq \frac{C_1 d_1 d_2 r \log(\frac{2(d_1 + d_2)}{\delta})}{T_1}, \qquad (7)$$

*for $C_1 = 36(4\sigma_0^2 + S_f^2)M$ and some nonzero constant $\mu^*$.*

The proof of Theorem 4.1 is based on a novel adaptation of Stein-typed Lemmas and is deferred to Appendix B. We believe this oracle bound is non-trivial since the rate of convergence is better than that deduced from the restricted strong convexity (details in Appendix J) even without assuming the sub-Gaussian property. We also present an intuitive explanation on why our Stein-type method is better than existing matrix estimation approaches under our problem setting in Appendix J.2 Remark. In addition, this bound also holds under a more general SIM in Eqn. (2) other than just GLM. Furthermore, although there exists a non-zero constant $\mu^*$ in the error term, it will not affect the singular vectors and subspace estimation of $\Theta^*$ at all.

After acquiring the estimated $\widehat{\Theta}$ in stage 1, we can obtain the corresponding SVD as $\widehat{\Theta} = [\widehat{U}, \widehat{U}_\perp] \widehat{D} [\widehat{V}, \widehat{V}_\perp]^\top$, where $\widehat{U} \in \mathbb{R}^{d_1 \times r}, \widehat{U}_\perp \in \mathbb{R}^{d_1 \times (d_1 - r)}, \widehat{V} \in \mathbb{R}^{d_2 \times r}$ and $\widehat{V}_\perp \in \mathbb{R}^{d_2 \times (d_2 - r)}$. And we assume the SVD of the matrix $\Theta^*$ can be represented as $\Theta^* = UDV^\top$ where $U \in \mathbb{R}^{d_1 \times r}$ and $V \in \mathbb{R}^{d_2 \times r}$. To transform the original generalized matrix bandits into generalized linear bandit

---

**Algorithm 2** LowGLM-UCB

---

**Input:** $T_2, k, \mathcal{X}_0$, the probability rate $\delta$, penalization parameters $(\lambda_0, \lambda_\perp)$.
Initialize $M_1(c_\mu) = \sum_{i=1}^{T_1} x_{s_1,i} x_{s_1,i}^\top + \Lambda/c_\mu$.
**for** $t \geq 1$ **do**
    Estimate $\hat{\theta}_t$ according to (10).
    Choose the arm $x_t = \arg\max_{x \in \mathcal{X}_0} \{\mu(x^\top \hat{\theta}_t) + \rho_t(\delta) \|x\|_{M_t^{-1}(c_\mu)}\}$, receive $y_t$,
    Update $M_{t+1}(c_\mu) \longleftarrow M_t(c_\mu) + x_t x_t^\top$.
**end for**

---

problems, we follow the works in [17] and penalize those covariates that are complementary to $\widehat{U}$ and $\widehat{V}$. Specifically, we could orthogonally rotate the parameter space $\Theta$ and the action set $\mathcal{X}$ as:

$$\Theta' = [\widehat{U}, \widehat{U}_\perp]^\top \Theta [\widehat{V}, \widehat{V}_\perp], \quad \mathcal{X}' = [\widehat{U}, \widehat{U}_\perp]^\top \mathcal{X} [\widehat{V}, \widehat{V}_\perp],$$

Define the total dimension $p := d_1 d_2$, the effective dimension $k := d_1 d_2 - (d_1 - r)(d_2 - r)$ and the $r$-th largest singular value for $\Theta^*$ as $D_{rr}$, and vectorize the new arm space $\mathcal{X}'$ and admissible parameter space as shown in Eqn. (3) and (4). Then for the true parameter $\theta^*$ after transformation, we know that $\theta_{k+1:p}^* = \text{vec}(\Theta_{r+1:d_1,r+1:d_2}^{*,\prime})$ is almost null based on results in [32] and Theorem 4.1:

$$\left\|\theta_{k+1:p}^*\right\|_2 = \left\|\widehat{U}_\perp^\top U D V^\top \widehat{V}_\perp\right\|_F \leq \left\|\widehat{U}_\perp^\top U\right\|_F \left\|\widehat{V}_\perp^\top V\right\|_F \cdot \|D\|_{\text{op}} \lesssim \frac{d_1 d_2 r}{T_1 D_{rr}^2} \log\left(\frac{d_1 + d_2}{\delta}\right) := S_\perp. \tag{8}$$

Therefore, this problem degenerates to an equivalent $d_1 d_2-$dimensional generalized linear bandit with a sparse structure (i.e. last $p - k$ entries of $\theta^*$ are almost null according to Eqn. (8)). To reload the notation we define $\mathcal{X}_0, \Theta_0$ as the new feature set and parameter space as shown in Algorithm 1.

**Remark.** Note the magnitude of $D_{rr}$ would be free of $d$ since $\Theta^*$ contains only $r$ nonzero singular values, and hence we assume that $D_{rr} = \Theta(1/\sqrt{r})$ under Assumption 3.4. This issue has been ignored in all previous analysis of explore-then-commit-type algorithms (e.g. ESTR [17], Low-ESTR [26]), where the final regret bound of them should be of order $\tilde{O}(d^{3/2} r \sqrt{T})$ instead of the originally-used $\tilde{O}(d^{3/2}\sqrt{rT})$ because of the existence of $D_{rr}$. However, due to the fact that $r \ll d$, it is intriguing and crucial to reduce the order of $d$ in the final regret bound, while the order of $r$ is less important. As shown in Table 1, the regret bounds of all previous low-rank matrix bandits algorithms depend on $d$ by the order $3/2$, and we would show that both of our frameworks could improve the cumulative regret bound by a factor of $\sqrt{d}$, which coincides with the dependence of $d$ in the minimax lower bound of the low-rank matrix bandit problem deduced in [26]. A detailed analysis is presented in the rest of our paper.

### 4.2 Stage 2 of G-ESTT

After reducing the original generalized matrix bandit problem into an identical $p$-dimensional generalized linear bandit problem in stage 2, we can reformulate the problem in the following way: at each round $t$, the agent chooses a vector $x_t$ of dimension $p$ from the transformed action set $\mathcal{X}_0$, and observes a noisy reward $y_t = \mu(x_t^\top \theta^*) + \eta_t$. To make use of our additional knowledge shown in Eqn. (8), we propose LowGLM-UCB as an extension of the standard generalized linear bandit algorithm GLM-UCB [11] combined with self-normalized martingale technique [1]. Specifically, we consider the following maximum quasi-likelihood estimation problem shown in Eqn. (9) for each round with a weighted regularizer, where the regularizer is $\|\theta\|_\Lambda^2/2 = \theta^\top \Lambda \theta/2$ for some positive definite diagonal matrix $\Lambda = \text{diag}(\lambda_0, \ldots, \lambda_0, \lambda_\perp, \ldots, \lambda_\perp)$ with $\lambda_0$ only applied to the first $k$ diagonal entries. By enlarging $\lambda_\perp$, we ensure more penalization forced on the last $p - k$ element of $\theta^*$ as desired.

$$\hat{\theta}_t = \arg\max_\theta \widetilde{L}_t^\Lambda(\theta),$$

$$\widetilde{L}_t^\Lambda(\theta) = \sum_{i=1}^{T_1} \left[y_{s_1,i} x_{s_1,i}^\top \theta - b(x_{s_1,i}^\top \theta)\right] + \sum_{i=1}^{t-1} \left[y_i x_i^\top \theta - b(x_i^\top \theta)\right] - \frac{1}{2}\|\theta\|_\Lambda^2. \tag{9}$$

Here $x_{s_1,i}$ in Eqn. (9) is the special vectorization shown in Eqn. (3) of $[\widehat{U}, \widehat{U}_\perp]^\top X_i [\widehat{V}, \widehat{V}_\perp]$ where $X_i$ is the arm we randomly pull at $i$-th step in stage 1, and $y_{s_1,i}$ is the corresponding payoff we observe. $x_i$ in the second summation of Eqn. (9) refers to the arm we pull at $i$-th step in stage 2. Since $\widetilde{L}_t^\Lambda(\theta)$ is a strictly concave function of $\theta$, we have its gradient equal to 0 at the maximum $\hat{\theta}_t$, i.e. $\nabla_\theta \widetilde{L}_t^\Lambda(\theta)|_{\hat{\theta}_t} = 0$. In what follows, for $t \geq 2, \theta \in \mathbb{R}^p$ we define the function $g_t(\theta)$ and have that

$$\nabla_\theta \widetilde{L}_t^\Lambda(\theta) = \sum_{i=1}^{T_1} y_{s_1,i}\, x_{s_1,i} + \sum_{i=1}^{t-1} y_i\, x_i - \Bigg( \underbrace{\sum_{i=1}^{T_1} \mu(x_{s_1,i}{}^\top \theta) x_{s_1,i} + \sum_{i=1}^{t-1} \mu(x_i{}^\top \theta) x_i + \Lambda\theta}_{:= g_t(\theta)} \Bigg),$$

$$\nabla_\theta \widetilde{L}_t^\Lambda(\theta)|_{\hat{\theta}_t} = 0 \quad \Longrightarrow \quad g_t(\hat{\theta}_t) = \sum_{i=1}^{T_1} y_{s_1,i}\, x_{s_1,i} + \sum_{i=1}^{t-1} y_i\, x_i. \tag{10}$$

We also define a matrix function $M_t(s) = \sum_{i=1}^{T_1} x_{s_1,i} x_{s_1,i}^\top + \sum_{k=1}^{t-1} x_k x_k^\top + \Lambda/s$ for $s \in \mathbb{R}^+$ and denote $V_t := M_t(1)$. Furthermore, a remarkable benefit of reusing the actions $\{X_i\}_{i=1}^{T_1}$ we randomly pull in stage 1 is that they contain more randomness and are preferable to the ones we select based on some strategy in stage 2 regarding the parameter estimation because most vector recovery theory requires sufficient randomness during sampling. More inspiring, the projection step in the tradition GLM-UCB [11], which might be nonconvex and hence hard to solve, is no longer required due the consistency of $\hat{\theta}_t$ after reutilizing $\{X_i\}_{i=1}^{T_1}$. Specifically, if we assume the true parameter $\theta^*$ lies in the interior of $\Theta_0$ and the sampling distribution $\mathcal{D}$ satisfies sub-Gaussian property with parameter $\sigma$, and Assumption 3.4, 3.5 held, then we can show that $\left\| \hat{\theta}_t - \theta^* \right\|_2 \leq 1$ holds with probability at least $1 - \delta$ as long as $T_1 \geq ((\hat{C}_1\sqrt{p} + \hat{C}_2\sqrt{\log(1/\delta)})/\sigma^2)^2 + 2B/\sigma^2$ holds for some absolute constants $\hat{C}_1, \hat{C}_2$ with the definition $B := 16\sigma_0^2(p + \log(1/\delta))/c_\mu^2$. An intuitive explanation along with a rigorous proof are deferred to Appendix D due to the space limit. The proposed LowGLM-UCB is shown in Algorithm 2, and its regret analysis is presented in Theorem C.1 in Appendix.

Notice that we can simply replace $M_t(c_\mu)$ by $V_t$ in Algorithm 2, and the regret bound would increase at most up to a constant factor (Appendix G). A potential drawback of Algorithm 2 is that in each iteration we have to calculate $\hat{\theta}_t$, which might be computationally expensive. We could resolve this problem by only recomputing $\hat{\theta}_t$ whenever $|M_t(c_\mu)|$ increases significantly, i.e. by a constant factor $C > 1$ in scale. And consequently we only need to solve the Eqn. (10) for $O(\log(T_2))$ times up to the horizon $T_2$, which remarkably saves the computation. Meanwhile, the bound of the regret would only increase by a constant multiplier $\sqrt{C}$. We call this modified algorithm as PLowGLM-UCB with the initial letter "P" standing for "Parsimonious". Its pseudo-code and regret analysis are given in Appendix H.1. Equipped with LowGLM-UCB in stage 2, we deduce the overall regret of G-ESTT:

### 4.2.1 Overall regret of G-ESTT

To quantify the performance of our algorithm, we first define $\alpha_t^x(\cdot)$ and $\beta_t^x(\cdot)$ as

$$\alpha_t(\delta) := \frac{k_\mu}{c_\mu}\left( \sigma_0 \sqrt{k\log(1 + \frac{c_\mu S_0^2 t}{k\lambda_0}) + \frac{c_\mu S_0^2 t}{\lambda_\perp} - \log(\delta^2)} + \sqrt{c_\mu}(\sqrt{\lambda_0}S_0 + \sqrt{\lambda_\perp}S_\perp)\right), \tag{11}$$

$$\beta_t^x(\delta) := \alpha_t(\delta)\, \|x\|_{M_t^{-1}(c_\mu)}. \tag{12}$$

And the following Theorem 4.2 exhibits the overall regret bound for G-ESTT.

**Theorem 4.2.** (Regret of G-ESTT) *Suppose we set $T_1 \asymp \sqrt{d_1 d_2 r T \log((d_1 + d_2)/\delta)}/D_{rr}$, and we invoke LowGLM-UCB (or PLowGLM-UCB) in stage 2 with $\rho_t(\delta) = \alpha_{t+T_1}(\delta/2), p = d_1 d_2, k = (d_1 + d_2)r - r^2, \lambda_\perp = c_\mu S_0^2 T/(k\log(1 + c_\mu S_0^2 T/(k\lambda_0)))$, and the rotated arm sets $\mathcal{X}_0$ and available parameter space $\Theta_0$. The overall regret of G-ESTT is, with probability at least $1 - \delta$,*

$$Regret_T = \tilde{O}\left( (\frac{\sqrt{r d_1 d_2}}{D_{rr}} + k)\sqrt{T}\right)$$

Note G-ESTT improves all existing regret bounds by a $\sqrt{d}$ factor based on Table 1 given that $r \ll d$.

---

**Algorithm 3** Generalized Explore Subspace Then Subtract (G-ESTS)

---

**Input:** $\mathcal{X}, T, T_1, \mathcal{D}$, the probability rate $\delta$, parameters for Stage 2: $\lambda, \lambda_\perp$.

**Stage 1: Subspace Estimation**

1: Randomly choose $X_t \in \mathcal{X}$ according to $\mathcal{D}$ and record $X_t, Y_t$ for $t = 1, \ldots T_1$.
2: Obtain $\widehat{\Theta}$ from Eqn. (6), and calculate its full SVD as $\widehat{\Theta} = [\widehat{U}, \widehat{U}_\perp] \widehat{D} [\widehat{V}, \widehat{V}_\perp]^\top$ where $\widehat{U} \in \mathbb{R}^{d_1 \times r}, \widehat{V} \in \mathbb{R}^{d_2 \times r}$.

**Stage 2: Low Dimensional Bandits**

3: Rotate the arm feature set: $\mathcal{X}' := [\widehat{U}, \widehat{U}_\perp]^\top \mathcal{X} [\widehat{V}, \widehat{V}_\perp]$ and the admissible parameter space: $\Theta' := [\widehat{U}, \widehat{U}_\perp]^\top \Theta [\widehat{V}, \widehat{V}_\perp]$.
4: Define the vectorized arm set so that the last $(d_1 - r) \cdot (d_2 - r)$ components are negligible, and then **drop** them:

$$\mathcal{X}_{0,sub} := \{\text{vec}(\mathcal{X}'_{1:r,1:r}), \text{vec}(\mathcal{X}'_{r+1:d_1,1:r}), \text{vec}(\mathcal{X}'_{1:r,r+1:d_2})\}, \tag{13}$$

and also refine the parameter set accordingly:

$$\Theta_{0,sub} := \{\text{vec}(\Theta'_{1:r,1:r}), \text{vec}(\Theta'_{r+1:d_1,1:r}), \text{vec}(\Theta'_{1:r,r+1:d_2})\}. \tag{14}$$

5: For $T_2 = T - T_1$ rounds, invoke any generalized linear bandit algorithm with $\mathcal{X}_{0,sub}, \Theta_{0,sub}, k = (d_1 + d_2)r - r^2$.

---

## 4.3 Stage 2 of G-ESTS

Although G-ESTT is more efficient than all existing algorithms on our problem setting, it still needs to calculate the MLE in high dimensional space which might be increasingly formidable with large sizes of feature matrices. Note this computational issue remains ubiquitous among most bandit algorithms on high dimensional problems with sparsity, not to mention these algorithms rely on multiple unspecified hyperparameters. Therefore, to handle this practical issue, we propose another fast and efficient framework called G-ESTS in this section.

Inspired by the success of dimension reduction in machine learning [34], we propose G-ESTS as shown in Algorithm 3. And we summarize the core idea of G-ESTS as: After rearranging the vectorization of the action set $\mathcal{X}'$ and the unknown $\Theta'^*$ as we have shown in Eqn. (3) and (4) for G-ESTT, we can simply exclude, rather than penalize, the subspaces that are complementary to the rows and columns of $\widehat{\Theta}$. In other words, we could remove the last $p - k$ entries directly, i.e. Eqn. (13) and (14). Intriguingly, not only can we get a low-dimensional ($k$) generalized linear bandit problem in stage 2, where redundant dimensions are excluded and hence any state-of-the-art algorithms could be readily invoked. But also we obtain the optimal regret bound of scale $\tilde{O}(dr\sqrt{T})$ again as shown in Table 1 with $D_r r = O(1/\sqrt{r})$. Besides, G-ESTS requires exceedingly less computation compared with other existing algorithms since consequently we get a low-dimensional bandit problem. In summary, our G-ESTS is superior regarding both theoretical guarantee and computational efficiency.

### 4.3.1 Overall regret of G-ESTS

**Theorem 4.3.** (Regret of G-ESTS) *Suppose we set $T_1 \asymp \sqrt{d_1 d_2 r T \log((d_1 + d_2)/\delta)}/D_{rr}$, and we invoke any efficient generalized linear bandit algorithm with regret bound $\tilde{O}(k\sqrt{T})$ [2] in stage 2 with $p = d_1 d_2, k = (d_1 + d_2)r - r^2$, and the reduced arm sets $\mathcal{X}_{0,sub}$ and available parameter space $\Theta_{0,sub}$. The overall regret of G-ESTS is, with probability at least $1 - \delta$,*

$$Regret_T = \tilde{O}\left((\frac{\sqrt{r d_1 d_2}}{D_{rr}} + k)\sqrt{T}\right)$$

According to Theorem 4.2 and 4.3, both G-ESTT and G-ESTS could achieve an improved $\tilde{O}(dr\sqrt{T})$ regret bound with $D_{rr} = \Theta(1/\sqrt{r})$. Surprisingly, this matches the lower bound of low-rank matrix bandits presented in [26] up to logarithm terms. To the best of our knowledge, our methods are the first two methods that attain this optimal rate. In addition, given the finite arm set, our G-ESTS

---

[2] Modern generalized linear bandit algorithms can achieve $\tilde{O}(k\sqrt{T})$ bound of regret.

Table 2: Time in minutes required to make decisions all over round $T$ in simulations (480 arms).

| $d$ | 10 | | 12 | |
|---|---|---|---|---|
| $r$ | 1 | 2 | 1 | 2 |
| G-ESTS | 39.46 | 45.41 | 41.28 | 48.52 |
| G-ESTT | 516.14 | 531.95 | 520.25 | 539.83 |
| SGD-TS | 99.57 | 101.34 | 101.82 | 104.42 |
| LowESTR | 401.88 | 419.15 | 410.31 | 425.92 |

framework can even achieve a better regret bound of order $\tilde{O}(\sqrt{d_1 d_2 rT}/D_{rr})$ (detailed proof in Appendix F.2), and this result is better than the lower regret bound with infinite arms by a $k\sqrt{T}$ term.

In the following experiments, We will implement the SGD-TS algorithm [9] in stage 2 of G-ESTS since SGD-TS could efficiently proceed with only $O(dT)$ complexity for $d-$dimensional features over $T$ rounds. Therefore, the total computational complexity of stage 2 is at most $O(T_2(d_1 + d_2)r)$, which is significantly less than that of other methods for low-rank matrix bandits (e.g. LowESTR [26]). And the total time complexity of G-ESTS would only scale $O(T_1 d_1 d_2/\epsilon^2 + T_2(d_1 + d_2)r)$ where $\epsilon$ is the accuracy for subgradient methods in stage 1. This fact also firmly validates the practical superiority of our G-ESTS approach. We naturally believe that this G-ESTS framework can be easily implemented in the linear setting as a special case of GLM, where in stage 2 one can utilize any linear bandit algorithm accordingly. In addition, we can easily modify our approaches for the contextual setting by merely transforming the action sets at each iteration with the same regret bound. More details with pseudo-codes for the contextual case are in Appendix H.3.

## 5  Experiments

In this section, we show by simulation experiments that our proposed G-ESTT (with LowGLM-UCB), G-ESTS (with SGD-TS) outperform existing algorithms for the generalized low-rank matrix bandit problems. Since we are the first to propose a practical algorithm for this problem, currently there is no existing literature for comparison. In order to validate the advantage of utilizing low-rank structure and generalized reward functions, we compare with the original SGD-TS after naïvely flattening the $d_1$ by $d_2$ matrices without using the low-rank structure, and LowESTR [26], which works well for linear low-rank matrix bandits.

We simulate a dataset with $d_1 = d_2 = 10\,(12)$ and $r = 1\,(2)$: when $r = 1$, we set the diagonal matrix $\Theta^*$ as $\text{diag}(\Theta^*) = (0.8, 0, \cdots, 0)$. When $r = 2$, we set $\Theta^* = v_1 v_1^\top + v_2 v_2^\top$ for two random orthogonal vectors $v_1, v_2$ with $\|v_1\|_2 = \|v_2\|_2 = 3$. For arms we draw $480\,(1000)$ random matrices from $\{X \in \mathbb{R}^{d_1 \times d_2} : \|X\|_F \leq 1\}$, and we build a logistic model where the payoff $y_t$ is drawn from a Bernoulli distribution with mean $\mu(X_t^\top \theta^*)$. More details on the hyper-parameter tuning are in Appendix I. Each experiment is repeated 100 times for credibility and the average regret, along with standard deviation, is displayed in Figure 1. Note that our experiments are more comprehensive than those in [26]. And due to the expensive time complexity of UCB-based baselines (Table 2), it is formidable for us to increase $d$ here.

From the plots, we observe that our algorithms G-ESTT and G-ESTS always achieve less regret compared with LowESTR and SGD-TS in all four scenarios consistently. Intriguingly, in the warm-up period SGD-TS incurs less regret compared with our methods due to the sacrifice of random sampling in stage 1, but our proposed framework quickly overtakes SGD-TS after utilizing the low-rank structure as desired. This phenomenon exactly coincides with our theory. Notice that G-ESTT is slightly better than G-ESTS in the case for $r = 2$ especially in the very beginning of stage 2, and we believe it is because that our G-ESTT could reutilize the actions in stage 1 and hence could yield more robust performance when switching to stage 2. However, G-ESTS would gradually catch up with G-ESTT in the long run as expected. Besides, it costs G-ESTS extremely less running time than other existing methods to update the decisions due to its dimensional reduction as shown in Table 2. We also observe that the cumulative regret of G-ESTS tends to become better eventually if we increase $T_1$ decently. (Further investigation and plots for 1000 arms are in Appendix I.) Moreover, to pre-check the efficiency of our Stein's lemma-based method for subspace estimation shown in Eqn.

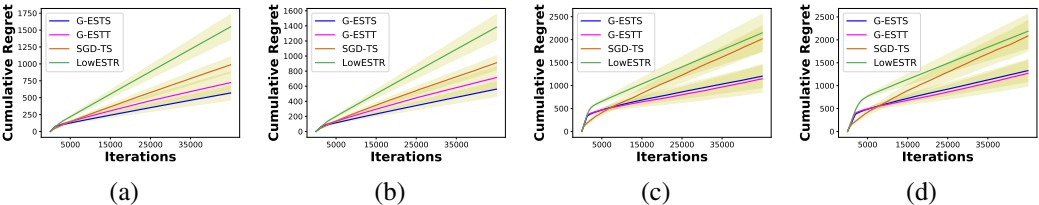

Figure 1: Plots of regret curves of algorithm G-ESTS, G-ESTT, SGD-TS and LowESTR under four settings (480 arms). (a): diagonal $\Theta^*$ $d_1 = d_2 = 10, r = 1$; (b): diagonal $\Theta^*$ $d_1 = d_2 = 12, r = 1$; (c): non-diagonal $\Theta^*$ $d_1 = d_2 = 10, r = 2$; (d): non-diagonal $\Theta^*$ $d_1 = d_2 = 12, r = 2$.

(6), we also tried some other low-rank subspace detection algorithms for comparison. The details are also deferred to Appendix I.4 due to the space limit.

## 6 Conclusion

In this paper, we discussed the generalized linear low-rank matrix bandit problem. We proposed two novel and efficient frameworks called G-ESTT and G-ESTS, and both methods could achieve an improved $\widetilde{O}(dr\sqrt{T})$ bound of regret under some mild conditions, which matches the optimal lower bound of our problem setting up to polylogarithmic terms. The practical superiority of our proposed frameworks is also validated under comprehensive experiments.

There are several directions for our future work. Firstly, although the GLM has achieved tremendous success in various settings, recently some other models [39] that are proved to be powerful were proposed, so we can study these modern frameworks in the low-rank matrix case. Secondly, it seems more reasonable to continuously update the subspace estimation via online SVD over the total time horizon [17] on-the-fly with a randomized strategy. Therefore, we can consider something like arbitrarily choosing an admissible subset of the action set at each round [31] without hurting the regret too much.

## Acknowledgments and Disclosure of Funding

We are grateful for the constructive comments from the anonymous reviewers and area chair. This work was partially supported by the National Science Foundation under grants CCF-1934568, DMS-1811405, DMS-1811661, DMS-1916125, DMS-2113605, DMS-2210388, IIS-2008173, and IIS-2048280. CJH is also supported by Samsung, Google, Sony and the Okawa Foundation.

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
