# Appendix

## A   Clarification about $\sigma_0^2$

It is a common assumption that the random noise $\eta_t$ in Eqn. (2) is a sub-Gaussian random variable in GLM, and here we would briefly explain this assumption.

**Lemma A.1.** (Sub-Gaussian property for GLM residuals) *For any generalized linear model with a probability density function or probability mass function of the canonical form*

$$f(Y = y; \theta, \phi) = \exp\left(\frac{y\theta - b(\theta)}{\phi} + c(y, \phi)\right),$$

*where the function $b(\cdot)$ is Lipschitz with parameter $k_\mu$. Then we can conclude that the random variable $(Y - b'(\theta)) = (Y - \mu(\theta))$ satisfies sub-Gaussian property with parameter at most $\sqrt{\phi\, k_\mu}$.*

*Proof.* We prove the Lemma A.1 based on its definition directly. For any $t \in \mathbb{R}$, we have:

$$
\begin{aligned}
\mathbb{E}\left[\exp\{t\,(Y - b'(\theta))\}\right] &= \int_{-\infty}^{+\infty} \exp\left\{t(y - b'(\theta)) + \frac{y\theta - b(\theta)}{\phi} + c(y, \phi)\right\} dy \\
&= \int_{-\infty}^{+\infty} \exp\left\{\frac{(\theta + \phi t)y - b(\theta + \phi t)}{\phi} + c(y, \phi)\right\} \\
&\qquad \times \exp\left\{\frac{b(\theta + \phi t) - b(\theta) - \phi t b'(\theta)}{\phi}\right\} dy \\
&= \exp\left\{\frac{b(\theta + \phi t) - b(\theta) - \phi t b'(\theta)}{\phi}\right\} \\
&\overset{(i)}{=} \exp\left\{\frac{t^2 \phi\, b''(\theta + \delta\phi t)}{2}\right\} \leq \exp\left\{\frac{t^2\, \phi\, k_\mu}{2}\right\} := \exp\left\{\frac{t^2\, \sigma_0^2}{2}\right\},
\end{aligned}
$$

where the equality (i) is based on the remainder of Taylor expansion. $\qquad\square$

This theorem tells us that it is a standard assumption that the noise $\eta_t$ in Eqn. (2) is a sub-Gaussian random variable. For instance, if we assume the inverse link function $\mu(\cdot)$ is globally Lipschitz with parameter $k_\mu$, we can simply take $\sigma_0^2 = k_\mu \phi$. And this assumption also widely holds under a class of GLMs such as the most popular Logistic model.

## B   Proof of Theorem 4.1

### B.1   Useful Lemmas

**Lemma B.1.** (Sub-guassian moment bound) *For sub-Gaussian random variable $X$ with parameter $\sigma^2$, i.e.*

$$\mathbb{E}(\exp(sX)) \leq \exp\left(\frac{\sigma^2 s^2}{2}\right), \quad \forall s \in \mathbb{R}.$$

*Then we have $Var(X) = \mathbb{E}(X^2) \leq 4\sigma^2$.*

*Proof.* It holds that,

$$
\begin{aligned}
\mathbb{E}(X^2) &= \int_0^{+\infty} P(X^2 > t)\, dt \\
&= \int_0^{+\infty} P(|X| > \sqrt{t})\, dt \\
&\leq 2 \int_0^{+\infty} \exp\left(\frac{-t^2}{4\sigma^2}\right) dt \\
&= 4\sigma^2 \int_0^{+\infty} e^{-u}\, du, \quad u = t/(2\sigma^2) \\
&= 4\sigma^2
\end{aligned}
$$

$\qquad\square$

**Lemma B.2.** *(Generalized Stein's Lemma, [8]) For a random variable $X$ with continuously differentiable density function $p : \mathbb{R}^d \to \mathbb{R}$, and any continuously differentiable function $f : \mathbb{R}^d \to \mathbb{R}$. If the expected values of both $\nabla f(X)$ and $f(X) \cdot S(X)$ regarding the density $p$ exist, then they are identical, i.e.*

$$\mathbb{E}[f(X) \cdot S(X)] = \mathbb{E}[\nabla f(X)].$$

This is a very famous result in the area of Stein's method, and we would omit its proof.

**Lemma B.3.** *([27]) Let $Y_1, \ldots, Y_n \in \mathbb{R}^{d_1 \times d_2}$ be a sequence of independent real random matrices, and assume that*

$$\sigma_n^2 \geq \max \left( \left\| \sum_{j=1}^n \mathbb{E}(Y_j Y_j^\top) \right\|_{op}, \left\| \sum_{j=1}^n \mathbb{E}(Y_j^\top Y_j) \right\|_{op} \right).$$

*Then for any $t \in \mathbb{R}^+$ and $\nu \in \mathbb{R}^+$, it holds that,*

$$P \left( \left\| \sum_{j=1}^n \tilde{\psi}_\nu(Y_j) - \sum_{j=1}^n \mathbb{E}(Y_j) \right\|_{op} \geq t\sqrt{n} \right) \leq 2(d_1 + d_2) \exp \left( \nu t \sqrt{n} + \frac{\nu^2 \sigma_n^2}{2} \right)$$

The detailed proof of this lemma is based on a series of work proposed in [27]. And we would omit it here as well. Based on Lemma B.2 and B.3, we would propose the following Lemma B.4 adapted from the work in [37]. And this Lemma serves as a crux for the proof of Theorem 4.1.

**Lemma B.4.** *$L : \mathbb{R}^{d_1 \times d_2} \to \mathbb{R}$ is the loss function defined in Eqn. (6). Then by setting*

$$t = \sqrt{2 d_1 d_2 M (4\sigma_0^2 + S_f^2) \log \left( \frac{2(d_1 + d_2)}{\delta} \right)},$$

$$\nu = \frac{t}{(4\sigma_0^2 + S_f) M d_1 d_2 \sqrt{T_1}} = \sqrt{\frac{2 \log \left( \frac{2(d_1 + d_2)}{\delta} \right)}{T_1 d_1 d_2 M (4\sigma_0^2 + S_f^2)}},$$

*we have with probability at least $1 - \delta$, it holds that*

$$P \left( \|\nabla L(\mu^* \Theta^*)\|_{op} \geq \frac{2t}{\sqrt{T_1}} \right) \leq \delta,$$

*where $\mu^* = \mathbb{E}[\mu'(\langle X, \Theta^* \rangle)] \geq c_\mu > 0$.*

*Proof.* Based on the definition of our loss function $L(\cdot)$ in Eqn. (6), we have that

$$\nabla_x L(\mu^* \Theta^*) = 2\mu^* \Theta^* - \frac{2}{T_1} \sum_{i=1}^{T_1} \tilde{\psi}_\nu(y \cdot S(x))$$

$$= 2\mathbb{E}[\mu'(\langle X_1, \Theta^* \rangle)]\Theta^* - \frac{2}{T_1} \sum_{i=1}^{T_1} \tilde{\psi}_\nu(y_i \cdot S(X_i))$$

$$\overset{(i)}{=} 2\mathbb{E}[\mu(\langle X_1, \Theta^* \rangle) S(X_1)] - \frac{2}{T_1} \sum_{i=1}^{T_1} \tilde{\psi}_\nu(y_i \cdot S(X_i))$$

$$\overset{(ii)}{=} 2 \left[ \mathbb{E}(Y_1 \cdot S(X_1)) - \frac{1}{T_1} \sum_{i=1}^{T_1} \tilde{\psi}_\nu(y_i \cdot S(X_i)) \right]$$

where we have (i) due to the generalized Stein's Lemma (Lemma B.2), and (ii) comes from the fact that the random noise $\eta_1 = y_1 - \mu(\langle X_1, \Theta^* \rangle)$ is zero-mean and independent with $X_1$. Therefore, in order to implement the Lemma B.3, we can see that it suffices to get $\sigma^2$ defined as:

$$\sigma^2 = \max \left( \left\| \sum_{j=1}^n \mathbb{E}[y_j^2 S(X_j) S(X_j)^\top] \right\|_{op}, \left\| \sum_{j=1}^n \mathbb{E}[y_j^2 S(X_j)^\top S(X_j)] \right\|_{op} \right).$$

It holds that,

$$
\left\| \sum_{j=1}^{T_1} \mathbb{E}[y_j^2 S(X_j)S(X_j)^\top] \right\|_{\mathrm{op}} \leq T_1 \times \left\| \mathbb{E}[y_1^2 S(X_1)S(X_1)^\top] \right\|_{\mathrm{op}}
$$

$$
= T_1 \times \left\| \mathbb{E}[(\eta_1 + \mu(\langle X_1, \Theta^* \rangle))^2 S(X_1)S(X_1)^\top] \right\|_{\mathrm{op}}
$$

$$
= T_1 \times \left\| \mathbb{E}[\eta_1^2 S(X_1)S(X_1)^\top] + \mathbb{E}[\mu(\langle X_1, \Theta^* \rangle))^2 S(X_1)S(X_1)^\top] \right\|_{\mathrm{op}}
$$

$$
= T_1 \times \left\| \mathbb{E}(\eta_1^2)\mathbb{E}[S(X_1)S(X_1)^\top] + \mathbb{E}[\mu(\langle X_1, \Theta^* \rangle))^2 S(X_1)S(X_1)^\top] \right\|_{\mathrm{op}}
$$

$$
\overset{(i)}{\leq} T_1 \times \left\| 4\sigma_0^2 \, \mathbb{E}[S(X_1)S(X_1)^\top] + S_f^2 \, \mathbb{E}[S(X_1)S(X_1)^\top] \right\|_{\mathrm{op}}
$$

$$
= (4\sigma_0^2 + S_f^2)T_1 \times \left\| \mathbb{E}[S(X_1)S(X_1)^\top] \right\|_{\mathrm{op}}
$$

where the inequality (i) comes from the fact that $|\mu(\langle X_1, \Theta^* \rangle)| \leq S_f$, and $S(X_1)S(X_1)^\top$ is always positive semidefinite. Next, since we know that $\mathbb{E}[S(X_1)S(X_1)^\top]$ is always symmetric and positive semidefinite, and hence we have

$$
\left\| \mathbb{E}[S(X_1)S(X_1)^\top] \right\|_{\mathrm{op}} \leq \left\| \mathbb{E}[S(X_1)S(X_1)^\top] \right\|_{\mathrm{nuc}} = \mathbf{trace}(\mathbb{E}[S(X_1)S(X_1)^\top])
$$

$$
= \mathbb{E}[\mathbf{trace}(S(X_1)S(X_1)^\top)] = \mathbb{E}(\sum_{i=1}^{d_1}\sum_{j=1}^{d_2} S_{ij}(X_1)^2)
$$

$$
\leq d_1 d_2 M
$$

Therefore, we have that

$$
\left\| \sum_{j=1}^{T_1} \mathbb{E}[y_j^2 S(X_j)S(X_j)^\top] \right\|_{\mathrm{op}} \leq (4\sigma_0^2 + S_f^2)d_1 d_2 T_1 M.
$$

And similarly, we can prove that

$$
\left\| \sum_{j=1}^{T_1} \mathbb{E}[y_j^2 S(X_j)^\top S(X_j)] \right\|_{\mathrm{op}} \leq (4\sigma_0^2 + S_f^2)d_1 d_2 T_1 M.
$$

Therefore, we can take $\sigma^2 = (4\sigma_0^2 + S_f^2)d_1 d_2 T_1 M$ consequently. By using Lemma B.3, we have

$$
P\left( \|\nabla L(\mu^*\Theta^*)\|_{\mathrm{op}} \geq \frac{2t}{\sqrt{T_1}} \right) \leq 2(d_1 + d_2)\exp\left( -\nu t\sqrt{T_1} + \frac{\nu^2(4\sigma_0^2 + S_f^2)Md_1 d_2 T_1}{2} \right)
$$

By plugging the values of $t$ and $\nu$ in Lemma B.4, we finish the proof. $\qquad\square$

## B.2 Proof of Theorem 4.1

Since the estimator $\widehat{\Theta}$ minimizes the regularized loss function defined in Eqn. (6), we have

$$
L(\widehat{\Theta}) + \lambda_{T_1}\left\| \widehat{\Theta} \right\|_{\mathrm{nuc}} \leq L(\mu^*\Theta^*) + \lambda_{T_1}\|\mu^*\Theta^*\|_{\mathrm{nuc}}.
$$

And due to the fact that $L(\cdot)$ is a quadratic function, we have the following expression based on multivariate Taylor's expansion:

$$
L(\widehat{\Theta}) - L(\mu^*\Theta^*) = \langle \nabla L(\mu^*\Theta^*), \Theta \rangle + 2\|\Theta\|_F^2, \quad \text{where } \Theta = \widehat{\Theta} - \mu^*\Theta^*.
$$

By rearranging the above two results, we can deduce that

$$
2\|\Theta\|_F^2 \leq -\langle \nabla L(\mu^*\Theta^*), \Theta \rangle + \lambda_{T_1}\|\mu^*\Theta^*\|_{\mathrm{nuc}} - \lambda_{T_1}\left\| \widehat{\Theta} \right\|_{\mathrm{nuc}}
$$

$$
\overset{(i)}{\leq} \|\nabla L(\mu^*\Theta^*)\|_{\mathrm{op}}\|\Theta\|_{\mathrm{nuc}} + \lambda_{T_1}\|\mu^*\Theta^*\|_{\mathrm{nuc}} - \lambda_{T_1}\left\| \widehat{\Theta} \right\|_{\mathrm{nuc}}, \tag{15}
$$

where (i) comes from the duality between matrix operator norm and nuclear norm. Next, we represent the saturated SVD of $\Theta^*$ in the main paper as $\Theta^* = UDV^\top$ where $U \in \mathbb{R}^{d_1 \times r}$ and $V \in \mathbb{R}^{d_2 \times r}$, and here we would work on its full version, i.e.

$$\Theta^* = (U, U_\perp) \begin{pmatrix} D & 0 \\ 0 & 0 \end{pmatrix} (V, V_\perp)^\top = (U, U_\perp) D^* (V, V_\perp)^\top,$$

where we have $U_\perp \in \mathbb{R}^{d_1 \times (d_1 - r)}$, $D^* \in \mathbb{R}^{d_1 \times d_2}$ and $V_\perp \in \mathbb{R}^{d_2 \times (d_2 - r)}$. Furthermore, we define

$$\Lambda = (U, U_\perp)^\top \Theta (V, V_\perp) = \begin{pmatrix} U^\top \Theta V & U^\top \Theta V_\perp \\ U_\perp^\top \Theta V & U_\perp^\top \Theta V_\perp \end{pmatrix} = \Lambda_1 + \Lambda_2$$

where we write

$$\Lambda_1 = \begin{pmatrix} 0 & 0 \\ 0 & U_\perp^\top \Theta V_\perp \end{pmatrix}, \quad \Lambda_2 = \begin{pmatrix} U^\top \Theta V & U^\top \Theta V_\perp \\ U_\perp^\top \Theta V & 0 \end{pmatrix}$$

Afterwards, it holds that

$$
\begin{aligned}
\left\| \widehat{\Theta} \right\|_{\mathrm{nuc}} &= \| \mu^* \Theta^* + \Theta \|_{\mathrm{nuc}} = \left\| (U, U_\perp)(\mu^* D^* + \Lambda)(V, V_\perp)^\top \right\|_{\mathrm{nuc}} \\
&= \| \mu^* D^* + \Lambda \|_{\mathrm{nuc}} + \| \mu^* D^* + \Lambda_1 + \Lambda_2 \|_{\mathrm{nuc}} \\
&\geq \| \mu^* D^* + \Lambda_1 \|_{\mathrm{nuc}} - \| \Lambda_2 \|_{\mathrm{nuc}} \\
&= \| \mu^* D \|_{\mathrm{nuc}} + \| \Lambda_1 \|_{\mathrm{nuc}} - \| \Lambda_2 \|_{\mathrm{nuc}} \\
&= \| \mu^* \Theta^* \|_{\mathrm{nuc}} + \| \Lambda_1 \|_{\mathrm{nuc}} - \| \Lambda_2 \|_{\mathrm{nuc}},
\end{aligned}
$$

which implies that

$$\| \mu^* \Theta^* \|_{\mathrm{nuc}} - \left\| \widehat{\Theta} \right\|_{\mathrm{nuc}} \leq \| \Lambda_2 \|_{\mathrm{nuc}} - \| \Lambda_1 \|_{\mathrm{nuc}} \tag{16}$$

Combine Eqn. (15) and (16), we have that

$$2 \| \Theta \|_F^2 \leq \left( \| \nabla L(\mu^* \Theta^*) \|_{\mathrm{op}} + \lambda_{T_1} \right) \| \Lambda_2 \|_{\mathrm{nuc}} + \left( \| \nabla L(\mu^* \Theta^*) \|_{\mathrm{op}} - \lambda_{T_1} \right) \| \Lambda_1 \|_{\mathrm{nuc}}$$

Then, we refer to the setting in our Lemma B.4, and we choose $\lambda = 4t/\sqrt{T_1}$ where the value of $t$ is determined in Lemma B.4, i.e.

$$\lambda_{T_1} = 4 \sqrt{\frac{2(4\sigma_0^2 + S_f^2) M d_1 d_2 \log(2(d_1 + d_2)/\delta)}{T_1}},$$

we know that $\lambda_{T-1} \geq 2 \| \nabla L(\mu^* \Theta^*) \|_{\mathrm{op}}$ with probability at least $1 - \delta$ for any $\delta \in (0, 1)$. Therefore, with probability at least $1 - \delta$, we have

$$2 \| \Theta \|_F^2 \leq \frac{3}{2} \lambda_{T_1} \| \Lambda_2 \|_{\mathrm{nuc}} - \frac{1}{2} \lambda_{T_1} \| \Lambda_1 \|_{\mathrm{nuc}} \leq \frac{3}{2} \lambda_{T_1} \| \Lambda_2 \|_{\mathrm{nuc}}$$

. Since we can easily verify that the rank of $\Lambda_2$ is at most $2r$, and by using Cauchy-Schwarz Inequality we have that

$$2 \| \Theta \|_F^2 \leq \frac{3}{2} \lambda_{T_1} \sqrt{2r} \| \Lambda_2 \|_F \leq \frac{3}{2} \lambda_{T_1} \sqrt{2r} \| \Lambda \|_F = \frac{3}{2} \lambda_{T_1} \sqrt{2r} \| \Theta \|_F,$$

which implies that

$$\| \Theta \|_F \leq \frac{3}{4} \sqrt{2r} \lambda_{T_1} = 6 \sqrt{\frac{(4\sigma_0^2 + S_f^2) M d_1 d_2 r \log(\frac{2(d_1 + d_2)}{\delta})}{T_1}},$$

and it concludes our proof. $\qquad \square$

## C   Theorem C.1 and its analysis

### C.1   Theorem C.1

**Theorem C.1.** (Regret of LowGLM-UCB) *Under Assumption 3.4 and 3.5, for any fixed failure rate $\delta \in (0, 1)$, if we run the LowGLM-UCB algorithm with $\rho_t(\delta) = \alpha_{t+T_1}(\delta/2)$ and*

$$\lambda_\perp \asymp \frac{c_\mu S_0^2 T}{k \log(1 + \frac{c_\mu S_0^2 T}{k \lambda_0})},$$

*then the bound of regret for LowGLM-UCB ($Regret_{T_2}$) achieves $\widetilde{O}(k\sqrt{T} + TS_\perp)$, with probability at least $1 - \delta$.*

## C.2 Proposition C.2 with its proof

We firstly present the following important Proposition C.2 for obtaining the upper confidence bound.

**Proposition C.2.** *For any $\delta, t$ such that $\delta \in (0,1)$, $t \geq 2$, and for $\beta_t^x(\delta)$ defined in Eqn.* (11) *and* (12)*, with probablity $1 - \delta$, it holds that*

$$|\mu(x^\top \theta^*) - \mu(x^\top \hat{\theta}_t)| \leq \beta_{t+T_1}^x(\delta), \tag{17}$$

*simultaneously for all $x \in \mathbb{R}$ and all $t \geq 2$.*

### C.2.1 Technical Lemmas

**Lemma C.3.** (Adapted from Abbasi-Yadkori et al., 2011, Theorem 1) *Let $\{\mathcal{F}_t\}_{t=0}^\infty$ be a filtration and $\{x_t\}_{t=0}^\infty$ be an $\mathbb{R}^d$-valued stochastic process adapted to $\mathcal{F}_t$. Let $\{\eta_t\}_{t=0}^\infty$ be a real-valued stochastic process such that $\eta_t$ is adapted to $\mathcal{F}_t$ and is conditionally $\sigma_0$-sub-Gaussian for some $\sigma_0 > 0$, i.e.*

$$\mathbb{E}[\exp(\lambda \eta_t)|\mathcal{F}_t] \leq \exp\left(\frac{\lambda^2 \sigma_0^2}{2}\right), \qquad \forall \lambda \in \mathbb{R}.$$

*Consider the martingale $S_t = \sum_{k=1}^t \eta_k x_k$ and the process $V_t = \sum_{k=1}^t x_k x_k^\top + \Lambda$ when $t \geq 2$. And $\Lambda$ is fixed and independent with sample random variables after time $m$. For any $\delta > 0$, with probability at least $1 - \delta$, we have the following result simultaneously for all $t \geq m + 1$:*

$$\|S_t\|_{V_t^{-1}} \leq \sigma_0 \sqrt{\log(\det(V_t)) - \log(\delta^2 \det(\Lambda))}.$$

We defer the proof for this lemma to Section C.2.3 since a lot of technical details are involved.

**Lemma C.4.** *For any two symmetric positive definite matrix $A, B \in R^{p \times p}$ such that $A \preceq B$, we have $AB^{-1}A \preceq A$.*

*Proof.* Since $A \preceq B$ and both of them are invertible matrices, we have $B^{-1} \preceq A^{-1}$ directly based on positive definiteness property. Conjugate with $A$ on both sides we can directly obtain $AB^{-1}A \preceq A$. $\square$

**Lemma C.5.** (Valko et al., 2014, Lemma 5) *For any $T \geq 1$, let $V_{T+1} = \sum_{i=1}^T x_i x_i^\top + \Lambda \in R^p$ where $\Lambda = diag\{\lambda_1, \ldots, \lambda_p\}$. And we assume that $\|x_i\|_2 \leq S$. Then:*

$$\log \frac{|V_{T+1}|}{|\Lambda|} \leq \max_{\{t_i\}_{i=1}^p} \sum_{i=1}^p \log\left(1 + \frac{S^2 t_i}{\lambda_i}\right),$$

*where the maximum is taken over all possible positive real numbers $\{t_i\}_{i=1}^p$ such that $\sum_{i=1}^p t_i = T$*

*Proof.* We aim to bound the determinant $|V_{T+1}|$ under the coordinate constrains $\|x_i\|_2 \leq S$. Let's denote

$$U(x_1, \ldots, x_T) = |\Sigma + \sum_{t=1}^T x_t x_t^\top|.$$

Based on the property of the sum of rank-1 matrices (e.g. Valko et al., 2014, Lemma 4), we know that the maximum of $U(x_1, \ldots, x_T)$ is reached when all $x_t$ are aligned with the axes:

$$U(x_1, \ldots, x_T) = \max_{\substack{x_1, \ldots, x_T; \\ x_t \in S \cdot \{e_1, \ldots, e_N\}}} |\Sigma + \sum_{t=1}^T x_t x_t^\top| = \max_{\substack{t_1, \ldots, t_N \text{ positive integers;} \\ \sum_{i=1}^N t_i = T}} |diag(\lambda_i + t_i)|$$

$$\leq \max_{\substack{t_1, \ldots, t_N \text{ positive integers;} \\ \sum_{i=1}^N t_i = S^2 T}} \prod_{i=1}^N (\lambda_i + S^2 t_i).$$

$\square$

### C.2.2 Proof of Proposition C.2

*Proof.* Recall our definition of $g_t(\theta)$ and its gradient accordingly as

$$g_t(\theta) = \sum_{i=1}^{T_1} \mu(x_{s_1,i}{}^\top \theta)x_{s_1,i} + \sum_{k=1}^{t-1} \mu(x_k^\top \theta)x_k + \Lambda\theta,$$

$$\nabla_\theta g_t(\theta) = \sum_{i=1}^{T_1} \mu'(x_{s_1,i}{}^\top \theta)x_{s_1,i}x'_{s_1,i} + \sum_{k=1}^{t-1} \mu'(x_k^\top \theta)x_kx_k^\top + \Lambda \overset{(i)}{\succeq} c_\mu M_t(c_\mu), \qquad (18)$$

where the relation (i) holds if $\theta \in \Theta_0$. Based on Assumptions, we know the gradient $\nabla_\theta g_t(\theta)$ is continuous. Then the Fundamental Theorem of Calculus will imply that

$$g_t(\theta^*) - g_t(\hat{\theta}_t) = G_t(\theta^* - \hat{\theta}_t),$$

where

$$G_t = \int_0^1 \nabla_\theta g_t(s\theta^* + (1-s)\hat{\theta}_t)\, ds.$$

Since we assume that the inverse link function $\mu(\cdot)$ is $k_\mu$−Lipshitz, and the matrix $G_t$ is always invertible due to the fact that at least we have $G_t \succeq \Lambda$, we can obtain the following result. Notice the inequality (i) comes from the fact that $G_t \succeq c_\mu M_t(c_\mu)$ and hence $M_t(c_\mu)^{-1}/c_\mu \succeq G_t^{-1}$.

$$|\mu(x^\top \theta^*) - \mu(x^\top \hat{\theta}_t)| \leq k_\mu |x^\top(\theta^* - \hat{\theta}_t)| = k_\mu |x^\top G_t^{-1}(g_t(\theta^*) - g_t(\hat{\theta}_t))|$$

$$\leq k_\mu \|x\|_{G_t^{-1}} \left\| g_t(\theta^*) - g_t(\hat{\theta}_t) \right\|_{G_t^{-1}} \overset{(i)}{\leq} \frac{k_\mu}{c_\mu} \|x\|_{M_t(c_\mu)^{-1}} \left\| g_t(\theta^*) - g_t(\hat{\theta}_t) \right\|_{M_t(c_\mu)^{-1}}.$$

In addition, based on the definition of $\hat{\theta}_t$ in Equation (10), we have $g_t(\hat{\theta}_t) - g_t(\theta^*) = \sum_{k=1}^{T_1}(y_{s_1,k} - \mu(x_{s_1,k}^\top \theta^*))x_{s_1,k} + \sum_{k=1}^{t-1}(y_k - \mu(x_k^\top \theta^*))x_k - \Lambda\theta^* = \sum_{k=1}^{T_1} \eta_{s_1,k}x_{s_1,k} + \sum_{k=1}^{t-1} \eta_k x_k - \Lambda\theta^*$. Therefore,

$$|\mu(x^\top \theta^*) - \mu(x^\top \hat{\theta}_t)| \leq \frac{k_\mu}{c_\mu} \|x\|_{M_t(c_\mu)^{-1}} \left\| g_t(\hat{\theta}_t) - g_t(\theta^*) \right\|_{M_t^{-1}(c_\mu)}$$

$$\leq \frac{k_\mu}{c_\mu} \|x\|_{M_t(c_\mu)^{-1}} \left( \left\| \sum_{k=1}^{T_1} \eta_{s_1,k}x_{s_1,k} + \sum_{k=1}^{t-1} \eta_k x_k \right\|_{M_t^{-1}(c_\mu)} + \|\Lambda\theta^*\|_{M_t^{-1}(c_\mu)} \right). \qquad (19)$$

Now, let's use Lemma C.3 to bound the term $\left\| \sum_{k=1}^{T_1} \eta_{s_1,k}x_{s_1,k} + \sum_{k=1}^{t-1} \eta_k x_k \right\|_{M_t^{-1}(c_\mu)}$. If we define the filtration $\mathcal{F}_t := \{\{x_t, x_{t-1}, \eta_{t-1}, \ldots, x_1, \eta_1\} \cup \{x_{s_1,k}, \eta_{s_1,k}\}_{k=1}^{T_1}\}$, then for any $\delta \in (0,1)$, with probability $1 - \delta$, it holds that for all $t \geq 2$,

$$\left\| \sum_{k=1}^{T_1} \eta_{s_1,k}x_{s_1,k} + \sum_{k=1}^{t-1} \eta_k x_k \right\|_{M_t^{-1}(c_\mu)} \leq \sigma_0 \sqrt{\log\left(\frac{|M_t(c_\mu)|}{|\frac{\Lambda}{c_\mu}|}\right) - 2\log(\delta)},$$

where based on Lemma C.5,

$$\log\left(\frac{|M_t(c_\mu)|}{|\frac{\Lambda}{c_\mu}|}\right) \leq \max_{\substack{t_i \geq 0, \\ \sum_{i=1}^t t_i = t+T_1}} \sum_{i=1}^p \log\left(1 + \frac{c_\mu S_0^2 t_i}{\lambda_i}\right)$$

$$\leq k \log\left(1 + \frac{c_\mu S_0^2}{k\lambda_0}(t + T_1)\right) + (d - k)\log\left(1 + \frac{c_\mu S_0^2}{(d-k)\lambda_\perp}(t + T_1)\right)$$

$$\leq k \log\left(1 + \frac{c_\mu S_0^2}{k\lambda_0}(t + T_1)\right) + \frac{c_\mu S_0^2}{\lambda_\perp}(t + T_1). \qquad (20)$$

And next by Lemma C.4, we have

$$\|\Lambda\theta^*\|_{M_t^{-1}(c_\mu)} = c_\mu \left\|\frac{\Lambda}{c_\mu}\theta^*\right\|_{M_t^{-1}(c_\mu)} \leq \sqrt{c_\mu}\|\theta^*\|_\Lambda \leq \sqrt{c_\mu}(\sqrt{\lambda_0}S_0 + \sqrt{\lambda_\perp}S_\perp). \quad (21)$$

Combine Equation (20) and (21) into Equation (19), we finish our proof. $\qquad\square$

Since Equation (17) in Proposition C.2 holds simultaneously for all $x \in \mathbb{R}$ and $t \geq 1$, the following conclusion holds.

**Corollary C.6.** *For any random variable $z$ defined in $\mathbb{R}$, we have the following holds*

$$|\mu(z^\top\theta^*) - \mu(z^\top\hat{\theta}_t)| \leq \beta_{t+T_1}^z(\delta),$$

*with probability at least $1 - \delta$. Furthermore, for any sequence of random variable $\{z_t\}_{t=2}^T$, with probability $1 - \delta$ it holds that*

$$|\mu(z_t^\top\theta^*) - \mu(z_t^\top\hat{\theta}_t)| \leq \beta_{t+T_1}^{z_t}(\delta),$$

*simultaneously for all $t \geq 1$.*

### C.2.3 Proof of Lemma C.3

For the proof of Lemma C.3 we will need the following two lemmas, and we will use the same notations as in Lemma C.3 in this section.

**Lemma C.7.** *Let $\lambda \in \mathbb{R}^d$ be arbitrary and consider any $t \geq 0$*

$$M_t^\lambda = \exp\left(\sum_{s=1}^t \left[\frac{\eta_s(\lambda^\top x_s)}{\sigma_0} - \frac{1}{2}(\lambda^\top x_s)^2\right]\right).$$

*Let $\tau$ be a stopping time with respect to the filtration $\{\mathcal{F}_t\}_{t=0}^{+\infty}$. Then $M_t^\lambda$ is a.s. well defined and $\mathbb{E}(M_\tau^\lambda) \leq 1$.*

*Proof.* We claim that $\{M_t^\lambda\}$ is a supermartingale. Let

$$D_t^\lambda = \exp\left(\frac{\eta_s(\lambda^\top x_s)}{\sigma_0} - \frac{1}{2}(\lambda^\top x_s)^2\right)$$

Observe that by conditional $\sigma_0$-sub-Gaussianity of $\eta_t$ we have $\mathbb{E}[D_t^\lambda|\mathcal{F}_{t-1}] \leq 1$. Clearly, $D_t^\lambda$ and $M_t^\lambda$ is $\mathcal{F}_t$-measurable. Moreover,

$$\mathbb{E}[M_t^\lambda|\mathcal{F}_{t-1}] = \mathbb{E}[M_1^\lambda\cdots D_{t-1}^\lambda D_t^\lambda|\mathcal{F}_{t-1}] = D_1^\lambda\ldots D_{t-1}^\lambda\mathbb{E}[D_t^\lambda|\mathcal{F}_{t-1}] \leq M_{t-1}^\lambda,$$

which implies that $M_t^\lambda$ is a supermartingale with its expected value upped bounded by 1. To show that $M_t^\lambda$ is well defined. By the convergence theorem for nonnegative supermartingales, $\lim_{t\to\infty} M_t^\lambda$ is a.s. well-defined, which indicates that $M_\tau^\lambda$ is also well-defined for all $\tau \in \mathbb{N}^+ \cup \{+\infty\}$. By Fatou's Lemma, it holds that

$$\mathbb{E}[M_\tau^\lambda] = \mathbb{E}[\liminf_{t\to\infty} M_{\min\{t,\tau\}}^\lambda] \leq \liminf_{t\to\infty} \mathbb{E}[M_{\min\{t,\tau\}}^\lambda] \leq 1.$$

$\qquad\square$

**Lemma C.8.** *For any positive semi-definite matrix $P \in \mathbb{R}^{d\times d}$ and positive definite matrix $Q \in \mathbb{R}^{d\times d}$, and any $x, a \in \mathbb{R}^d$, it holds that*

$$\|x - a\|_P^2 + \|x\|_Q^2 = \|x - (P+Q)^{-1}Pa\|_{P+Q}^2 + \|a\|_P^2 - \|Pa\|_{(P+Q)^{-1}}^2.$$

This lemma could be easily proved based on elementary calculation and hence its proof would be omitted here.

**Lemma C.9.** *Let $\tau$ be a stopping time with $\tau > m$ on the filtration $\{\mathcal{F}_t\}_{t=0}^\infty$. Then for $\delta > 0$, with probability $1 - \delta$,*

$$||S_\tau||_{V_\tau^{-1}}^2 \leq 2\sigma_0^2 \log \left( \frac{det(V_\tau)^{1/2} det(\Lambda)^{-1/2}}{\delta} \right).$$

*Proof.* W.l.o.g., assume that $\sigma_0 = 1$. Denote

$$\tilde{V}_t = V_t - \Lambda = \sum_{s=1}^t x_s x_s^\top, \qquad M_t^\lambda = \exp \left( (\lambda^\top S_t) - \frac{1}{2} ||\lambda||_{\tilde{V}_t}^2 \right).$$

Note by Lemma C.7, we naturally have that $\mathbb{E}[M_t^\lambda] \leq 1$.

Since in round $m + 1$, we get the diagonal positive definite matrix $\Lambda$ with its elements independent with samples after round $m$. Let $z$ be a Gaussian random variable that is independent with other random variables after round $m$ with covariance $\Lambda^{-1}$. Define

$$M_t = \mathbb{E}[M_t^z | \mathcal{F}_\infty], \quad t > m,$$

where $\mathcal{F}_\infty$ is the tail $\sigma$-algebra of the filtration. Clearly, it holds that $\mathbb{E}[M_\tau] = \mathbb{E}[\mathbb{E}[M_\tau^z | z, \mathcal{F}_\infty]] \leq \mathbb{E}[1] \leq 1$. Let $f$ be the density of $z$ and for a positive definite matrix $P$ let $c(P) = \sqrt{(2\pi)^d / \det(P)}$. Then for $t > m$ it holds that,

$$M_t = \int_{\mathbb{R}^d} \exp \left( (\lambda^\top S_t) - \frac{1}{2} ||\lambda||_{\tilde{V}_t}^2 \right) f(\lambda) d\lambda$$

$$= \frac{1}{c(\Lambda)} \exp \left( \frac{1}{2} ||S_t||_{\tilde{V}_t^{-1}}^2 \right) \int_{\mathbb{R}^d} \exp \left( -\frac{1}{2} \left\{ \left\| \lambda - \tilde{V}_t^{-1} S_t \right\|_{\tilde{V}_t}^2 + ||\lambda||_\Lambda^2 \right\} \right) d\lambda.$$

Based on Lemma C.8, it holds that

$$\left\| \lambda - \tilde{V}_t^{-1} S_t \right\|_{\tilde{V}_t}^2 + ||\lambda||_\Lambda^2 = \left\| \lambda - V_t^{-1} S_t \right\|_{V_t}^2 + \left\| \tilde{V}_t^{-1} S_t \right\|_{\tilde{V}_t}^2 - ||S_t||_{V_t^{-1}}^2$$

, and this implies that

$$M_t = \frac{1}{c(\Lambda)} \exp \left( \frac{1}{2} ||S_t||_{V_t^{-1}}^2 \right) \int_{\mathbb{R}^d} \exp \left( -\frac{1}{2} \left\| \lambda - V_t^{-1} S_t \right\|_{V_t}^2 \right) d\lambda$$

$$= \left( \frac{\det(\Lambda)}{\det(V_t)} \right)^{1/2} \exp \left( -\frac{1}{2} \left\| \lambda - V_t^{-1} S_t \right\|_{V_t}^2 \right).$$

Now, from $\mathbb{E}[M_\tau] \leq 1$, we have that for $\tau > m$

$$P \left( ||S_\tau||_{V_\tau^{-1}}^2 > \log \left( \frac{\det(V_\tau)}{\delta^2 \det(\Lambda)} \right) \right) = P \left( \frac{\exp \left( \frac{1}{2} ||S_\tau||_{V_\tau^{-1}}^2 \right)}{\delta^{-1} (\det(V_\tau)/\det(\Lambda))^{1/2}} > 1 \right)$$

$$\leq \mathbb{E} \left[ \frac{\exp \left( \frac{1}{2} ||S_\tau||_{V_\tau^{-1}}^2 \right)}{\delta^{-1} (\det(V_\tau)/\det(\Lambda))^{1/2}} \right] \leq \mathbb{E}[M_\tau] \delta \leq \delta.$$

$\square$

Combining Lemma C.7-C.9. We now contruct a stopping time and define the bad event:

$$B_t(\delta) := \left\{ w : ||S_t||_{V_t^{-1}}^2 > \sigma_0^2 \log \left( \frac{\det(V_t)}{\delta^2 \det(\Lambda)} \right) \right\}.$$

And we are interested in bounding the probability that $\cup_{t>m} B_t(\delta)$ happens. Define $\tau(w) = \min\{t > m : w \in B_t(\delta)\}$. Then $\tau$ is a stopping time and it holds that,

$$\cup_{t>m} B_t(\delta) = \{w : \tau(w) < \infty\}.$$

Then we have that

$$P \left[ \cup_{t>m} B_t(\delta) \right] = P[m < \tau < \infty] = P \left[ ||S_\tau||_{V_\tau^{-1}}^2 > \sigma_0^2 \log \left( \frac{\det(V_\tau)}{\delta^2 \det(\Lambda)} \right), \tau > m \right] \leq \delta.$$

This concludes our proof of Lemma C.3.

## C.3  Proposition C.10 with its proof

We denote the optimal action $x^* = \arg\max_{x\in\mathcal{X}_0}\mu(x^\top\theta^*)$.

**Proposition C.10.** *For all $\delta \in (0,1)$, with probability $1 - \delta$, it holds that*

$$\mu(x^{*\top}\theta^*) - \mu(x_t^\top\theta^*) \le 2\beta_{t+T_1}^{x_t}\left(\frac{\delta}{2}\right),$$

*simultaneously for all $t \in \{2, 3, \ldots, T_2\}$.*

*Proof.* According to Corollary C.6, outside of the event of measure can be bounded by $\delta/2$:

$$\mu(x_t^\top\hat\theta_t) - \mu(x_t^\top\theta^*) \le \beta_{t+T_1}^{x_t}\left(\frac{\delta}{2}\right) \quad \text{for all } t \in \{2, 3, \ldots, T_2\}.$$

Similarly, with probability at least $1 - \delta/2$ it holds that

$$\mu(x^{*\top}\theta^*) - \mu(x^{*\top}\hat\theta_t) \le \beta_{t+T_1}^{x^*}\left(\frac{\delta}{2}\right) \quad \text{for all } t \in \{2, 3, \ldots, T_2\}.$$

Besides, by the choice of $x_t$ in Algorithm 2

$$\mu(x^{*\top}\hat\theta_t) - \mu(x_t^\top\hat\theta_t) = \mu(x^{*\top}\hat\theta_t) + \beta_{t+T_1}^{x^*}\left(\frac{\delta}{2}\right) - \mu(x_t^\top\hat\theta_t) - \beta_{t+T_1}^{x^*}\left(\frac{\delta}{2}\right)$$

$$\le \mu(x_t^\top\hat\theta_t) + \beta_{t+T_1}^{x_t}\left(\frac{\delta}{2}\right) - \mu(x_t^\top\hat\theta_t) - \beta_{t+T_1}^{x^*}\left(\frac{\delta}{2}\right)$$

$$= \beta_{t+T_1}^{x_t}\left(\frac{\delta}{2}\right) - \beta_{t+T_1}^{x^*}\left(\frac{\delta}{2}\right).$$

By combining the former inequalities we finish our proof. $\qquad\square$

## C.4  Proof of Theorem C.1

*Proof.* Based on Proposition C.10 we have

$$\mu(x^{*\top}\theta^*) - \mu(x_t^\top\theta^*) \le 2\beta_{t+T_1}^{x_t}\left(\frac{\delta}{2}\right) = 2\alpha_{t+T_1}\left(\frac{\delta}{2}\right)\|x_t\|_{M_t^{-1}(c_\mu)} \le 2\alpha_T\left(\frac{\delta}{2}\right)\|x_t\|_{M_t^{-1}(c_\mu)}.$$

Since we know that $\mu(x^{*\top}\theta^*) - \mu(x^\top\theta^*) \le k_\mu(x^{*\top}\theta^* - x^\top\theta^*) \le 2k_\mu S_0^2$ for all possible action $x$, and we can safely expect that $\alpha_{T_2}(\delta/2) > k_\mu S_0^2$ (at least by choosing $\sigma_0 = k_\mu \max\{S_0^2, 1\}$), then the regret of Algorithm 2 can be bounded as

$$Regret_{T_2} \le 2k_\mu S_0^2 + \sum_{t=2}^{T_2}\min\{\mu(x^{*\top}\theta^*) - \mu(x_t^\top\theta^*), 2k_\mu S_0^2\}$$

$$\le 2k_\mu S_0^2 + 2\alpha_T\left(\frac{\delta}{2}\right)\sum_{t=2}^{T_2}\min\{\|x_t\|_{M_t^{-1}(c_\mu)}, 1\}$$

$$\overset{(i)}{\le} 2k_\mu S_0^2 + 2\alpha_T\left(\frac{\delta}{2}\right)\sqrt{T_2}\sqrt{\sum_{t=2}^{T_2}\min\{\|x_t\|_{M_t^{-1}(c_\mu)}^2, 1\}}.$$

where the ineuqlity (i) comes from Cauchy-Schwarz inequality. And a commonly-used fact (e.g. [1], Lemma 11) yields that

$$\sum_{i=2}^{t}\min\{\|x_i\|_{M_i^{-1}(c_\mu)}^2, 1\} \le 2\log\left(\frac{|M_{t+1}(c_\mu)|}{|M_2(c_\mu)|}\right) \le 2\log\left(\frac{|M_{t+1}(c_\mu)|}{|\frac{\Lambda}{c_\mu}|}\right)$$

$$\le k\log\left(1 + \frac{c_\mu S_0^2}{k\lambda_0}(t + T_1)\right) + \frac{c_\mu S_0^2}{\lambda_\perp}(t + T_1).$$

Finally, by using the argument in Eqn. (20) and then plugging in the chosen value for $\lambda_\perp = \dfrac{c_\mu S_0^2 T}{k \log(1 + \frac{c_\mu S_0^2 T}{k \lambda_0})}$, we have

$$Regret_{T_2} \leq 2k_\mu S_0^2 +$$

$$\frac{2k_\mu}{c_\mu} \left( \sigma_0 \sqrt{2k \log\left(1 + \frac{c_\mu S_0^2}{k\lambda_0} T\right) - 2\log\left(\frac{\delta}{2}\right)} + \sqrt{c_\mu}\left(\sqrt{\lambda_0}S_0 + \sqrt{\frac{c_\mu S_0^2 T}{k\log\left(1 + \frac{c_\mu S_0^2}{k\lambda_0}T\right)}} S_\perp \right)\right)$$

$$\times \sqrt{T_2}\sqrt{4k\log\left(1 + \frac{c_\mu S_0^2}{k\lambda_0}T\right)},$$

which gives us the final bound in Theorem C.1. $\qquad\square$

# D  Consistency of $\hat{\theta}_t^{\mathbf{new}}$ in Algorithm 2

W.l.o.g. we assume that $\{\theta : \|\theta - \theta^*\|_2 \leq 1\} \subseteq \Theta^*$, or otherwise we can modify the contraint of $c_\mu$ in Assumption 3.5 as $c_\mu := \inf_{\{x \in \mathcal{X}_0, \|\theta - \theta^*\|_2 \leq 1\}} \mu'(x^\top \theta) > 0$. And we also assume that $\|x\|_2 \leq 1$ for $x \in \mathcal{X}_0$.

Adapted from the proof of Theorem 1 in [23], define $G(\theta) = g(\theta) - g(\theta^*) = \sum_{i=1}^{T_1}(\mu(x_{s_1,i}^\top \theta) - \mu(s_1, i^\top \theta^*))x_{s_1,i} + \sum_{i=1}^{n}(\mu(x_i^\top \theta) - \mu(x_i^\top \theta^*))x_i + \Lambda(\theta - \theta^*)$. W.l.o.g we suppose $c_\mu \leq 1$ based on argument in Appendix G. Then it holds that for any $\theta_1, \theta_2 \in \mathbb{R}^p$

$$G(\theta_1) - G(\theta_2) =$$
$$\left[ \sum_{i=1}^{T_1}(\mu'(x_{s_1,i}^\top \theta) - \mu(s_1, i^\top \theta^*))x_{s_1,i}x_{s_1,i}^\top + \sum_{i=1}^{n}(\mu'(x_i^\top \theta) - \mu(x_i^\top \theta^*))x_i x_i^\top + \Lambda \right](\theta_1 - \theta_2).$$

By denoting $V = \sum_{i=1}^{T_1} x_{s_1,i}x_{s_1,i}^\top + \sum_{i=1}^{n} x_i x_i^\top + \Lambda$. We have

$$(\theta_1 - \theta_2)^\top(G(\theta_1) - G(\theta_2)) \geq (\theta_1 - \theta_2)^\top(c_\mu V)(\theta_1 - \theta_2) > 0$$

Therefore, the rest of proof would be identical to that of Step 1 in the proof of Theorem 1 in [23]. Based on the step 1 in the proof of Theorem 1 in [23], we have

$$\|G(\theta)\|_{V^{-1}}^2 \geq c_\mu^2 \lambda_{\min}(V) \|\theta - \theta^*\|_2^2.$$

as long as $\|\theta - \theta^*\|_2 \leq 1$. Then Lemma A of [6] and Lemma 7 of [23] suggest that we have

$$\left\|\hat{\theta} - \theta^*\right\| \leq \frac{4\sigma}{c_\mu}\sqrt{\frac{p + \log(1/\delta)}{\sigma^2}} \leq 1,$$

when $\lambda_{\min}(V) \geq 16\sigma^2[p + \log(1/\delta)]/c_\mu^2$ for any $\delta > 0$. Therefore, it suffices to show that the condition $\lambda_1 \geq 16\sigma^2[p + \log(1/\delta)]/c_\mu^2$ for any $\delta > 0$ holds with high probability (e.g. $1 - \delta$), and we utilize the Proposition 1 of [23], which is given as follows:

**Proposition** (Proposition 1 of [23]): *Define $V_n = \sum_{t=1}^{n} x_t x_t^\top (+\Lambda)$ where $x_i$ is drawn iid from some distribution $\nu$ with suppost in the unit ball, $\mathbb{B}^d$. Furthermore, let $\Sigma = \mathbb{E}(x_t x_t^\top)$ be the second moment matrix, and $B$ and $\delta$ be two positive constants. Then, there exists positive universal constants $C_1$ and $C_2$ such that $\lambda_{\min}(V_n) \geq B$ with probability at least $1 - \delta$, as long as*

$$n \geq \left(\frac{C_1\sqrt{d} + C_2\sqrt{\log(1/\delta)}}{\lambda_{\min}(\Sigma)}\right)^2 + \frac{2B}{\lambda_{\min}(\Sigma)}$$

Therefore, we can dedeuce that $\left\|\hat{\theta}_t - \theta^*\right\|_2 \leq 1$ holds with probability at least $1 - \delta$ as long as $T_1 \geq ((\hat{C}_1\sqrt{p} + \hat{C}_2\sqrt{\log(1/\delta)})/\lambda_1)^2 + 2B/\lambda_1$ holds for some absolute constants $\hat{C}_1, \hat{C}_2$ with the

definition $B := 16\sigma^2(p + \log(1/\delta))/c_\mu^2$. Notice that this condition could easily hold if $\lambda_1 \asymp \sigma^2$ is not diminutive in magnitude. Otherwise, we believe a tighter bound exists in that case, and we will leave it as a future work.

We also present an intuitive explanation for this consistency result: [23] proved the consistency of the MLE $\hat{\theta}_t$ without the regularizer. Regarding the penalty $\theta^\top \Lambda \theta$, for the first $k$ entries of $\hat{\theta}_t$ the penalized parameter $\lambda_0$ is small, and hence it will have mild effect after sufficient warm-up rounds $T_1$. For the remaining $(p - k)$ elements suffering large penalty, the estimated $\hat{\theta}_{t,k+1:p}$ would be ultra small in magnitude as desired since we argue that after the transformation $\theta^*_{k+1:p}$ will also be insignificant. This implies that $\left\|\hat{\theta}_{t,k+1:p} - \theta^*_{k+1:p}\right\|_2$ is well contronlled. As a result, the estimated $\hat{\theta}_t$ tends to be consistent.

# E    Analysis of Theorem 4.2

## E.1    Proof of Theorem 4.2

*Proof.* Let us define $r_t = \max_{X \in \mathcal{X}} \mu(\langle X, \Theta^* \rangle) - \mu(\langle X_t, \Theta^* \rangle)$, the instantaneous regret at time $t$. We can easily bound the regret for stage 1 as $\sum_{t=1}^{T_1} r_t \leq 2S_f T_1$. For the second stage, we have a bound according to Theorem C.1 (Theorem H.1):

$$\sum_{t=T_1+1}^{T} r_t \leq \widetilde{O}(k\sqrt{T} + \sqrt{\lambda_0 kT} + TS_\perp) \leq \widetilde{O}\left(k\sqrt{T} + \sqrt{\lambda_0 kT} + T\frac{d_1 d_2 r}{T_1 D_{rr}^2} \log\left(\frac{d_1 + d_2}{\delta}\right)\right).$$

Therefore, the overall regret is:

$$\sum_{t=1}^{T} r_t \leq \widetilde{O}\left(2S_f T_1 + k\sqrt{T} + \sqrt{\lambda_0 kT} + T\frac{d_1 d_2 r}{T_1 D_{rr}^2} \log\left(\frac{d_1 + d_2}{\delta}\right)\right).$$

After plugging the choice of $T_1$ given in Theorem 4.2, it holds that

$$\sum_{t=1}^{T} r_t \leq \widetilde{O}\left((\frac{\sqrt{rd_1 d_2}}{D_{rr}} + \sqrt{\lambda_0 k} + k)\sqrt{T}\right) \lesssim \widetilde{O}\left((\frac{\sqrt{rd_1 d_2}}{D_{rr}} + k)\sqrt{T}\right) = \widetilde{O}\left((d_1 + d_2)r\sqrt{T}\right).$$

$\square$

# F    Details of Theorem 4.3

## F.1    Proof of Theorem 4.3

*Proof.* Here we will overload the notation a little bit. Under the new arm feature set and parameter set after rotation, let $X^*$ be the best arm and $X_t$ be the arm we pull at round $t$ for stage 2. And we denote $x_{t,sub}$ be the vectorization of $X_t$ after removing the last $p - k$ covariates, and similarly define $x^*_{sub}$ and $\theta^*_{sub}$ as the subtracted version of $\text{vec}(X^*)$ and $\text{vec}(\Theta^*)$ respectively. We use $r_t = \mu(\langle X^*, \Theta^* \rangle) - \mu(\langle X_t, \Theta^* \rangle)$ as the instantaneous regret at round t for stage 2. Then it holds that, for $t \in [T_2]$

$$r_t = \mu(\langle X^*, \Theta^* \rangle) - \mu(x^{*\top}_{sub}\theta^*_{sub}) + \mu(x^{*\top}_{sub}\theta^*_{sub}) - \mu(x^\top_{t,sub}\theta^*_{sub}) + \mu(x^\top_{t,sub}\theta^*_{sub}) - \mu(\langle X_t, \Theta^* \rangle)$$

$$\leq k_\mu |\langle X^*, \Theta^* \rangle - x^{*\top}_{sub}\theta^*_{sub}| + k_\mu |\langle X_t, \Theta^* \rangle - x^\top_{t,sub}\theta^*_{sub}| + \mu(x^{*\top}_{sub}\theta^*_{sub}) - \mu(x^\top_{t,sub}\theta^*_{sub})$$

$$\leq k_\mu (\left\|\widehat{U}^\top_\perp X^* \widehat{V}_\perp\right\|_F + \left\|\widehat{U}^\top_\perp X_t \widehat{V}_\perp\right\|_F) \left\|\widehat{U}^\top_\perp UDV^\top \widehat{V}_\perp\right\|_F + \mu(x^{*\top}_{sub}\theta^*_{sub}) - \mu(x^\top_{t,sub}\theta^*_{sub})$$

$$\leq 2k_\mu S_0 \frac{d_1 d_2 r}{T_1 D_{rr}^2} \log\left(\frac{d_1 + d_2}{\delta}\right) + \mu(x^{*\top}_{sub}\theta^*_{sub}) - \mu(x^\top_{t,sub}\theta^*_{sub}).$$

Therefore, the overall regret can be bounded as

$$2S_f T_1 + \sum_{t=1}^{T_2} r_t \leq 2S_f T_1 + 2k_\mu S_0 \frac{d_1 d_2 r}{D_{rr}^2 T_1} T_2 + \sum_{t=1}^{T_2} \mu(x^{*\top}_{sub}\theta^*_{sub}) - \mu(x^\top_{t,sub}\theta^*_{sub}).$$

Since efficient low dimensional generalized linear bandit algorithm can achieve regret $\widetilde{O}(d\sqrt{T})$ where $d$ is the dimension of parameter and $T$ is the time horizon when no sparsity (low-rank structure) presents in the model. After plugging our carefully chosen $T_1$, the regret is

$$2S_f T_1 + 2k_\mu S_0 \frac{d_1 d_2 r}{T_1 D_{rr}^2} \log\left(\frac{d_1 + d_2}{\delta}\right) T_2 + \widetilde{O}(k\sqrt{T_2}) = \widetilde{O}\left((\frac{\sqrt{r d_1 d_2}}{D_{rr}} + k)\sqrt{T}\right)$$

$$= \widetilde{O}\left(dr\sqrt{T}\right). \qquad (22)$$

And here we use the condition that $D_{rr} = \Theta(1/\sqrt{r})$ since $\Theta^*$ has a low rank structure with rank$(\Theta^*) = r$. $\qquad\square$

### F.2   Modified Theorem 4.3 with finite action sets

Our ESTS framework could be used and attain a better regret bound when the action set is finitely large, and we only need to choose appropriate generalized linear bandit algorithms in stage 2 accordingly. When the action set is finite, we know that some state-of-the-art generalized linear bandit algorithms (e.g. SupCB-GLM [23]) could achieve regret bound as $O(\sqrt{dT})$ with $d$-dimensional contextual information over $T$ rounds, and it is also the minimax lower bound for this problem [21]. By using the same values for parameters in Theorem 4.3, we can show that the overall regret is after modifying Eqn. (22):

$$2S_f T_1 + 2k_\mu S_0 \frac{d_1 d_2 r}{T_1 D_{rr}^2} \log\left(\frac{d_1 + d_2}{\delta}\right) T_2 + \widetilde{O}(\sqrt{kT_2}) = \widetilde{O}\left((\frac{\sqrt{r d_1 d_2}}{D_{rr}} + \sqrt{k})\sqrt{T}\right)$$

$$= \widetilde{O}\left(\frac{\sqrt{d_1 d_2 r T}}{D_{rr}}\right).$$

Note here we get an improved regret bound when the action set is finite, and this regret bound is better than the optimal bound $\widetilde{O}((\sqrt{r d_1 d_2}/D_{rr} + k)\sqrt{T})$ for generalized low-rank matrix bandit with arbitrary arm set by omitting the term $\widetilde{O}(k\sqrt{T})$.

## G   Explanation of $V_t$ replacing $M_t(c_\mu)$

Technically we can always assume $c_\mu \in (0, 1]$ since we can always choose $c_\mu = 1$ when it can take values greater than 1. And when $c_\mu \le 1$ it holds that,

$$M_t(c_\mu) = \sum_{i=1}^{t-1} x_i x_i^\top + \frac{\Lambda}{c_\mu} \succeq \sum_{i=1}^{t-1} x_i x_i^\top + \Lambda = V_t.$$

Therefore, we can easily keep the exactly identical outline of our proof of the bound of regret for Algorithm 2 after replacing $M_t(c_\mu)$ by $V_t$ everywhere, and the result only change by a constant factor of $1/\sqrt{c_\mu}$, which would not be too large in most cases. However, in our algorithm and proof we still use $M_t(c_\mu)$ for a better theoretical bound.

## H   Additional Algorithms

### H.1   PLowGLM-UCB

We could modify Algorithm 2 by only recomputing $\hat\theta_t$ and whenever $|M_t(c_\mu)|$ increases by a constant factor $C > 1$ in scale, and consequently we only need to solve the Eqn. (10) for $O(\log(T_2))$ times up to the horizon $T_2$, which significantly alleviate the computational complexity. The pseudo-code of PLowGLM-UCB is given in Algorithm 4.

Theorem H.1 shows the regret bound of PLowGLM-UCB under Assumption 3.4 and 3.5.

**Theorem H.1.** (Regret of PLowGLM-UCB) *For any fixed failure rate $\delta \in (0, 1)$, if we run the PLowGLM-UCB algorithm with $\rho_t(\delta) = \alpha_{t+T_1}(\delta/2)$ and*

$$\lambda_\perp \asymp \frac{c_\mu S_0^2 T}{k \log(1 + \frac{c_\mu S_0^2 T}{k\lambda_0})}.$$

---

**Algorithm 4** PLowGLM-UCB

---

**Input:** $T_2, k, \mathcal{X}_0$, the probability rate $\delta$, penalization parameters $(\lambda_0, \lambda_\perp)$, the constant $C$.

1: Initialize $M_1(c_\mu) = \sum_{i=1}^{T_1} x_{s_1,i} x_{s_1,i}^\top + \Lambda/c_\mu$.
2: **for** $t \geq 1$ **do**
3:     **if** $|M_t(c_\mu)| > C|M_\tau(c_\mu)|$ **then**
4:         Estimate $\hat{\theta}_t$ according to (10).
5:         $\tau = t$
6:     **end if**
7:     Choose arm $x_t = \arg\max_{x \in \mathcal{X}_0} \{\mu(x^\top \hat{\theta}_\tau) + \rho_\tau(\delta)\|x\|_{M_t^{-1}(c_\mu)}\}$, receive $y_t$.
8:     Update $M_{t+1}(c_\mu) \longleftarrow M_t(c_\mu) + x_t x_t^\top$.
9: **end for**

---

*Then the regret of PLowGLM-UCB* $(Regret_{T_2})$ *satisfies, with probability at least* $1 - \delta$

$$\widetilde{O}(k\sqrt{T_2} + \sqrt{\lambda_0 k T} + TS_\perp) \cdot \sqrt{C} = \widetilde{O}(k\sqrt{T} + TS_\perp) \cdot \sqrt{C}.$$

Similarly, for PLowUCB-GLM we can also prove that the regret bound increase at most by a constant multiplier $\sqrt{C}$ by using the same lemma and argument we show in the following Section H.2. And we can get the bound of regret for PLowGLM-UCB in problem dependence case, and the bound will be exactly the same as that we have shown in Theorem C.1 except a constant multiplier $\sqrt{C}$.

## H.2 Proof of Theorem H.1

We use similar sketch of proof for Theorem 5 in [1]. First, we show the following lemma:

**Lemma H.2.** *([1], Lemma 12) Let $A$ and $B$ be two positive semi-definite matrices such that $A \preceq B$. Then, we have that*

$$\sup_{x \neq 0} \frac{x^\top A x}{x^\top B x} \leq \frac{|A|}{|B|}.$$

Then we can outline the proof of Theorem H.1 as follows.

*Proof.* Let $\tau_t$ be the value of $\tau$ at step $t$ in Algorithm H.1. By an argument similar to the one used in proof of Theorem C.1, we deduce that for any $x \in \mathbb{R}$ and all $t \geq 2$ simultaneously,

$$
\begin{aligned}
|\mu(x^\top \theta^*) - \mu(x^\top \hat{\theta}_{\tau_t})| &\leq \frac{k_\mu}{c_\mu} \left\| g_{\tau_t}(\theta^*) - g_{\tau_t}(\hat{\theta}_{\tau_t}) \right\|_{M_{\tau_t}^{-1}(c_\mu)} \|x\|_{M_{\tau_t}^{-1}(c_\mu)} \\
&= \frac{k_\mu}{c_\mu} \left\| g_{\tau_t}(\theta^*) - g_{\tau_t}(\hat{\theta}_{\tau_t}) \right\|_{M_{\tau_t}^{-1}(c_\mu)} \left\| M_{\tau_t}^{-\frac{1}{2}}(c_\mu) x \right\|_2 \\
&\leq \frac{k_\mu}{c_\mu} \left\| g_{\tau_t}(\theta^*) - g_{\tau_t}(\hat{\theta}_{\tau_t}) \right\|_{M_{\tau_t}^{-1}(c_\mu)} \left\| M_t^{-\frac{1}{2}}(c_\mu) x \right\|_2 \sqrt{\frac{|M_{\tau_t}^{-1}(c_\mu)|}{|M_t^{-1}(c_\mu)|}} \\
&\leq \frac{k_\mu}{c_\mu} \sqrt{C} \left\| g_{\tau_t}(\theta^*) - g_{\tau_t}(\hat{\theta}_{\tau_t}) \right\|_{M_{\tau_t}^{-1}(c_\mu)} \|x\|_{M_t^{-1}(c_\mu)} \leq \sqrt{C} \beta_{t+T_1}^x(\delta).
\end{aligned}
$$

where the last inequality comes from the proof of Proposition C.2 similarly. The rest of the proof will be mostly identical to that of Theorem C.1 and hence we would copy it here for completeness:

Based on Proposition C.10 we have

$$
\begin{aligned}
\mu(x^{*\top}\theta^*) - \mu(x_t^\top \theta^*) &\leq 2\sqrt{C}\beta_{t+T_1}^{x_t}\left(\frac{\delta}{2}\right) = 2\sqrt{C}\alpha_{t+T_1}\left(\frac{\delta}{2}\right)\|x_t\|_{M_t^{-1}(c_\mu)} \\
&\leq 2\sqrt{C}\alpha_T\left(\frac{\delta}{2}\right)\|x_t\|_{M_t^{-1}(c_\mu)}.
\end{aligned}
$$

---

**Algorithm 5** Generalized Explore Subspace Then Transform (G-ESTT)

---

**Input:** Action set $\{\mathcal{X}_t\}$, $T, T_1, \mathcal{D}$, the probability rate $\delta$, parameters for stage 2: $\lambda, \lambda_\perp$.

**Stage 1: Subspace Estimation**

1: Randomly choose $X_t \in \mathcal{X}$ according to $\mathcal{D}$ and record $X_t, Y_t$ for $t = 1, \ldots T_1$.

2: Obtain $\widehat{\Theta}$ by solving the following equation:

$$\widehat{\Theta} = \arg\min_{\Theta \in R^{d_1 \times d_2}} \frac{1}{T_1} \sum_{i=1}^{T_1} \{b(\langle X_i, \Theta \rangle) - y_i \langle X_i, \Theta \rangle\} + \lambda_{T_1} \|\Theta\|_{\text{nuc}}.$$

3: Obtain the full SVD of $\widehat{\Theta} = [\widehat{U}, \widehat{U}_\perp] \widehat{D} [\widehat{V}, \widehat{V}_\perp]^\top$ where $\widehat{U}$ and $\widehat{V}$ contains the first r left-singular vectors and the first r right-singular vectors respectively.

**Stage 2: Almost Low Rank Generalized Linear Bandit**

4: Rotate the admissible parameter space: $\Theta' := [\widehat{U}, \widehat{U}_\perp]^\top \Theta [\widehat{V}, \widehat{V}_\perp]$, and transform the parameter set as:

$$\Theta_0 := \{\text{vec}(\Theta'_{1:r,1:r}), \text{vec}(\Theta'_{r+1:d_1,1:r}), \text{vec}(\Theta'_{1:r,r+1:d_2}), \text{vec}(\Theta'_{r+1:d_1,r+1:d_2})\}.$$

5: **for** $t \geq T - T_1$ **do**

6:     Rotate the arm feature set: $\mathcal{X}'_t := [\widehat{U}, \widehat{U}_\perp]^\top \mathcal{X}_t [\widehat{V}, \widehat{V}_\perp]$.

7:     Define the vectorized arm set so that the last $(d_1 - r) \cdot (d_2 - r)$ components are almost negligible:

$$\mathcal{X}_{0,t} := \{\text{vec}(\mathcal{X}'_{\{1:r,1:r\},t}), \text{vec}(\mathcal{X}'_{\{r+1:d_1,1:r\},t}), \text{vec}(\mathcal{X}'_{\{1:r,r+1:d_2\},t}), \text{vec}(\mathcal{X}'_{\{r+1:d_1,r+1:d_2\},t})\}.$$

8:     Invoke LowGLM-UCB (PLowGLM-UCB or LowUCB-GLM) with the arm set $\mathcal{X}_{0,t}$, the parameter space $\Theta_0$, the low dimension $k = (d_1 + d_2)r - r^2$ and penalization parameter $(\lambda_0, \lambda_\perp)$ for one round. Update the matrix $M_t(c_\mu)$ or $V_t$ accordingly.

9: **end for**

---

Since we have that $\alpha_{T_2}(\delta/2) > k_\mu S_0^2$, the Regret of Algorithm 4 can be bounded as

$$Regret_{T_2} \leq 2k_\mu S_0^2 + \sum_{t=2}^{T_2} \min\{\mu(x^{*\top}\theta^*) - \mu(x_t^\top \theta^*), 2k_\mu S_0^2\}$$

$$\leq 2k_\mu S_0^2 + 2\sqrt{C}\alpha_T\left(\frac{\delta}{2}\right) \sum_{t=2}^{T_2} \min\{\|x_t\|_{M_t^{-1}(c_\mu)}, 1\}$$

$$\leq 2k_\mu S_0^2 + 2\sqrt{C}\alpha_T\left(\frac{\delta}{2}\right) \sqrt{T_2}\sqrt{\sum_{t=2}^{T_2} \min\{\|x_t\|_{M_t^{-1}(c_\mu)}^2, 1\}}.$$

where the last ineuqlity comes from Cauchy-Schwarz inequality. Finally, by a self-normalized martingale inequality ( [1], Lemma 11) and and then plugging in the chosen value for $\lambda_\perp = \dfrac{c_\mu S_0^2 T}{k \log(1 + \frac{c_\mu S_0^2 T}{k\lambda_0})}$, we have

$$Regret_{T_2} \leq 2k_\mu S_0^2 + \frac{2k_\mu}{c_\mu}\sqrt{C}$$

$$\times \left(\sigma_0\sqrt{2k \log\left(1 + \frac{c_\mu S_0^2}{k\lambda_0}T\right) - 2\log\left(\frac{\delta}{2}\right)} + \sqrt{c_\mu}\left(\sqrt{\lambda_0}S_0 + \sqrt{\frac{c_\mu S_0^2 T}{k \log\left(1 + \frac{c_\mu S_0^2}{k\lambda_0}T\right)}}S_\perp\right)\right)$$

$$\times \sqrt{T_2}\sqrt{4k \log\left(1 + \frac{c_\mu S_0^2}{k\lambda_0}T\right)},$$

which gives us the final bound in Theorem H.1.     $\square$

**Algorithm 6** Generalized Explore Subspace Then Subtract (G-ESTS)

**Input:** Action set $\{\mathcal{X}_t\}, T, T_1, \mathcal{D}$, the probability rate $\delta$, parameters for stage 2: $\lambda, \lambda_\perp$.

**Stage 1: Subspace Estimation**

1: **for** $t = 1$ **to** $T_1$ **do**
2:     Pull arm $X_t \in \mathcal{X}$ according to the distribution $\mathcal{D}$, observe payoff $y_t$.
3: **end for**
4: Obtain $\widehat{\Theta}$ by solving the following equation:

$$\widehat{\Theta} = \arg\min_{\Theta \in R^{d_1 \times d_2}} \frac{1}{T_1} \sum_{i=1}^{T_1} \{b(\langle X_i, \Theta \rangle) - y_i \langle X_i, \Theta \rangle\} + \lambda_{T_1} \|\Theta\|_{\text{nuc}}.$$

5: Obtain the full SVD of $\widehat{\Theta} = [\widehat{U}, \widehat{U}_\perp] \, \widehat{D} \, [\widehat{V}, \widehat{V}_\perp]^\top$ where $\widehat{U}$ and $\widehat{V}$ contains the first r left-singular vectors and the first r right-singular vectors respectively.

**Stage** 2: Low Rank Generalized Linear Bandit

6: Rotate the admissible parameter space: $\Theta' := [\widehat{U}, \widehat{U}_\perp]^\top \Theta [\widehat{V}, \widehat{V}_\perp]$, and transform the parameter set as:

$$\Theta_0 := \{\text{vec}(\Theta'_{1:r,1:r}), \text{vec}(\Theta'_{r+1:d_1,1:r}), \text{vec}(\Theta'_{1:r,r+1:d_2}), \text{vec}(\Theta'_{r+1:d_1,r+1:d_2})\}.$$

7: **for** $t \geq T - T_1$ **do**
8:     Rotate the arm feature set: $\mathcal{X}'_t := [\widehat{U}, \widehat{U}_\perp]^\top \mathcal{X}_t [\widehat{V}, \widehat{V}_\perp]$.
9:     Define the vectorized arm set so that the last $(d_1 - r) \cdot (d_2 - r)$ components are almost negligible, and then drop the last $(d_1 - r) \cdot (d_2 - r)$ components:

$$\mathcal{X}_{0,sub,t} := \{\text{vec}(\mathcal{X}'_{\{1:r,1:r\},t}), \text{vec}(\mathcal{X}'_{\{r+1:d_1,1:r\},t}), \text{vec}(\mathcal{X}'_{\{1:r,r+1:d_2\},t})\}.$$

10:     Invoke any modern generalized linear (contextual) bandit algorithm with the arm set $\mathcal{X}_{0,sub,t}$, the parameter space $\Theta_{0,sub}$, and the low dimension $k = (d_1 + d_2)r - r^2$ for one round.
11: **end for**

## H.3 Algorithms for the Contextual Setting

To show algorithm G-ESTT and G-ESTS for the contextual setting, where the arm set $\mathcal{X}_t = \{X_{i,t}\}$ may vary over time $t = [T]$, we would firstly update some notations besides the ones we have defined in Section 4.2. We denote the time-dependent action set $\mathcal{X}_t$ after rotation as:

$$\mathcal{X}'_t = [\widehat{U}, \widehat{U}_\perp]^\top \mathcal{X} [\widehat{V}, \widehat{V}_\perp],$$

And we modify the notations of the vectorized arm set for G-ESTT and G-ESTS defined in Eqn. (3), (13) accordingly for each iteration:

$$\mathcal{X}_{0,t} := \{\text{vec}(\mathcal{X}'_{\{1:r,1:r\},t}), \text{vec}(\mathcal{X}'_{\{r+1:d_1,1:r\},t}), \text{vec}(\mathcal{X}'_{\{1:r,r+1:d_2\},t}), \text{vec}(\mathcal{X}'_{\{r+1:d_1,r+1:d_2\},t})\},$$
$$\mathcal{X}_{0,sub,t} := \{\text{vec}(\mathcal{X}'_{\{1:r,1:r\},t}), \text{vec}(\mathcal{X}'_{\{r+1:d_1,1:r\},t}), \text{vec}(\mathcal{X}'_{\{1:r,r+1:d_2\},t})\}.$$

Details can be found in Algorithm 5 and 6.

# I Additional Experimental Details

## I.1 Parameter Setup for Simulations

Here we present our parameter setting for algorithms involved in our experiment in Section 5.

**Basic setup**: horizon $T = 45000$. For the case where $d_1 = d_2 = 12$ and $r = 2$ we extend the horizon until 75000 in figures to display the superiority of our proposed algorithms more clearly. The 480 (1000) random matrices are sampled uniformly from $d_1 d_2$-dimensional unit sphere.

**LowESTR**: (same setup as in [26])

- failure rate: $\delta = 0.01$, the standard deviation: $\sigma = 0.01$ and the steps of stage 1: $T_1 = 1800$.

- penalization parameter in stage 1: $\lambda_{T_1} = 0.01\sqrt{\frac{1}{T_1}}$, and the gradient decent step size: 0.01.

- $B = 1, B_\perp = \frac{\sigma^2 (d_1 + d_2)^3 r}{T_1 D_{r,r}^2}, \lambda = 1, \lambda_\perp = \frac{T_2}{k \log(1 + T_2/\lambda)}$, grid search for $\sqrt{\beta_t}$ with multiplier in $\{0.2, 1, 5\}$.

**SGD-TS**: (details in [9])

- grid search for exploration rates in $\{0.1, 1, 10\}$.
- grid search for $C$ in $\{1, 3, 5, 7\}$.
- grid search for initial step sizes in $\{0.01, 0.1, 1, 5, 10\}$.

**G-ESTT**: (LowGLM-UCB in Stage 2)

- failure rate: $\delta = 0.01$, and the steps of stage 1: $T_1 = 1800$.
- $S_0 = 1, \Theta = \{X \in \mathbb{R}^{d_1 \times d_2} : \|X\|_F \leq 1\}$ for the case $r = 1$, and $S_0 = 5, \Theta = \{X \in \mathbb{R}^{d_1 \times d_2} : \|X\|_F \leq 5\}$ for the case $r = 2$.
- penalization in solving Eqn. (6) with $\lambda_{T_1}$ suggested in Theorem 4.1. (We believe that a simple grid search near this value would be better.)
- $p_{ij}$ set to be centered normal distribution with standard deviation $1/d$ in Stage 1. Specifically, at each round we randomly select a matrix $X_{rand,t}$ based on this $\{p_{ij}\}$ elementwisely, and then pull the arm that is closest to $X_{rand,t}$ w.r.t. $\|\cdot\|_F$ among all candidates in the arm set.
- proximal gradient descent with backtracking line search solving Eqn. (6), step size set to $0.1$.
- $\lambda_0 = 1, \lambda_\perp = \frac{c_\mu^2 S_0^2 T_2}{k \log\left(1 + \frac{c_\mu S_0^2 T_2}{k \lambda_0}\right)}, S_\perp = \frac{d_1 d_2 r}{T_1 D_{rr}^2} \log\left(\frac{d_1 + d_2}{\delta}\right)$, grid search for exploration bonus with multiplier in $\{0.2, 1, 5\}$.

**G-ESTS**: (SGD-TS in Stage 2)

- The steps of stage 1: $T_1 = 1800$.
- penalization in solving Eqn. (6) with $\lambda_{T_1}$ suggested in Theorem 4.1. (We believe that a simple grid search near this value would be better.)
- $p_{ij}$ set to be centered normal distribution with standard deviation $1/d$ in Stage 1. Specifically, at each round we randomly select a matrix $X_{rand,t}$ based on this $\{p_{ij}\}$ elementwisely, and then pull the arm that is closest to $X_{rand,t}$ w.r.t. $\|\cdot\|_F$ among all candidates in the arm set.
- proximal gradient descent with backtracking line search solving Eqn. (6), step size set to $0.1$.
- use the same setup for SGD-TS as we have listed.

### I.2 Additonal experimental results

Here we display the regret curves of algorithms under four settings with 1000 arms in Figure 2, where our proposed G-ESTS and G-ESTT also dominate other methods regarding both accuracy and computation.

### I.3 Comparison between G-ESTT and G-ESTS

In this section we compare the performance of our two frameworks G-ESTT and G-ESTS, and it is obvious that both these two proposed methods work better than the existing LowESTR and state-of-the-art generalized linear bandit algorithms under our problem setting based on Figure 1 and 2. Notice that G-ESTT and G-ESTS perform similarly well under the scenario $r = 1$ (G-ESTS is slightly better). However, for the case $r = 2$, we find that G-ESTT achieve less cumulative regret than G-ESTS does. We believe it is because that, on the one hand, G-ESTS depends more on the precision of estimate $\widehat{\Theta}$, which becomes more challenging for the case $r = 2$. On the other hand, for G-ESTS how to reuse the random-selected actions in stage 1 is also tricky, and we will leave it as a future work. Therefore, G-ESTT (with LowUCB-GLM) quickly takes the lead in the very beginning of stage 2 since LowUCB-GLM can yield a consistent estimator early in stage 2 by reclaiming the randomly-chosen actions.

However, we find that G-ESTS is incredibly faster than other methods (including G-ESTT) as it only spends about one tenth of the running time of LowESTR until convergence as shown in Table 2. Notice that G-ESTT with LowUCB-GLM is a little bit slower since it utilizes more samples for estimation in each iteration for better performance. Moreover, we conduct another simulation for the case $r = 2, d_1 = d_2 = 12$ where we additionally choose $T_1 = 3200$, and the results are displayed

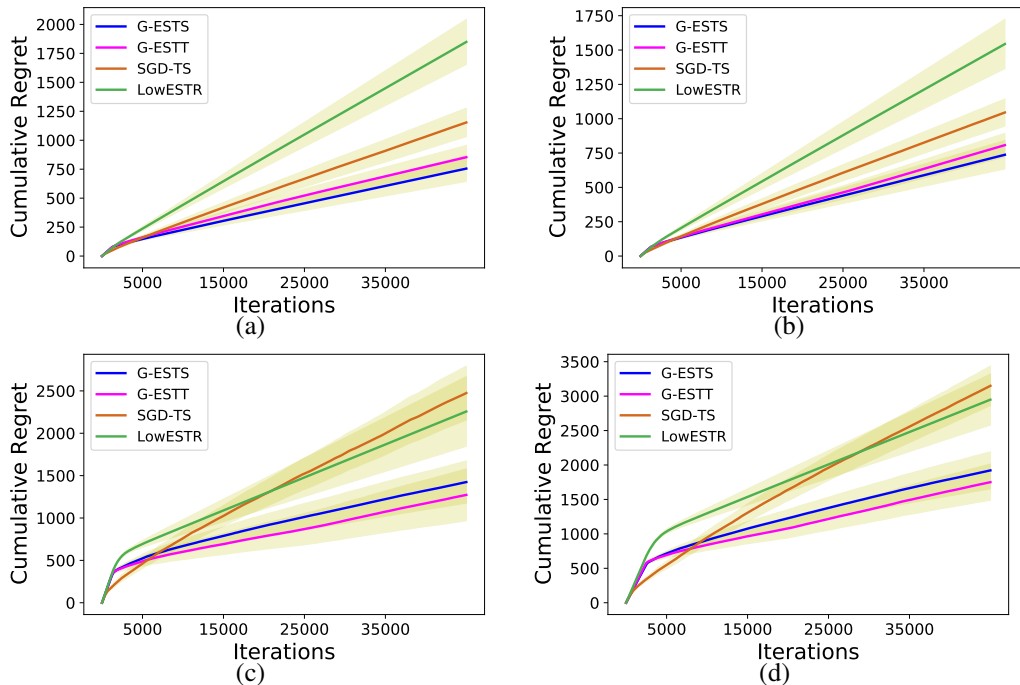

Figure 2: Plots of regret curves of algorithm G-ESES, SGD-TS and LowESTR under four settings (1000 arms). (a): diagonal $\Theta^*$ $d_1 = d_2 = 10, r = 1$; (b): diagonal $\Theta^*$ $d_1 = d_2 = 12, r = 1$; (c): non-diagonal $\Theta^*$ $d_1 = d_2 = 10, r = 2$; (d): non-diagonal $\Theta^*$ $d_1 = d_2 = 12, r = 2$.

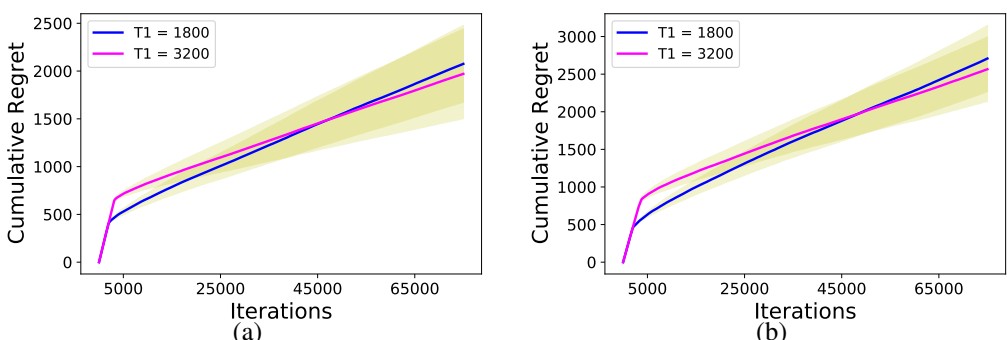

Figure 3: Plots of regret curves of algorithm G-ESES the scenario $d_1 = d_2 = 12, r = 2$ under $T_1 = 1800$ and $T_1 = 3200$ (a): fixed $480$ arms; (b): fixed $1000$ arms.

in Figure 3 after 100 times repeated simulations. We observe that by appropriately enlarging the length of stage 1 ($T_1$), G-ESTS would perform better in the long run as we expect, since a more accurate estimation of $\Theta^*$ could be obtained. Therefore, we can conclude our proposed G-ESTS could perform prominently with parsimonious computation by mildly tuning the length of stage 1 ($T_1$).

## I.4    Comparison with other matrix subspace detection methods

To pre-check the efficiency of our Stein's lemma-based method for subspace estimation, we also tried the nuclear-norm regularized log-likelihood maximization with its details introduced in the following Appendix I. Particularly, we could solve the regularized negative log-likelihood minimization problem with nuclear norm penalty as shown in Eqn. (23).

Specifically, we consider the two cases of our simulations: 480 arms, $d = 10$, $r = 1$ (Figure 1(a) case) and 480 arms, $d = 10$, $r = 2$ (Figure 1(c) case). We used the same setting as described in Appendix I above ($T_1 = 1800$, $T = 45000$), and implemented proximal gradient descent with the backtracking line

Table 3: Comparison between our proposed Stein's lemma-based method and the log-likelihood maximization method for low-rank matrix subspace estimation.

| Case | Low-rank detection method | Regret | Transformed error |
|------|---------------------------|--------|-------------------|
| Figure 1(a) | Stein's lemma-based method | G-ESTT:723.27, G-ESTS:510.80 | 0.086 |
| | Log-likelihood maximization | G-ESTT:724.96, G-ESTS:515.25 | 0.089 |
| Figure 1(c) | Stein's lemma-based method | G-ESTT:1088.26, G-ESTS:1106.71 | 0.542 |
| | Log-likelihood maximization | G-ESTT:1136.54, G-ESTS:1198.39 | 0.583 |

search for optimization. The average regret cumulative regret along with the average transformed error $\left\|\theta^*_{(k+1):p}\right\|_2$ defined in Eqn. (8) are reported in Table 3.

Therefore, we can see that our low-rank matrix detection method outperforms the regularized log-likelihood maximization method, especially when the underlying parameter matrix is complicated (Figure 1(c) case). This is also consistent with our theoretical analysis, as we will show in the following Appendix J that the theoretical bound of loss $\left\|\hat{\Theta} - \Theta^*\right\|_F^2$ is of order $d^3 r/T_1$ using the regularized log-likelihood maximization method, which is worse than the convergence rate of our proposed method in Theorem 4.1.

## J   Bonus: Matrix Estimation with Restricted Strong Convexity

### J.1   Methodology

As we have mentioned in our main paper, we can achieve a better matrix recovery rate regarding the Frobenius norm by using generalized first-order Stein's Lemma on Eqn. (6) other than using the restricted strong convexity [35]. Specifically, [26] provided a line of proof to show the convergence rate as

$$\left\|\hat{\Theta} - \Theta^*\right\|_F = O\left(\sqrt{\frac{(d_1 + d_2)^3 r}{T_1}}\right)$$

only in the linear reward framework during time horizon $T_1$. Therefore, it implies that we may not be not able to find a better bound than $O(\sqrt{d_1 + d_2)^3 r/T_1})$ in GLM since the linear model is a special case of GLM. This fact implies that our matrix estimation method is superior regarding the theoretical bound. However, for the completeness of our work, we also approach the matrix estimation problem by using the restricted strong convexity theory to see whether we could get the same covergence rate $O(\sqrt{d_1 + d_2)^3 r/T_1})$ in GLM as in the linear case. Specifically, we use the regularized negative log-likelihood minimization with nuclear norm penalty for the loss function in stage 1, and consequently we are able to get the same bound as in the linear case. Notice that this work is also non-trivial since constructing the restricted strong convexity for the generalized linear low-rank matrix estimation requires us to use a truncation argument and a peeling technique [30], which is completely different that used in simple linear case [26]. Therefore, to facilitate further study in this area and for the completeness of our work, we would present the detailed proof here in the following as a bonus. Notice that if we use the following method in stage 1, we could get achieve the same regret bound as a scale of $\widetilde{O}((d_1 + d_2)^{3/2}\sqrt{rT})$ in time $T$ by using both G-ESTT or G-ESTS with stage 2 invariant as our main paper. And this alternative bound is worse than the one we get in our main work.

Loss function: we consider the following well-defined regularized negative log-likelihood minimization problem with nuclear norm penalty in stage 1:

$$\widehat{\Theta} = \arg\min_{\Theta \in R^{d_1 \times d_2}} L_{T_1}(\Theta) + \lambda_{T_1} \left\|\Theta\right\|_{\text{nuc}}, \quad \text{where}$$

$$L_{T_1}(\Theta) = \frac{1}{T_1} \sum_{i=1}^{T_1} \{b(\langle X_i, \Theta \rangle) - y_i \langle X_i, \Theta \rangle\}, \tag{23}$$

Different assumptions with notations reloaded:

**Assumption J.1.** There exists a sampling distribution $\mathcal{D}$ over $\mathcal{X}$ with covariance matrix of $\text{vec}(X)$ as $\Sigma \in \mathbb{R}^{d_1 d_2 \times d_1 d_2}$, such that $\lambda_{\min}(\Sigma) = \lambda_1$ and $\text{vec}(X)$ is sub-Gaussian with parameter $\sigma = \lambda_2$ such that $\lambda_1/\lambda_2^2$ can be absolutely bounded.

**Assumption J.2.** The norm of true parameter $\Theta^*$ and feature matrices in $\mathcal{X}$ is bounded: there exists $S \in \mathbb{R}^+$ such that for all arms $X \in \mathcal{X}$, $\|X\|_F, \|\Theta^*\|_F \leq S$; $\|X\|_{\text{op}}, \|\Theta^*\|_{\text{op}} \leq S_2 (S_2 \leq S)$.

**Assumption J.3.** The inverse link function $\mu(\cdot)$ is continuously differentiable, Lipschitz with constant $k_\mu$. $c_\mu \geq \inf_{\Theta \in \Theta, X \in \mathcal{X}} \mu'(\langle X, \Theta \rangle) > 0$ and $c_\mu \geq \inf_{\{|x| < (S+2)\sigma c_2\}} \mu'(x) > 0$ for some constant $c_2$.

Here we could safely choose $\sigma = 1/\sqrt{d_1 d_2}$ [26] as default. Without loss of generality, we can assume that $c_- \sigma^2 \leq \{\lambda_1, \lambda_2^2\} \leq c_+ \sigma^2$ for some absolute constant $c_-, c_+$ for the simplicity of following theoretical analysis. Assumption J.1 implies that if $X$ is sampled from the distribution $\mathcal{D}$, then for any $\Delta \in \mathbb{R}^{d_1 \times d_2}$ satisfying $\|\Delta\|_F \leq 1$, we have:

$$\mathbb{E}[\langle X, \Delta \rangle^2] = \text{vec}(\Delta)^\top \Sigma \text{vec}(\Delta) \geq \lambda_1 \geq c_- \sigma^2 := \alpha; \tag{24}$$

$$\mathbb{E}[\langle X, \Delta \rangle^4] \leq 16\lambda_2^4 \leq 16c_+^2 \sigma^4 := \beta. \tag{25}$$

## J.2 Theorem

**Theorem J.4.** (Bounds for GLM via another loss function in Eqn. (23)) *For any low-rank generalized linear model with samples $X_1 \ldots, X_{T_1}$ drawn from $\mathcal{X}$ according to $\mathcal{D}$ in Assumption J.1, and Assumption J.2, J.3 hold. Then the optimal solution to the nuclear norm regularization problem* (23) *with $\lambda_{T_1} = \Omega(\sigma\sqrt{(d - \log(\delta))/T_1})$ would satisfy:*

$$\left\|\widehat{\Theta} - \Theta^*\right\|_F^2 \asymp \frac{d}{T_1 \sigma^2} r \asymp \frac{d^3 r}{T_1}, \tag{26}$$

*with probability at least $1 - \delta$ given the condition $dr \lesssim \sigma^2 T_1$ and $(1 + \sigma)^2 dr \lesssim T_1$ hold.*

To prove this theorem, roughly speaking we firstly deduce the restricted strong convexity condition for our optimization problem with high probability, and then extend some previous results on the oracle inequality of estimation error.

**Remark.** We explain why in theory our Stein's-lemma-based method is better than this classic matrix estimation approach for our problem setting here: Intuitively, we take advantage of the fact that only singular subspaces spanned by $\Theta^*$ are required for our transformation in stage 2, but its exact singular values are not necessary. Therefore, we introduced our Stein's-lemma-based quadratic optimization problem, which particularly focuses on subspace detection and hence is more appropriate for our explore-then-commit frameworks. In other words, our Stein's-lemma-based matrix recovery method could only detect the subspace precisely, but cannot estimate the exact singular values since there are some unknown non-zero constant $\mu^*$ in the loss $\left\|\hat{\Theta} - \Theta^*\right\|_F$. However, our frameworks only rely on the singular subspaces spanned by the estimate $\hat{\Theta}$ instead of its exact singular values in stage 2, and hence this sacrifice (introducing $\mu^*$) does not affect the Stage 2 and regret bound at all. Note that all existing low-rank matrix estimation methods (e.g. log-likelihood maximization shown here) would waste some information for exact singular value recovery, and hence are inefficient for our problem.

## J.3 Restricted Strong Convexity

**Definition J.5.** (Restricted strong convexity (RSC), [28]). Given the cost function $L_{T_1}(\Theta)$ defined in (6) and $X_1, \ldots, X_{T_1} \in \mathbb{R}^{d_1 \times d_2}$, the first-order Taylor-series error is defined as:

$$\mathcal{E}_{T_1}(\Delta) := L_{T_1}(\Theta^* + \Delta) - L_{T_1}(\Theta^*) - \langle \nabla L_{T_1}(\Theta^*), \Delta \rangle.$$

For a given norm $\|\cdot\|$ and regularizer $\Phi(\cdot)$, the cost function satisfies a *restricted strong convexity* (RSC) condition with radius $R > 0$, curvature $\kappa > 0$ and tolerance $\tau^2$ if

$$\mathcal{E}_{T_1}(\Delta) \geq \frac{\kappa}{2} \|\Delta\|_F^2 - \tau_{T_1}^2 \Phi^2(\Delta), \quad \text{for all } \|\Delta\|_F \leq R.$$

**Theorem J.6.** (RSC for GLM under distribution $\mathcal{D}$). *Consider any low-rank generalized linear model with samples $X_1 \ldots, X_{T_1}$ drawn from $\mathcal{X}$ according to $\mathcal{D}$ in Assumption J.1, and Assumption J.2 and J.3 hold. Then there exists constants $c_3, c_4$ such that with probability $1 - \delta$, we have the* RSC *condition holds:*

$$\mathcal{E}_{T_1}(\Delta) \geq c_3\sigma^2 c_\mu \|\Delta\|_F^2 - (c_4\sigma^2 + 2\sigma)\left(\sqrt{\frac{d_1}{T_1}} + \sqrt{\frac{d_2}{T_1}}\right) c_\mu \|\Delta\|_{\text{nuc}}^2 \quad \text{for all } \|\Delta\|_F \leq 1$$

(27)

*with $\kappa = c_3\sigma^2 c_\mu$, $\tau_{T_1}^2 = (c_4\sigma^2 + 2\sigma)\left(\sqrt{\frac{d_1}{T_1}} + \sqrt{\frac{d_2}{T_1}}\right) c_\mu$, $R = 1$, $\|\cdot\| = \|\cdot\|_F$ and $\Phi(\cdot) = \|\cdot\|_{\text{nuc}}$*

*for $T_1 = O(\log(\log_2(d)/\delta))$.*

*Remark* J.7. The radius $R$ in Theorem J.6 can be adapted to any finite positive constant keeping the same proof outline. And the required sample size $T_1$ only change in logarithmic power, which can be easily satisfied.

### J.3.1 Proof of Theorem J.6

To prove theorem J.6, we use a truncation argument and the peeling technique [30, 35]:

Using the property of the remainder in the Taylor series, we have

$$\mathcal{E}_{T_1}(\Delta) = \frac{1}{T_1}\sum_{i=1}^{T_1} \mu'\left(\langle X_i, \Theta^*\rangle + t\langle X_i, \Delta\rangle\right)\langle X_i, \Delta\rangle^2,$$

for some $t \in [0, 1]$. Based on (24) and (25) we will set two truncation parameters $K_1^2 = 4\beta/\alpha$ and $K_2^2 = 4\beta S^2/\alpha$ for further use. For any $\|\Delta\|_F = \delta \in (0, 1]$, we set $\tau = K_1\delta$ and a trunction function $\phi_\tau(v) = v^2 \cdot I_{\{|v| \leq 2\tau\}}$. Then we have:

$$\mathcal{E}_{T_1}(\Delta) \geq \frac{1}{T_1}\sum_{i=1}^{T_1} \mu'\left(\langle X_i, \Theta^*\rangle + t\langle X_i, \Delta\rangle\right)\phi_\tau(\langle X_i, \Delta\rangle)I_{\{|\langle X_i, \Theta^*\rangle| \leq K_2\}}.$$

The right hand side would always be 0 if $|\langle X_i, \Theta^*\rangle + t\langle X_i, \Delta\rangle| > 2K_1 + K_2$, which implies the following result based on Assumption J.3:

$$\mathcal{E}_{T_1}(\Delta) \geq c_\mu\frac{1}{T_1}\sum_{i=1}^{T_1} \phi_\tau(\langle X_i, \Delta\rangle)I_{\{|\langle X_i, \Theta^*\rangle| \leq K_2\}}.$$

Therefore, it suffices to how that for all $\delta \in (0, 1]$ and for $\|\Delta\|_F = \delta$, we have:

$$\frac{1}{T_1}\sum_{i=1}^{T_1} \phi_{\tau(\delta)}(\langle X_i, \Delta\rangle)I_{\{|\langle X_i, \Theta^*\rangle| \leq K_2\}} \geq a_1\delta^2 - a_2\|\Delta\|_{\text{nuc}}\,\delta,$$

(28)

for some parameters $a_1$ and $a_2$ since the inequality $\|\Delta\|_F \leq \|\Delta\|_{\text{nuc}}$ always holds. Note the fact that $\phi_{\tau(\delta)}(\langle X_i, \Delta\rangle) = \delta^2\phi_{\tau(1)}(\langle X_i, \Delta/\delta\rangle)$, then for any $\|\Delta\|_F = \delta$ such that $\delta \in (0, 1]$, we can apply bound (28) to the rescaled unit-norm matrix $\Delta/\delta$ to obtain:

$$\frac{1}{T_1}\sum_{i=1}^{T_1} \phi_\tau(1)(\langle X_i, \Delta/\delta\rangle)I_{\{|\langle X_i, \Theta^*\rangle| \leq K_2\}} \geq a_1 - a_2\|\Delta/\delta\|_{\text{nuc}},$$

which implies that it suffices to show (28) holds when $\delta = 1$, i.e.

$$\frac{1}{T_1}\sum_{i=1}^{T_1} \phi_\tau(\langle X_i, \Delta\rangle)I_{\{|\langle X_i, \Theta^*\rangle| \leq K_2\}} \geq a_1 - a_2\|\Delta\|_{\text{nuc}}, \quad \text{for all } \|\Delta\|_F = 1.$$

Then we can construct another truncation function $\tilde{\phi}_\tau(v)$ with parameter at most $2\tau = 2K_1$ as

$$\tilde{\phi}_\tau(v) = v^2 I_{\{|v| \leq \tau\}} + (v - 2\tau)^2 I_{\{\tau < v \leq 2\tau\}} + (v + 2\tau)^2 I_{\{-2\tau \leq v < -\tau\}}.$$

Then it suffices to show that

$$\frac{1}{T_1} \sum_{i=1}^{T_1} \tilde{\phi}_\tau(\langle X_i, \Delta \rangle) I_{\{|\langle X_i, \Theta^* \rangle| \leq K_2\}} \geq a_1 - a_2 \|\Delta\|_{\text{nuc}}, \quad \text{for all } \|\Delta\|_F = 1.$$

And for a given radius $r \geq 1$, define the random variable

$$Z_{T_1}(r) = \sup_{\substack{\|\Delta\|_F = 1, \\ \|\Delta\|_{\text{nuc}} \leq r}} \left| \frac{1}{T_1} \sum_{i=1}^{T_1} \tilde{\phi}_\tau(\langle X_i, \Delta \rangle) I_{\{|\langle X_i, \Theta^* \rangle| \leq K_2\}} - \mathbb{E}\left( \tilde{\phi}_\tau(\langle X, \Delta \rangle) I_{\{|\langle X, \Theta^* \rangle| \leq K_2\}} \right) \right|.$$

Firstly, we can prove that

$$\mathbb{E}[\tilde{\phi}_\tau(\langle X, \Delta \rangle) I_{\{|\langle X, \Theta^* \rangle| \leq K_2\}}] \geq \frac{1}{2}\alpha, \tag{29}$$

by using the chosen values for $K_1$ and $K_2$ to show that

$$\mathbb{E}[\tilde{\phi}_\tau(\langle X, \Delta \rangle)] \geq \frac{3}{4}\alpha, \quad \mathbb{E}[\tilde{\phi}_\tau(\langle X, \Delta \rangle) I_{\{|\langle X, \Theta^* \rangle| > K_2\}}] \leq \frac{1}{4}\alpha.$$

Specifically, since we have

$$\mathbb{E}[\tilde{\phi}_\tau(\langle X, \Delta \rangle)] \geq \mathbb{E}[\langle X, \Delta \rangle^2 I_{\{|\langle X, \Theta^* \rangle| \leq \tau\}}] \geq \alpha - \mathbb{E}[\langle X, \Delta \rangle^2 I_{\{|\langle X, \Theta^* \rangle| > \tau\}}]$$

And we can show that the last term is at most $\alpha/4$ based on the Markov's inequality and Cauchy-Schwarz inequality:

$$\mathbb{E}[\langle X, \Delta \rangle^2 I_{\{|\langle X, \Theta^* \rangle| > \tau\}}] \leq \sqrt{\mathbb{E}[\langle X, \Delta \rangle^4]} \sqrt{P(|\langle X, \Theta^* \rangle| > \tau)} \leq \sqrt{\beta} \sqrt{\frac{\beta}{\tau^4}} \leq \frac{\alpha}{4}.$$

And similarly we can prove that $\mathbb{E}[\tilde{\phi}_\tau(\langle X, \Delta \rangle) I_{\{|\langle X, \Theta^* \rangle| > K_2\}}] \leq \alpha/4$. On the other hand, by our choice $\tau = K_1$, the empirical process defining $Z_{T_1}(r)$ is based on functions bounded in absolute value by $K_1^2$. Thus, the functional Hoeffding inequality (Theorem 3.26 in [35]) implies that

$$P\left( Z_{T_1}(r) \geq \mathbb{E}(Z_{T_1}(r)) + \sigma r \left( \sqrt{\frac{d_1}{T_1}} + \sqrt{\frac{d_2}{T_1}} \right) + \frac{\alpha}{4} \right) \leq$$

$$\exp\left( -\frac{n \left( \sigma r \left( \sqrt{\frac{d_1}{T_1}} + \sqrt{\frac{d_2}{T_1}} \right) + \frac{\alpha}{4} \right)^2}{4K_1^4} \right). \tag{30}$$

To bound the expected value term $\mathbb{E}(Z_{T_1}(r))$, we introduce an i.i.d sequence of Rademacher variables $\{\varepsilon_i\}_{i=1}^{T_1}$ and then use the symmetrization argument:

$$
\begin{aligned}
\mathbb{E}(Z_{T_1}(r)) &= \mathbb{E}\left[\sup_{\substack{\|\Delta\|_F=1,\\ \|\Delta\|_{\text{nuc}}\leq r}} \left| \frac{1}{T_1}\sum_{i=1}^{T_1} \tilde{\phi}_\tau(\langle X_i,\Delta\rangle)I_{\{|\langle X_i,\Theta^*\rangle|\leq K_2\}} - \mathbb{E}\left(\tilde{\phi}_\tau(\langle X,\Delta\rangle)I_{\{|\langle X,\Theta^*\rangle|\leq K_2\}}\right)\right|\right] \\
&= \mathbb{E}\left[\sup_{\substack{\|\Delta\|_F=1,\\ \|\Delta\|_{\text{nuc}}\leq r}} \left| \frac{1}{T_1}\sum_{i=1}^{T_1} \tilde{\phi}_\tau(\langle X_i,\Delta\rangle)I_{\{|\langle X_i,\Theta^*\rangle|\leq K_2\}} - \mathbb{E}\left(\frac{1}{T_1}\sum_{i=1}^{T_1}\tilde{\phi}_\tau(\langle Y_i,\Delta\rangle)I_{\{|\langle Y_i,\Theta^*\rangle|\leq K_2\}}\right)\right|\right] \\
&\leq \mathbb{E}_{X_i,Y_i}\left[\sup_{\substack{\|\Delta\|_F=1,\\ \|\Delta\|_{\text{nuc}}\leq r}} \left| \frac{1}{T_1}\sum_{i=1}^{T_1} \tilde{\phi}_\tau(\langle X_i,\Delta\rangle)I_{\{|\langle X_i,\Theta^*\rangle|\leq K_2\}} - \frac{1}{T_1}\sum_{i=1}^{T_1}\tilde{\phi}_\tau(\langle Y_i,\Delta\rangle)I_{\{|\langle Y_i,\Theta^*\rangle|\leq K_2\}}\right|\right] \\
&= \mathbb{E}_{X_i,Y_i,\varepsilon_i}\left[\sup_{\substack{\|\Delta\|_F=1,\\ \|\Delta\|_{\text{nuc}}\leq r}} \left| \frac{1}{T_1}\sum_{i=1}^{T_1} \varepsilon_i\left(\tilde{\phi}_\tau(\langle X_i,\Delta\rangle)I_{\{|\langle X_i,\Theta^*\rangle|\leq K_2\}} - \tilde{\phi}_\tau(\langle Y_i,\Delta\rangle)I_{\{|\langle Y_i,\Theta^*\rangle|\leq K_2\}}\right)\right|\right] \\
&\leq 2\mathbb{E}_{X_i,\varepsilon_i}\left[\sup_{\substack{\|\Delta\|_F=1,\\ \|\Delta\|_{\text{nuc}}\leq r}} \left| \frac{1}{T_1}\sum_{i=1}^{T_1} \varepsilon_i\tilde{\phi}_\tau(\langle X_i,\Delta\rangle)I_{\{|\langle X_i,\Theta^*\rangle|\leq K_2\}}\right|\right] \\
&\overset{(i)}{\leq} 8K_1\mathbb{E}_{X_i,\varepsilon_i}\left[\sup_{\substack{\|\Delta\|_F=1,\\ \|\Delta\|_{\text{nuc}}\leq r}} \left| \frac{1}{T_1}\sum_{i=1}^{T_1} \varepsilon_i\langle\Delta,X_i\rangle\right|\right] \overset{(ii)}{\leq} 8K_1 r\cdot\mathbb{E}_{X_i,\varepsilon_i}\left[\left\|\frac{1}{T_1}\sum_{i=1}^{T_1}\varepsilon_i X_i\right\|_{\text{op}}\right],
\end{aligned}
\tag{31}
$$

where the inequality (i) comes from Rademacher contraction property and (ii) is by the duality between matrix $\|\cdot\|_2$ and $\|\cdot\|_{\text{nuc}}$ norms. Using the previous conclusion (Exercise 9.8 in [35]), we have

$$
\mathbb{E}_{X_i,\varepsilon_i}\left[\left\|\frac{1}{T_1}\sum_{i=1}^{T_1}\varepsilon_i X_i\right\|_{\text{op}}\right] \leq \sigma c_5 c_+\left(\sqrt{\frac{d_1}{T_1}}+\sqrt{\frac{d_2}{T_1}}\right),
\tag{32}
$$

where $c_5$ is an independent absolute constant. Combine (30), (31) and (32), we have

$$
P\left(Z_{T_1}(r)\geq (8K_1 c_5 c_+ + 1)\sigma r\left(\sqrt{\frac{d_1}{T_1}}+\sqrt{\frac{d_2}{T_1}}\right)+\frac{\alpha}{4}\right) \leq \exp\left(-\frac{T_1\left(\sigma r\left(\sqrt{\frac{d_1}{T_1}}+\sqrt{\frac{d_2}{T_1}}\right)+\frac{\alpha}{4}\right)^2}{4K_1^4}\right).
\tag{33}
$$

According to (29) and (33), we prove the following conclusion for any fixed value of radium $r$:

$$
P\left(\sup_{\substack{\|\Delta\|_F=1,\\ \|\Delta\|_{\text{nuc}}\leq r}}\mathcal{E}_{T_1}(\Delta)<\frac{1}{4}\alpha c_\mu - (8K_1 c_5 c_+ + 1)\left(\sqrt{\frac{d_1}{T_1}}+\sqrt{\frac{d_2}{T_1}}\right)\sigma c_\mu r\right)\leq
$$

$$
\exp\left(-\frac{T_1\left(\sigma r\left(\sqrt{\frac{d_1}{T_1}}+\sqrt{\frac{d_2}{T_1}}\right)+\frac{\alpha}{4}\right)^2}{4K_1^4}\right).
\tag{34}
$$

Since we have $\|\Delta\|_F = 1$, based on Cauchy-Schwarz inequality we have $1 \leq \|\Delta\|_{\text{nuc}} \leq \sqrt{d}$. To prove the RSC we use a peeling argument to extend $r$ to all possible values. Define the event:

$$E := \left\{ \text{There exists } \Delta \text{ s.t. } \|\Delta\|_F = 1, \; \mathcal{E}_{T_1}(\Delta) < \frac{1}{4}\alpha c_\mu - (16K_1 c_5 c_+ + 2) \right.$$

$$\left. \times \left( \sqrt{\frac{d_1}{T_1}} + \sqrt{\frac{d_2}{T_1}} \right) \sigma c_\mu \|\Delta\|_{\text{nuc}} \right\} \qquad (35)$$

$$V_i := \{2^{i-1} \leq \|\Delta\|_{\text{nuc}} < 2^i\}, \quad i = 1, \ldots, \left\lceil \frac{1}{2} \log_2(d) \right\rceil + 1.$$

Then we can conclude that $E \subseteq \bigcup_{i=1}^{\lceil \frac{1}{2} \log_2(d) \rceil + 1} (E \cap V_i)$. And we can show the probability of each partition event $(E \cap V_i)$ can be upper bounded by (34):

$$P(E \cap V_i) = P \left( \sup_{\substack{\|\Delta\|_F = 1, \\ 2^{i-1} \leq \|\Delta\|_{\text{nuc}} < 2^i}} \mathcal{E}_{T_1}(\Delta) < \frac{1}{4}\alpha c_\mu - (16K_1 c_5 c_+ + 2) \left( \sqrt{\frac{d_1}{T_1}} + \sqrt{\frac{d_2}{T_1}} \right) \sigma c_\mu \|\Delta\|_{\text{nuc}} \right)$$

$$\leq P \left( \sup_{\substack{\|\Delta\|_F = 1, \\ 2^{i-1} \leq \|\Delta\|_{\text{nuc}} < 2^i}} \mathcal{E}_{T_1}(\Delta) < \frac{1}{4}\alpha c_\mu - (8K_1 c_5 c_+ + 1) \left( \sqrt{\frac{d_1}{T_1}} + \sqrt{\frac{d_2}{T_1}} \right) \sigma c_\mu 2^i \right)$$

$$\leq \exp \left( - \frac{T_1 \left( 2^i \sigma \left( \sqrt{\frac{d_1}{T_1}} + \sqrt{\frac{d_2}{T_1}} \right) + \frac{\alpha}{4} \right)^2}{4K_1^4} \right),$$

which implies that

$$P(E) \leq \log_2(d) \exp \left( - \frac{T_1 \left( 2\sigma \left( \sqrt{\frac{d_1}{T_1}} + \sqrt{\frac{d_2}{T_1}} \right) + \frac{\alpha}{4} \right)^2}{4K_1^4} \right).$$

We complete our proof of Theorem J.6 by noticing that the constants $c_3, c_4$ in (27) only depend on the absolute constants $c_5, c_+$ and $c_-$ through our proof. $\qquad \square$

## J.4   Technical Lemmas

**Lemma J.8.** (Bound for GLM with nuclear regualarization, [28, 35]) *Consider the negative log-likelihood cost function $L_{T_1}(\cdot)$ defined in 6 and observations $X_1, \ldots, X_{T_1}$ satisfy a specific RSC condtion in Definition 1, such that*

$$\mathcal{E}_{T_1}(\Delta) \geq \frac{\kappa}{2} \|\Delta\|_F^2 - \tau_{T_1}^2 \|\Delta\|_{\text{nuc}}^2, \quad \text{for all } \|\Delta\| \leq 1.$$

*Then under the "good" event: $\mathcal{G}(\lambda_{T_1}) := \{\|\nabla L_{T_1}(\Theta^*)\|_{\text{op}} \leq \lambda_{T_1}/2\}$, and the following two conditions hold:*

$$\tau_{T_1}^2 r \leq \frac{\kappa}{128}, \quad 4.5 \frac{\lambda_{T_1}^2}{\kappa^2} r \leq 1.$$

*Then any optimal solution to Eqn. 23 satisfies the bound*

$$\left\| \widehat{\Theta} - \Theta^* \right\|_F^2 \leq 4.5 \frac{\lambda_{T_1}^2}{\kappa^2} r. \qquad (36)$$

$\qquad \square$

## J.5 Proof of Theorem J.4

According to Theorem J.6, there exists two absolute constants $c_3, c_4$ such that with probability at least $1 - \delta$, we have the RSC condition holds:

$$\mathcal{E}_{T_1}(\Delta) \geq c_3 \sigma^2 c_\mu \|\Delta\|_F^2 - (c_4 \sigma^2 + 2\sigma) \left( \sqrt{\frac{d_1}{T_1}} + \sqrt{\frac{d_2}{T_1}} \right) c_\mu \|\Delta\|_{\text{nuc}}^2 \quad \text{for all } \|\Delta\|_F \leq 1.$$

To implement Lemma 1, we would like to to figure out the value for regularization parameter $\lambda_{T_1}$ such that the event $\mathcal{G}(\lambda_{T_1})$ can hold with high probability and simultaneously the bound in (36) can be well controlled. The proof is by using the covering argument and Bernstein's inequality to bound the operator norm.

Let $\xi_i = \langle X_i, \Theta^* \rangle$, we have $\|\nabla L_{T_1}(\Theta^*)\|_{\text{op}} = \left\| \frac{1}{n} \sum_{i=1}^{T_1} (b'(\xi_i) - y_i) X_i \right\|_{\text{op}}$, and for all $i \in [T_1]$

$$\mathbb{E}[(b'(\xi_i) - y_i) X_i] = \mathbb{E}\left[ X_i \, \mathbb{E}[b'(\xi_i) - y_i \mid X_i] \right] = 0.$$

Let $\mathcal{S}^{d_1}$ ($\mathcal{S}^{d_2}$) be the $d_1$ ($d_2$) dimensional Euclidean-norm unit sphere, and $\mathcal{N}^{d_1}$ ($\mathcal{N}^{d_2}$) be the $1/4$ covering on $\mathcal{S}^{d_1}$ ($\mathcal{S}^{d_2}$) and $\Xi(A) = \sup_{\substack{u \in \mathcal{N}^{d_1}, \\ v \in \mathcal{N}^{d_2}}} u^\top A v$ for all $A \in \mathbb{R}^{d_1 \times d_2}$. By the proof of Lemma 1 in [10], we know that

$$\|A\|_{\text{op}} \leq \frac{16}{7} \Xi(A). \tag{37}$$

Besides, based on the properties of Orlicz-1 norm and Orlicz-2 norm, we have:

$$\left\| (b'(\xi_i) - y_i) u^\top X_i v \right\|_{\psi_1} \leq \|(b'(\xi_i) - y_i)\|_{\psi_2} \left\| u^\top X_i v \right\|_{\psi_2} \leq c_6 \sqrt{k_\mu} \lambda_2, \quad \text{for all } u \in S^{d_1}, v \in S^{d_2}.$$

For some absolute constant $c_6$ (e.g. $c_6 = 6$). Then for any fixed $u \in S^{d_1}$, $v \in S^{d_2}$, by Berstein's inequality we have

$$P\left( \left| \frac{1}{T_1} \sum_{i=1}^{T_1} (b'(\xi_i) - y_i) u^\top X_i v \right| > t \right) \leq 2 \exp\left[ -c_7 \min\left( \frac{T_1 t^2}{c_6^2 k_\mu \lambda_2^2}, \frac{T_1 t}{c_6 \sqrt{k_\mu} \lambda_2} \right) \right].$$

Then by the combination over all the union bounds and relation (37) we can claim that

$$P\left( \left\| \frac{1}{T_1} \sum_{i=1}^{T_1} (b'(\xi_i) - y_i) X_i \right\|_{\text{op}} > \frac{16}{7} t \right) \leq 2 \, 7^{d_1 + d_2} \exp\left[ -c_7 \min\left( \frac{T_1 t^2}{c_6^2 k_\mu \lambda_2^2}, \frac{T_1 t}{c_6 \sqrt{k_\mu} \lambda_2} \right) \right].$$

Then the event $\{\|\nabla L_{T_1}(\Theta^*)\|_2 \geq \frac{16}{7} t\}$ holds with probability $1 - \delta$ if

$$t = \sqrt{k_\mu} \lambda_2 \max\left( \sqrt{\frac{c_6 (d_1 + d_2) \log(7) + c_6 \log(2/\delta)}{T_1}}, \frac{c_6 (d_1 + d_2) \log(7) + c_6 \log(2/\delta)}{T_1} \right)$$

$$= \Omega\left( \sqrt{\frac{d_1 + d_2 - \log(\delta)}{T_1}} \, \sigma \right).$$

Since we assume $(d_1 + d_2) \lesssim T_1$. By taking $\lambda_{T_1} = \frac{32}{7} t \asymp \sqrt{\frac{d_1 + d_2 - \log(\delta)}{T_1}} \sigma$. We complete the proof of Theorem J.4 and obtain the scale of the bound in (26) after plugging the chosen values of $\kappa$ and $\lambda_{T_1}$ into (36). $\qquad\square$

Notice that the loss function here shown in Eqn. (23) is also convex and hence could be solved by a wide class of optimization methods (e.g. subgradient descent algorithm), and we have the convergence rate of matrix estimation as

$$\left\| \widehat{\Theta} - \Theta^* \right\|_F = \widetilde{O}\left( \sqrt{\frac{d^3 r}{T_1}} \right)$$

which is greater than the rate we deduced in Theorem 4.1 of our main paper as a factor of $\widetilde{O}(\sqrt{d})$. This also explicitly manifests the high efficiency of our methods with Stein's method in stage 1. Furthermore, if we use the methodology in Appendix J for stage 1 instead, and keep all the algorithms setting in state 2 fixed and modify $T_1$ accordingly, we can show that our G-ESTT and G-ESTS can achieve the same regret bound $\tilde{O}((d_1 + d_2)^{3/2}\sqrt{rT})$ as in [26].

We also offer an explanation why we get an improved convergent bound by using Stein's type Lemma in Theorem 4.1 than using the RSC in Theorem J.4: Intuitively, we obtain the improved convergence rate of order $d^2 r/T$ in stage 1 at the expense of introducing a non-zero constant $\mu^*$ in the loss $\left\|\widehat{\Theta} - \mu^*\Theta^*\right\|_F$. However, our algorithm only uses the singular subspaces spanned by $\widehat{\Theta}$ instead of its exact singular values and hence this sacrifice ($\mu^*$) does not affect the regret. This implies our matrix recovery method based on Stein's Lemma is more efficient for subspace learning.