# OpenReview forum: "Efficient Frameworks for Generalized Low-Rank Matrix Bandit Problems"
_NeurIPS.cc/2022/Conference — NeurIPS 2022 Accept_

### Official Review · Reviewer_gC7G · 2022-07-01

**Rating:** 7
**Confidence:** 4
**Soundness:** 3 good
**Presentation:** 3 good
**Contribution:** 4 excellent

**Summary:**

This paper deals with the generalized low-rank matrix bandit problem, where for each time the learner choose one action with its own feature matrix, and receives a noisy reward by the inner product of the chosen action feature matrix and a hidden matrix $\Theta^*$ with know rank $r$.

This paper uses ingenious Stein's method based approach to estimate the subspace even more accurately compare to the previous approaches, and this is the core originality of this paper. Thanks to this new estimation, the author invented the algorithms G-ESTT and G-ESTS, which shows $O(dr\sqrt{T}/D_{rr})$ regret bound which matches to the known regret lower bound of the generalized linear bandit problem. The second algorithm, G-ESTS, removes the redundant dimensional parts in calculation and still theoretically performs well.

The author also shows the practicality of their algorithm by the experiment.

**Questions:**

Main question (will change my opinion)

about the reusing the data

I feel quite suspicious about reusing the data. In line 195-206, they advertise that they need no more projection or additional sampling by reusing the data from Stage 1. However, I doubt about the independence. The samples in Stage 1 already affects the coordinates, and this re-organizing coordinate affects the calculation of $ {L}_t^{\Lambda} (\theta) $, especially about the last regularizer term $\|\theta\|_\Lambda$. The new coordinate is decided by $X_i ( i=1, \cdots t)$ and every calculation is based on this new coordinate rotation. $\Lambda$ penalizes some of the new coordinates where the experiment results from $X_i ( i=1\cdots t)$ shows high regret. Are you sure that you can guarantee the independence of the result and okay to mix the data from the 1st stage and 2nd stage?

I believe that this is the point that the author should clarify.



Other questions

1) Could you tell me intuitively why the Stein's method works better to estimate matrix completion compare to the previous approaches? I read your proof and understand that your proof will be correct (about theorem 2), but I want to get some hunch about why it shows more accurate estimation ($\frac{dr}{T}$) in terms of Frobenius norm than the previous approach ($\frac{\sqrt{d^3r}}{T}$).

2) This paper assumes that we already got a distribution $\mathcal{D}$ which satisfies a certain assumption. It would be great if this paper describes how to get such a distribution from a given action set. I know that you described something in Line 129-135, but it doesn't make sense to me, even after I read your Appendix I. You said,

1-find the convex hull of $\mathcal{X}$ which contains the ball of radius R
2-$p_ij$ is now centered gaussian with variance $\frac{R^2}{d_1 d_2}$

Are you sure that sample from $\mathcal{D}$ always lies in $\mathcal{X}$? For example, imagine a thin shell action set with out-radius R and inner radius 0.99R. Then there might be almost no hope that your distribution lies in this shell, right? In fact, there's no detail description about the sampling method in Appendix I. Please let me know if there's a detail I missed.

There are several papers related with this exploration planning in the linear bandit field, especially using something like G-Optimality. I want this level clear, if possible.

3) Is the minimum eigenvalue condition really necessary in this problem? Could you introduce any related works where this term is inevitable? At least (Jang et al, 2021) didn't have the minimum eigenvalue term in the algorithm as far as I remember.

**Limitations:**

1) There is a hidden factor in regret which is the 'minimum eigenvalue' term. It is not intuitive to me that when minimum eigenvalue is extremely small, this matrix can be additional dimensional dependency which makes this paper weaker.

2) The way to draw the samples should be suggested. The author should prove that their assumption is really a mild one.

**Strengths And Weaknesses:**

1) Originality:
1-1) Strength

Their strength is the new subspace estimation method, which is based on a different approach called Stein's method. As far as I know, there was no similar work using Stein's method in this field.

They also provide a bold subtraction approach (in ESTS). Previous papers with similar structure with this paper usually keep caring of the redundant dimensions, $(r+1: d_1, r+1:d_2)$, for the rest of the bandit calculation (mainly for the theoretical safety). This paper proves that the learner don't need to care the redundant dimensions anymore.

This paper also introduces Stein's method for the matrix completion, which provides more accurate estimation than the previous approaches.

1-2) Weakness:

This paper is basically follows the similar structure with ESTR (Jun et al, 2019, Lu et al, 2021) and shares the similar drawback.
Most importantly, the regret bound depends on the size of the minimum eigenvalue of $\Theta^*$.
This is of course not intuitive - it's like, for the traditional multi-armed bandit sense, the regret increases when the reward of the minimum reward arm is small. Usually the gap between the optimal and sub-optimal is important in the regret analysis of the bandit, not the minimum one.
I think this is a kind of the hidden dimensional dependency for the result of this paper, and should be improved in the future.



2) Quality - Problematic

Overall, the proof seems correct for the core part(Theorem 4.1 and Theorem 4.2 - assuming their re-using the data is not problematic) so their claim is supported by theoretical analysis. It also provides several theoretical discussions in the Appendix, like Appendix J.

However, I feel quite suspicious about reusing the data. In line 195-206, they advertise that they need no more projection or additional sampling by reusing the data from Stage 1. However, I doubt about the independence. The samples in Stage 1 already affects the coordinates, and this re-organizing coordinate affects the calculation of ${L}_t^{\Lambda} (\theta)$, especially about the last regularizer term $\|\theta\|_\Lambda$. The new coordinate is decided by $X_i (i=1, \cdots t)$ and every calculation is based on this new coordinate rotation. $\Lambda$ penalizes some of the new coordinates where the experiment results from $X_i (i=1, \cdots t)$ shows high regret. Are you sure that you can guarantee the independence of the result and okay to mix the data from the 1st stage and 2nd stage?

I believe that this is the point that the author should clarify.


Plus, I am not clear how to get the distribution $\mathcal{D}$ if only action sets are given. If it is difficult to create such distribution, then Assumption 3.3 is not trivial, which makes this paper weaker.


3) Clarity:

Basically, the paper describes what they do with some clarity, but of course there are some typos and abuse of notations. I think these are minor, and I believe these drawbacks will be fixed after this paper is accepted.
- The use of $S$ is very confusing. Sometimes it is the bound of $\|\Theta^*\|$, and sometimes it is the score function when the distribution is 'obvious'. I was spending quite a time for finding the line 119.
- In line 152, I want the writers to clarify that they will apply the function entrywise.
- In line 175 and Eq. (8), I believe this result is not that obvious. At least make a section in the Appendix will be great (or at least the exact place of reference)
- In Eq. (10), the time index for two summation ($T_1$ and $t$) confuses the reader. I believe it might be the first exploration stage and the second bandit stage, but it confuses me first.
- In Lemma B.2, what is g?
- In the last equation of line 457, it seems like the index is missing in the summation.
- In line 458, first inequality, $n$ to $T_1$

Not about clarity, but it is known that if X is $\sigma$-subgaussian, then $var(X)\leq \sigma^2$. See the Lemma 5.4 of the book 'Bandit Algorithm', Lattimore and Szepesvari




4) Significance:

4-1) Strength

This paper proves the matching upper bound compare to the lower bound proved by Lu et al (2017), and finishes the argument what is the optimal regret order for the generalized low-rank bandit. It also introduces a new method called Stein's method to this field.

4-2) Weakness

This matrix bandit field itself is a relatively small field, so it would be better if the author explains how his ingenious approach can be extended to other larger fields.

---

> ### Author Response · Authors · 2022-08-02
> **Thank you very much for your insightful comments. We sincerely appreciate your time and detailed review, and please see our responses below:**
>
> Thanks again for helping improve the quality of our paper
>
> Weakness1 ***"dependence on the minimum eigenvalue of $\Theta^\*$...."** and Q3 **"Is the minimum eigenvalue condition really necessary in this problem?......"**
>
> We will respond to these two points together. Thanks for pointing it out. Since our frameworks rely on the subspace detection in Stage 1, we would assume the minimum non-zero singular value is not too small so that the corresponding subspace could be distinguished from the null space. All the explore-then-commit-type low-rank matrix bandit algorithms would depend on this condition. (e.g. ESTR, LowESTR). And since under our problem setting the parameter matrix $\Theta^*$ ($d_1 \times d_2$) has a low rank structure, i.e. r=rank($\Theta^*$) $<< d_1,d_2$, we believe it is quite reasonable to assume the smallest non-zero singular value is of constant scale, or at least not suppressed by some order of $d$. Furthermore, Jang et al [2] mainly study the bilinear bandit, which is only a special case of our problem setting. And their creative theoretical analysis is not applicable when the action space is no longer bilinear.
>
> Weakness 2 and main question **"about reusing the data"**
>
> Thanks for your perceptive thought. We will explain why our regret bound given in Theorem 4.2, and the consistency of the M-estimator $\hat\theta_t$ of $\tilde L_t^{\Lambda} (\theta)$ holds intuitively and theoretically.
>
> The proof of regret bound in Theorem 4.2 is mainly based on Theorem C.1 and Proposition C.2 given in Appendix C. And their proof doesn't involve the dependence between samples in Stage 1 and the coordinate rotation rule. Specifically, for the reused data $x_{s_1,k}$ and $y_{s_1,k}$, we mainly use the fact that $y_{s_1,k} - \mu(x_{s_1,k}^\top \theta^*)= \eta_{s_1,k}$ and then use the martingale inequality on these i.i.d. sub-Gaussian errors $\eta$. Note that $x_{t}^\top \theta^* = <X_t,\Theta^*>$ always holds since we apply the same orthogonal transformation and vectorization rule on $X_t$ and $\Theta^*$, and the dependence between the coordinate system and the reused data will not affect this equality. For the penalization matrix $\Lambda$ we mainly use the fact that $\Lambda$ is a positive definite matrix.
>
> For the consistency of M-estimator $\hat\theta_t$, we first offer an intuitive explanation: [1] proved the consistency of the MLE $\hat\theta_{t}$ without the regularizer $\theta^\top \Lambda \theta$. Regarding the penalty $\theta^\top \Lambda \theta$, for the first $k$ entries of $\theta$, the penalized parameter $\lambda_0$ is small, and hence it will have minimal effect after we have sufficient samples. For the remaining $(p-k)$ elements suffering a large penalty, the estimated $\hat \theta_{t,k+1:p}$ would be very small in magnitude as desired since we argue that after the transformation ${\theta_{k+1:p}^*}$ will also be insignificant.
> This implies that $||{\hat \theta_{t,k+1:p}-\theta_{k+1:p}^*}||$ is close to null. As a result, the estimated $\hat \theta_t$ tends to be consistent. Note $\Lambda$ mainly penalizes the coordinates that should be close to zero after our transformation and re-organization, which means this regularizer term actually improves but not hurts the consistency. We will add this explanation in Appendix D, and our experimental results also support this argument. In particular, G-ESTT would outperform G-ESTS in the beginning of stage 2 when the underlying parameter matrix $\Theta^*$ is complicated (Figure 1(c,d), Figure 2(c,d)), and it is because G-ESTT could reuse the samples in stage 1 to yield more robust performance when switching to stage 2.
>
> For the theoretical proof of consistency, we follow a similar argument as the proof of step 1 in Theorem 1 in [1]. And the most important part is to prove that $\lambda_{\min}(V)$ is large enough where $V = \sum_{i=1}^{T_1} x_{s1,i}x_{s1,i}^\top$. First, by assuming the sampling distribution $D$ satisfies sub-Gaussian property with $\sigma$, after any orthogonal transformation this property would still hold. Second, the eigenvalues of $V$ should be identical under arbitrary vectorization rules. (eigenvectors would be different) In other words, the eigenvalues of $V$ are independent of our transformation rule. Therefore, we could still use Proposition 1 in [1] (Proposition in Appendix D) to lower bound $\lambda_{\min}(V)$ after our special transformation.
>
> Weakness 3 **"typos"**
>
> Thank you for pointing them out. All spelling and grammatical errors pointed out here have been corrected in the rebuttal revision.
>
> We answer other questions and concerns in the other comment window due to the length limit. Please let us know if there is any question or comment you may have.
>
> [1] Provably optimal algorithms for generalized contextual bandits. Li et al. ICML 2017
>
> [2] Improved regret bounds of bilinear bandits using action space analysis. Jang et al. ICML 2021
>
> [3] Low-rank generalized linear bandit problems. Lu et al. AISTATS 2021

---

> > ### Comment · Reviewer_gC7G · 2022-08-05
> > **Response to the rebuttal**
> >
> > Thank you for your detailed explanation about it. I believe that you spend a lot of time for this rebuttal...... Thanks for your
> >
> > However, I still have some major questions.
> >
> > 1) About the reusing data, you said it is not related to independence. However, for the proof of C.2 you used Lemma C.3~C.5 which is the traditional technique in linear bandits, but it assumes $\Lambda$ is a constant. In this case, we can say your $\Lambda$ is a random variable decided by your first $t$ samples, since those first samples change the coordinate!
> >
> > Imagine this way - in Equation (9) you used the new coordinate. However, we can re-write the Equation (9) based on the original, untransformed actions and hypothesis. Then, you can see actually $\|\theta\|_\Lambda$ is affected by your first $t$ samples. It makes your $\lambda$, the 'constant' you used in Lemma C.3 or Eq. (19)~ Eq. (20) is not a constant anymore, but just some random variable. I want some explanation about it.
> >
> > Of course I know changing the coordinate does not change $<X, \Theta>$ value. However, $\|\theta\|_\Lambda= \theta^\top \Lambda \theta$ is, in fact, $\theta^top P^\top \Lambda P\theta$ where $P$ is the coordinate changing matrix and $P$ is dependent on the first $t$ data. In this case, I think you have to re-design inequality not from the Abbasi-Yadkori OFUL paper, but your own......
> >
> >
> >
> > 2) About the action set sampling...... it is even more suspicious that you are talking about how to 'approximate' it...... I understand that you are trying to do the nearest neighborhood-ish approximation, but there's no theoretical proof for that sampling satisfies your assumption. It would be great if you can prove that kind of nearest neighborhood preserves Assumption 3.3 properly.

---

> > > ### Author Response · Authors · 2022-08-06
> > > **Thanks again for your comment. Please see our further responses.**
> > >
> > >
> > > 1.
> > > Thanks for your detailed explanation and careful review. Although $\Lambda$ and the coordinate changing matrix $P$ in your comment are determined by the action matrix $X_t \, (t=1,\dots,T_1)$ we pulled in stage 1, we believe they are independent with the transformed vectors $x_{s_1, t} \, (t=1,\dots,T_1)$ we include in Eqn. (9). For example, if we implement any orthogonal transformation on the arm set and the action matrix $X_t \, (t=1,\dots,T_1)$ we pulled in stage 1 as $\{U_0 X_t V_0^\top\}\, (t=1,\dots,T_1)$ where $U_0$ and $V_0$ are $d_1 \times d_1$ and $d_2 \times d_2$ orthogonal matrices respectively. Then the solution to our matrix recovery problem in stage 1 becomes $U_0 \hat\Theta V_0^\top$ where $\hat\Theta$ is the solution without orthogonal transformation. If the SVD of $\Theta$ is equal to $[\hat U, \hat U_{\perp}] \hat D [\hat V, \hat V_{\perp}]^\top$ (line 167), we have the SVD of $U_0 \hat\Theta V_0^\top$ as $[U_0 \hat U, U_0  \hat U_{\perp}] \hat D [V_0 \hat V, V_0 \hat V_{\perp}]^\top$. And then the new rotation parameter space $\mathit{\Theta}$ and action set $\mathit{X}$ (line 171) could be calculated as:
> > >
> > > $$\mathit{\Theta}^\prime = [U_0 \hat U, U_0  \hat U_{\perp}]^\top U_0 \mathit{\Theta} V_0^\top [V_0 \hat V, V_0 \hat V_{\perp}] = [\hat U, \hat U_{\perp}]^\top \mathit{\Theta} [ \hat V, \hat V_{\perp}],$$
> > >
> > > and
> > >
> > > $$\mathit{X}^\prime = [U_0 \hat U, U_0  \hat U_{\perp}]^\top U_0 \mathit{X} V_0^\top [V_0 \hat V, V_0 \hat V_{\perp}] = [\hat U, \hat U_{\perp}]^\top \mathit{X} [ \hat V, \hat V_{\perp}],$$
> > >
> > > which are free of $U_0, V_0$ (same as line 171). As a consequence, we can see that although the coordinate changing matrix $P$ varies arbitrarily according to $U_0$ and $V_0$, the transformed vectors are still the same as ${x_{s_1, t}}\, (t=1,\dots,T_1)$. In other words, although the coordinate changing matrix $P$ depends on the original first $T_1$ samples, it is independent with the values of transformed vectors ${x_{s_1, t}}\, (t=1,\dots,T_1)$, and thus ${\Lambda}$ are not affected by the transformed samples at all. And we can not infer anything about the coordinate changing matrix $P$ from the transformed samples in stage 1. In our Eqn. (9), since we use everything after the same coordinate changing rule, then the dependence doesn’t exist as we explained above, and hence the previous lemmas on linear bandits could be similarly used here. Moreover, since in Eqn. (19) and line 515 we only try to bound the term simultaneously for all $t$ in stage 2 but not in stage 1, and $\Lambda$ is independent with the information in stage 2, the problem is well defined and $\Lambda$ is fixed at the beginning of stage 2.
> > >
> > > Does that sound good to you?
> > >
> > > 2.
> > > Since it may be difficult to determine the distribution $D$ from a given arm set, we show by a suite of simulations that our methods can still work well after the “approximation” step in practice. We include these simulations mainly to show that a modified version of our method could be used and perform well in practice. But we may not have time to rigorously prove that the score function of distribution $D$ and the cumulative regret could be preserved with this approximation before the deadline.
> > >
> > > We would introduce this practical modification of our algorithm at the beginning of our experiment section to make it clear in the revision. Thanks for helping improve our paper.
> > >
> > > Furthermore, we include simulations here to show the advantages of this approximation trick over other matrix detection methods in practice: To pre-check the efficiency of this approximation step on our Stein's lemma-based method, we also tried the nuclear-norm regularized log-likelihood maximization (Appendix J) in stage 1 for two cases of our simulations: 480 arms, d=10, r=1 (Figure 1(a) case) and 480 arms, d=10, r=2 (Figure 1(c) case). Specifically, we used the same setting as described in Appendix I, and implemented proximal gradient descent with the backtracking line search for optimization. The average cumulative regret along with the average transformed error $||\theta^*_{k+1:p}||_2$ defined in Eqn.(8) are reported below:
> > >
> > > | Case       | Low-rank detection method   | Regret                        | Transformed error |
> > > |------------|-----------------------------|-------------------------------|-------------------|
> > > | Figure1(a) | Stein's lemma-based method  | G-ESTT:723.27, G-ESTS:510.80   | 0.086             |
> > > | Figure1(a) | Log-likelihood maximization | G-ESTT:724.96, G-ESTS:515.25   | 0.089             |
> > > | Figure1(c) | Stein's lemma-based method  | G-ESTT:1088.26, G-ESTS:1106.71 | 0.542             |
> > > | Figure1(c) | Log-likelihood maximization | G-ESTT:1136.54, G-ESTS:1198.39 | 0.583             |
> > >
> > > Therefore, we can see that our low-rank matrix detection method (with approximation) outperforms the log-likelihood maximization method. And our G-ESTT and G-ESTS are remarkably better than the existing method (LowESTR, SGD-TS) with either matrix recovery method in stage 1.

---

> > > > ### Comment · Reviewer_gC7G · 2022-08-08
> > > > **Response of the response of the response**
> > > >
> > > > Hello.
> > > >
> > > > Thanks again for your detailed explanation again, but still I cannot understand.
> > > >
> > > >
> > > > 1) The explanation you made is, assuming you are right, as follows:
> > > >
> > > > * Even after some arbitrary 'corruption' on $X_t(t=1,...,T1)$ by $U0, V0$, if the result is same, it will exactly create the same action set as the case when we don't rotate.
> > > >
> > > > The main problem in your reusing-data is as follows:
> > > >
> > > > * Is the lemma C.3 still valid even if the regularizer matrix $\Lambda$ is dependent on the first T1 elements?
> > > >
> > > >
> > > > The thing I think is true is like this:
> > > >
> > > > - Whatever the coordinate changing matrix $P$ is, the inner product does not change $<X, \Theta> = <Px, P\Theta>$.
> > > > - However, it changes the last norm term $\Lambda$ by $P^\top \Lambda P$ and P depends on first $T_1$ elements.
> > > > - Lemma C.3 is not sure when the regularizer is dependent on the samples.
> > > >
> > > > and I am still thinking it as true.
> > > >
> > > > Plus, is your claim
> > > >
> > > > >Then the solution to our matrix recovery problem in stage 1 becomes $U_0\hat{\Theta}V_0^\top$ where $\hat{\Theta}$ is the solution without orthogonal transformation.
> > > >
> > > > really true? I want a proof for it. If my understanding is right, you are saying whatever corruption happens to the first $T_1$ experiments it makes the same SVD results.......
> > > >
> > > >
> > > > 2) I understand it is difficult to prove the validity of the approximation within a short period of time. I think I have to think it as a drawback of this paper.
> > > >
> > > > To be clear, what I really want to ask is, how much does the second score moment bound $M$ in Assumption 3.3 changes from your 'original' gaussian random matrix sampling to the 'true action sampling after the nearest neighborhood operation'. If it has dimensional dependency like $d_1 + d_2$, then it weakens your result quite much.
> > > >
> > > >
> > > >
> > > > For example, ignoring the score now, and think only about the deformation of the distribution. Starting from the uniform distribution, let's do the nearest neighborhood thing on the true action set where $a_0=(-1,0,.....0)$, $a_1=(1,0,.....)$, and all other (d-2) actions are pretty close to $a_1$ and assures $span(A)=R^d$. Then originally the covariance matrix of the uniform distribution is $1/d \times I_d$ (I think maybe it will be right), but now the new covariance matrix $\Sigma$ will be $\Sigma_{11} > \frac{1}{2}$ which is totally deformed from the original one by the dimensional order.
> > > >
> > > >
> > > >
> > > >
> > > > Additional question: According to your answer,
> > > >
> > > > >Note that all existing low-rank matrix estimation methods (e.g. log-likelihood maximization in Appendix J) would waste some information for exact singular value recovery, and hence are inefficient for our problem.
> > > >
> > > > You are arguing that the previous approaches focus on too much about the 'singular values'. However, according to my memory, 'Low-Rank Generalized Linear Bandit Problems' also used the LowESTR which takes only the coordinate directions and not about the singular value. Are you saying that your Stein's method is especially better in achieving only directions than the traditional method?

---

> > > > > ### Author Response · Authors · 2022-08-08
> > > > > **Response to your Q2 and additional question:**
> > > > >
> > > > > Q2. Thanks for your insightful comment. Since the bottleneck of the low-rank matrix bandit problem is still how to attain the theoretical lower bound, in this work we primarily focus on this theoretical issue and successfully resolve this open problem by introducing two methods G-ESTT and G-ESTS. We acknowledge that this approximation method may perform unstably in practice when the arm set is small. And according to the experiments in my last rebuttal, we can observe that G-ESTT and G-ESTS could perform efficiently and much better than all the existing algorithms (LowESTR, SGD-TS) when we use either the Stein’s-lemma-based matrix estimation algorithm or the classic low-rank matrix estimation approach (Appendix J) in stage 1 (although our method with approximation is better). Therefore, a very big practical advantage of our frameworks is that we could easily replace the algorithms in two stages with any other popular methods so that any state-of-the-art algorithms for high-dimensional estimation could be incorporated.
> > > > >
> > > > >
> > > > > Additional question:
> > > > > Thanks for your questions. We obtain a better convergence rate of order $d^2r/T$ in Stage 1, which leads to the improvement on the final regret bound. We obtain this improved convergence rate of order $d^2r/T$ in Stage 1 at the expense of introducing some non-zero constant $\mu^*$ in the loss $||\hat \Theta - \mu^* \Theta^*||_F$ as shown in Eqn. (5) in our Theorem 4.1. In other words, our Stein’s-lemma-based matrix recovery method could only detect the subspace precisely, but cannot estimate the exact singular values since there are some unknown non-zero constant $\mu^*$. However, our frameworks only rely on the singular subspaces spanned by the estimate $\hat \Theta$ instead of its exact singular values in stage 2, and hence this sacrifice (introducing $\mu^*$) does not affect the Stage 2 and regret bound at all. LowESTR uses the regularized log-likelihood maximization problem (similar as the one in Appendix J) in stage 1, and this method uses the exact loss $||\hat \Theta - \Theta^*||_F$ (without $\mu^*$) as the error in stage 1. Therefore, the regularized log-likelihood maximization problem will do both subspace estimation and the exact singular value estimation simultaneously. However, LowESTR only uses the directions but not the singular values in stage 2, so there is an information waste. In other words, stage 2 of LowESTR only relies on the subspace direction, but stage 1 of LowESTR estimates both the subspace direction and the exact singular value, and hence this incompatibility between two stages of LowESTR makes LowESTR only attain suboptimal regret bound
> > > > >
> > > > > Conclusively, the Stein's-method-based quadratic optimization problem we originally introduced in Eqn. (4) is more suitable for subspace exploration in low-rank matrix bandits.
> > > > >
> > > > > Thanks a lot for your insightful comment and question.

---

> > > > > > ### Comment · Reviewer_gC7G · 2022-08-09
> > > > > > **Thank you for your responses**
> > > > > >
> > > > > > Thanks for your patience and responses!
> > > > > >
> > > > > > Now I understand how your Stein's method helps to get a better regret bound, and I'm keep reading about the technical proof of new C.3, but thanks for your effort. I changed my point from 5 to 7.

---

> > > > > ### Author Response · Authors · 2022-08-08
> > > > > **Response to your Q1**
> > > > >
> > > > > Thanks for asking. After our careful thought during this whole week, we believe your concern could be resolved by proving a modified version of Theorem 1 in Abbasi-Yadkori OFUL paper. We will not use the argument in our last reply since a formal theorem may be better than an intuitive explanation. And we will show a more general result and lemma for a better presentation of our work: although the diagonal matrix $\Lambda$ depends on the first $T_1$ samples, the upper bound for Eqn. (19) still holds. We write the modified Lemma C.3 here and a complete proof is added in our new revision of appendix:
> > > > >
> > > > > Lemma C.3: (adapted from OFUL paper) Let $F_t, (t=1,\cdots)$ be a filtration and $\{x_t\}, (t=0,\cdots)$ be an $\mathbb{R}^d$-valued stochastic process adapted to $F_t$. Let $\{\eta_t\}, (t=0,\cdots)$ be a real-valued stochastic process such that $\eta_t$ is adapted to $F_t$ and is conditionally $\sigma_0$-sub-Gaussian for some $\sigma_0 > 0$, i.e.
> > > > > $$
> > > > >     E[\exp(\lambda \eta_t) \vert F_t] \leq \exp\left( \dfrac{\lambda^2 \sigma_0^2}{2} \right), \quad \quad \forall \lambda \in R.
> > > > > $$
> > > > > Consider the martingale $S_t = \sum_{k=1}^{t} \eta_{k} x_k$ and the process $V_t = \sum_{k=1}^{t} x_{k} x_{k}^{\top} + \Lambda$ when $t \geq 2$. And $\Lambda$ is fixed and independent with the samples after time $m$. For any $\delta > 0$, with probability at least $1-\delta$, we have the following result simultaneously for all $t \geq m+1$:
> > > > > \begin{align}
> > > > >     ||S_t||_{V_t^{-1}} \leq \sigma_0 \sqrt{\log (\det(V_t)) - \log(\delta^2 \det(\Lambda))}. \nonumber
> > > > > \end{align}
> > > > >
> > > > >
> > > > > Intuitively, this bound holds since we only obtain a simultaneous bound for $t \geq m+1$, and the diagonal matrix $\Lambda$ is a constant matrix after round $m$. Note for the Eqn. (19), we only try to bound the term simultaneously for all $t$ in stage 2 but not in stage 1, and the diagonal matrix $\Lambda$ is independent with any samples in stage 2. Therefore, we could use this adapted version of Lemma C.3 to get the same simultaneous upper bound.
> > > > >
> > > > > We sincerely appreciate your careful review and insightful discussion with us, which definitely helps us improve our paper. Thanks again to you!

---

> ### Author Response · Authors · 2022-08-02
> **Thank you very much for your insightful comments. We sincerely appreciate your time and detailed review, and please see our responses below:**
>
> Thanks for helping improve the quality of our work.
>
> Weakness 4 **"This matrix bandit field itself is a relatively small field, so it would be better if the author explains how his ingenious approach can be extended to other larger fields"**
>
> Thanks for your insightful suggestion.
>
> 1. We believe the matrix bandit has great potential to improve the performance of traditional contextual linear bandits in real-world applications. As most real-world datasets contain two-sided information for both users and items, e.g. movie recommendations (movie-user pair), dating market (male-female pair), cloth recommendations (top-bottom pair), people always concatenate the feature vectors of each pair when using the linear bandit algorithm. But this simple concatenation might lose the correlation inside each pair, and hence we believe it would be more reasonable to represent each pair by a feature matrix via the outer product. Consequently, it becomes a matrix bandit problem where each arm represents a pair of information. Therefore, for real-world applications of bandit algorithms, our approach has great potential.
>
> 2. Our subtraction idea (G-ESTS) could be implemented in any other sparse high-dimensional bandit problems, e.g. LASSO bandit by removing variables that are insignificant in the estimate. And this idea could transfer a high-dimensional bandit to a low-dimensional one, which remarkably reduces the computation.
>
> Q1 **"Could you tell me intuitively why the Stein's method works better to estimate matrix completion compared to the previous approach?"**
>
> Thanks for your careful review on our proof of Theorem 4.1. Intuitively, we take advantage of the fact that only singular subspaces spanned by $\Theta^*$ are required for our transformation in stage 2, but its exact singular values are not necessary. Therefore, we introduced our Stein’s-lemma-based quadratic optimization problem, which particularly focuses on subspace detection and hence is more appropriate for our explore-then-commit frameworks. Note that all existing low-rank matrix estimation methods (e.g. log-likelihood maximization in Appendix J) would waste some information for exact singular value recovery, and hence are inefficient for our problem.
>
> Limit2 **"The way to draw samples should be suggested"** and Q2 **"It would be great if this paper describes how to get such a distribution from a given action set......"**
>
> Thanks for your careful reading. In our simulations, we used the centered normal p.d.f. with variance $1/d_1d_2$ for each entry as $p_{ij}$, and randomly draw a matrix based on this distribution entrywisely, and then I simply pull the arm in the action set that is closest to the random matrix in Frobenius distance. And we drew the action matrices in the same way as [2] did: draw each entry from the standard normal distribution and then standardize them if necessary. This setting works very well as shown in our experiments. I will add this full description of our simulations in the revision.
>
> Therefore, in practice I think we could try the following procedure for sampling when the action set is large: first, we could standardize the sample space to unit sphere (ball), and then we could use a standard normal distribution with variance $1/d_1d_2$ to draw a random matrix entrywisely, and then choose the arm that is closest to this random matrix. This sampling technique works very well as shown in our simulations. If we know the distribution where the data comes from, then we can naturally use this underlying distribution.
>
> We answer other questions and concerns in the other comment window due to the length limit. Thanks again for your careful review. Please let us know if there is any further question or comment you may have.
>
> [1] Provably optimal algorithms for generalized contextual bandits. Li et al. ICML 2017
>
> [2] Improved regret bounds of bilinear bandits using action space analysis. Jang et al. ICML 2021
>
> [3] Low-rank generalized linear bandit problems. Lu et al. AISTATS 2021

---

### Official Review · Reviewer_SuJ7 · 2022-07-12

**Rating:** 5
**Confidence:** 3
**Soundness:** 3 good
**Presentation:** 3 good
**Contribution:** 3 good

**Summary:**

This paper proposes two efficient algorithms called G-ESTT and G-ESTS for generalized low-rank matrix bandit problems. Both algorithms achieve the optimal regret bound up to logarithm terms. G-ESTT extends the two-stage algorithm LowESTR to the nonlinear reward setting, and G-ESTS further drops all negligible entries to get a low-dimensional bandit, significantly reducing the computational costs.


**Questions:**

1. Since G-ESTT adopt the similar two-stage framework as LowESTR, I hope the authors can provide some intuitive explanation why G-ESTT can improve the regret bound by $\sqrt{d}$.
2. Assumption 3.3 seems proposed for fixed action set. If the action set is time-varying, should the sampling distribution be also time-varying?
3. In the experiments, Which sampling distribution is used in stage 1? the accuracy for subgradient methods?
4. SGD-TS algorithm is designed for the finite arm setting. I hope the authors can provide some efficient generalized linear bandit algorithms for infinite arms.
5. The proposed algorithms only achieves the optimal regret when $d_1=d_2$. What is the lower bound of the low-rank matrix bandits where $d_1$ is far from $d_2$? If the lower bound is $O(r\sqrt{d_1d_2T})$ rather than $O((d_1+d_2)r\sqrt{T})$, the proposed algorithms are suboptimal in this case.
6. What is the lower bound of generalized low-rank matrix bandit problem with finite arm?


**Limitations:**

no potential negative societal impact

**Strengths And Weaknesses:**

Strengths:

1. The paper is mostly clear.
2. The technical contribution is solid. The proposed algorithms achieve the optimal regret up to logarithm terms. In addition, the G-ESTS algorithm is very efficient.


Weaknesses:
1. The paper only contains experiments with synthetic data. It would be better if real data experiments were included.
2. Some details need more clarifications (see Question part).

---

> ### Author Response · Authors · 2022-08-02
> **Thanks for your valuable comments, please see our responses to your question 2-6**
>
> **"Assumption 3.3 seems proposed for fixed action set? In the experiments, which sampling distribution is used in Stage 1?" (Q2 and 3)**
>
> In theory we require the sampling distribution as some fixed distribution at each iteration, regardless of whether the action set is varying or not. This requirement might sound more reasonable under a commonly-used assumption (e.g. [2, 3]) that the arms in the time-varying action set are drawn from some fix distribution at each iteration. In our experiments, we use the centered normal p.d.f. with variance $1/d_1d_2$ for each entry, randomly draw a matrix based on this distribution entrywisely, and then pull the arm in the action set that is closest to the random matrix. This works very well as shown in our experiments. For the accuracy of matrix estimation in stage 1, here we report the mean transformed error $||\theta^*_{k+1:p}||_2$ defined in Eqn.(8) rather than the loss $||\hat \Theta - \mu^* \Theta^*||_F$ shown in Theorem 4.1 for each setting since the unknown constant $\mu^*$ in the loss may have different values:
>
> | Case        | Transformed error |
> |-------------|-------------------|
> | Figure 1(a) | 0.086             |
> | Figure 1(b) | 0.095             |
> | Figure 1(c) | 0.542             |
> | Figure 1(d) | 0.577             |
> | Figure 2(a) | 0.081             |
> | Figure 2(b) | 0.092             |
> | Figure 2(c) | 0.537             |
> | Figure 2(d) | 0.570             |
>
>
> **"SGD-TS algorithm is designed for the finite arm setting. I hope the authors can provide some efficient......"**
>
> Thank you very much for your suggestion. In theory, we can use any state-of-the-art generalized linear bandit for infinite arms (e.g. OFUL, LinTS) in the stage 2 of G-ESTS to attain the optimal lower regret bound. In practice, as long as the linear bandit algorithm could solve the continuous maximization problem under infinite arms efficiently, our G-ESTS could incorporate it in stage 2. And this could save much more computation than solving a continuous optimization problem in matrix space. While we believe it’s still a challenging topic to find a good generalized linear bandit algorithm for infinite arms in the bandit community.
>
> **"The proposed algorithms only achieve the optimal regret when $d_1 = d_2$. What is the lower bound of the low-rank matrix bandits where $d_1$ is far from $d_2$......?"**
>
> Thanks for raising this interesting and perceptive question. First, there is no current literature on the regret analysis when $d_1 >> d_2$. And the optimal regret bound $O(dr \sqrt{T})$ proposed in [1] (Theorem 6) is also deduced under the condition that $d_1=d_2$ (or $d_1 = \Theta(d_2)$). Therefore, whether the optimal regret bound is better than $O((d_1+d_2)r\sqrt{T})$ when $d_1$ is far from $d_2$ is still an open question. And all existing works in low-rank matrix bandits focus on the case when $d_1$ and $d_2$ are in the same scale.
>
> Second, even if we have $d_1 >> d_2$, our regret bound still improves all existing literature, since the regret bound of all previous methods depend on $d$ in the order of $\sqrt{d_1d_2d}$.
>
>
> **"What is the lower bound of generalized low-rank matrix bandit problem with finite arm?"**
>
> Thanks for pointing this out. This is still an open question. We proved that our G-ESTS algorithm can achieve $\tilde O(\sqrt{d_1d_2rT})$ regret bound under the finite arm, which indicates the lower bound of generalized low-rank matrix bandit problem with finite arm is at most of order $\tilde O(\sqrt{d_1d_2rT})$. This fact opens a potential future direction in this area.
>
> Thanks again for helping improve the quality of our paper. Please feel free to let us know if you have any further questions.
>
> [1] Low-rank generalized linear bandit problems. Lu et al. AISTATS 2021
>
> [2] Provably optimal algorithms for generalized linear contextual bandits. Li et al. ICML 2017
>
> [3] An efficient algorithm for generalized linear bandit: online stochastic gradient descent and Thompson sampling. Ding et al. AISTATS 2021

---

> > ### Comment · Reviewer_SuJ7 · 2022-08-08
> > **Thank you for the responses**
> >
> > Thank you for your answers to my questions. However, after reading all reviewers' comments and the authors' replies, I still want to keep my score at 5.

---

> > > ### Author Response · Authors · 2022-08-08
> > > **Thank you very much for your reading**
> > >
> > > Thanks for your reading and for helping us improve our paper, and we hope our rebuttal helps you better understand our work and resolves your concern. Please feel free to let us know if you have any further questions or comments.

---

> ### Author Response · Authors · 2022-08-02
> **Thanks a lot for your insightful comments, please see our responses to the weakness you pointed out and question 1 below:**
>
> **“Weakness: lack of real data implementation”**
>
> Thanks for your comment. Since there is no real data implementation in all previous matrix bandit literature, we don’t have a real dataset in hand. And the potential real data we plan to use is the benchmark Movielens 100K dataset. This dataset contains ratings 1,682 movies contributed by 943 users. After data preprocessing, we could get the feature vector for each movie and user via matrix factorization. And then we could represent each user-movie bundle by the outer product of their corresponding feature vectors. The model parameter matrix $\Theta^*$ is chosen as the average of feature matrices of some randomly selected user-movie bundles. But we don’t have enough time to finish this experiment for rebuttal.
>
> **"Since G-ESTT adopt similar two-stage framework as LowESTR,......why G-ESTT can improve the regret bound by $\sqrt{d}$?"**
>
> We obtain a better convergence rate of order $d^2r/T$ for the low-rank matrix recovery in Stage 1, which leads to the improvement in the final regret bound. Specifically, we take advantage of the fact that only singular subspaces spanned by $\Theta^*$ are required for our transformation in stage 2, but its exact singular values are not necessary. Therefore, we introduced our Stein’s-lemma-based quadratic optimization problem, which particularly focuses on subspace detection and hence is more appropriate for our explore-then-commit frameworks. Note that all existing low-rank matrix estimation methods (e.g. log-likelihood maximization in Appendix J) would waste some information for exact singular value recovery, and hence are inefficient for our problem.
>
> Thanks for this insightful question and we will add an intuitive explanation in the revision.

---

### Official Review · Reviewer_APoi · 2022-07-12

**Rating:** 5
**Confidence:** 3
**Soundness:** 4 excellent
**Presentation:** 4 excellent
**Contribution:** 3 good

**Summary:**

The paper considers generalized low-rank matrix bandit problems. The expected rewards are modeled as $\mu(\langle \Theta^*, X_t\rangle)$ and $\Theta^* \in \mathbb{R}^{d_1 \times d_2}$ is an unknown low-rank matrix with rank $r$.
For this problem, the authors firstly proposed the subspace extraction-based algorithm G-ESTT.
G-ESTT's subspace extraction guarantee is based on Stein's method.
In stage 2 of the G-ESTT, based on the extracted subspace, LowGLM-UCB is run.
The overall regret guarantee is improved from the previous algorithm.
They further proposed G-ESTS, which is an even faster (in time complexity) algorithm than G-ESTT.
G-ESTS achieves the same order of the bound of G-ESTT, and improved bound when the action set is finite.
Finally, numerical experiments on the synthetic data have shown the efficacy of the proposed algorithm in terms of the empirical time complexity and the overall regret.


**Questions:**

### Major question

With the constant upper bounds on the matrix norm, S_f, can D_{rr} be kept constant independent of d?
Intuitively, I feel that since the magnitude of the element of the matrix is suppressed by 1/d, etc., the size of the eigenvalue also depends on d for wide range of instances.
I would like to know if there are many examples that are not dependent on d.

### minor

line 180, k-dimensional?

line 194, x_t, should be x_k?

improvement from (d_1 + d_2)^{3/2} \sqrt{rT} to (d_1 + d_2)r \sqrt{T} is only in d_1 and d_2? for some regime, dependence on r is disimproved?

line 253, experments

line 244, why you say O(\sqrt{kT}) in theorem statement but O(k\sqrt{T}) in the footnote?

 LowGLM-UCB,  (P)LowGLM-UCB,  PLowGLM-UCB: namings are a little bit confusing.

It looks to me that G-ESTS is better than G-ESTT in every aspect. Is there any reason why you present G-ESTT too, instead of only presenting G-ESTS? Is there any advantage of G-ESTT over G-ESTS?

Why numerically, the algorithm is only compared with LowESTR?







**Limitations:**

See questions above.

**Strengths And Weaknesses:**

- Strengths

Improved regret guarantees the problem with standard assumptions and with additional assumptions.

Computationally efficient algorithm (G-ESTS) is proposed.

- Weaknesses

Explore then commit type algorithm (although two stages are sharing some data, I think it's true for subspace extractions.) is not practical.

---

> ### Author Response · Authors · 2022-08-01
> **Thanks a lot for your insightful comments, and please see our responses to your questions and concerns below**
>
> **"Major question: with the constant upper bounds on the matrix norm, $S_f$, can $D_{rr}$ be kept constant independent......?"**
>
> Thanks for raising this insightful question. Since under our problem setting the parameter matrix $\Theta^*$ ($d_1 \times d_2$) has a low rank structure, i.e. r=rank($\Theta^*$)  $<<d_1,d_2$, we believe it is quite reasonable to assume the smallest non-zero singular value is of constant scale, or at least not suppressed by some order of $d$. Note that the term $D_{rr}$ is inevitable in the regret analysis for all explore-then-commit-type methods (e.g. ESTR, LowESTR) since when $D_{rr}$ is close to zero it is impossible to detect the corresponding subspace. Although $\epsilon-$FALB [1] doesn't require this assumption, it is only designed for the bilinear bandit (a special case), and their regret analysis can't be extended to our problem setting.
>
> **"typos"**
>
> Thank you so much for your careful review. All spelling and grammatical errors pointed out here have been corrected in the rebuttal revision. In particular, we'd like to point out that in line 244 the order should be $\tilde O(k \sqrt{T})$, which is consistent with the footnote.
>
>
> **"improvement from $(d_1+d_2)^{3/2} \sqrt{rT}$ to $(d_1+d_2)r \sqrt{T}$, dependence on $r$ is disimproved?"**
>
> Note that under our problem setting we have $r << d_1,d_2$, and hence our regret bound with order $(d_1+d_2)r \sqrt{T}$ is evidently better than the previous one $(d_1+d_2)^{3/2} \sqrt{rT}$. Even if the assumption $r << d_1,d_2$ doesn’t hold, since $r \leq \min(d_1,d_2)$ is always true, our regret bound is always better or equal to the previous ones. Moreover, since our regret bound already matches the optimal lower bound of the low-rank matrix bandit problem presented in [2], we believe it is impossible to further improve the dependence on $r$.
>
>
> **"LowGLM-UCB, (P)LowGLM-UCB, PLowGLM-UCB namings":**
>
> Thanks for pointing out this naming issue. LowGLM-UCB is the algorithm presented in the main paper (Algorithm 2), PLowGLM-UCB is a computationally efficient version of LowGLM-UCB with a description in line 213 and details in Appendix H. (P)LowGLM-UCB in line 220 means LowGLM-UCB and PLowGLM-UCB. We would make these namings clear in the revision.
>
> **"It looks to me that G-ESTS is better than G-ESTT in every aspect. Is there any reason why you present G-ESTT too?"**
>
> We present G-ESTT ahead of G-ESTS for the completeness of our work. And a big advantage of G-ESTT over G-ESTS and all other explore-then-commit-type algorithms is that it could reuse the arms and response in the exploration stage again in Stage 2 (line 195). This could help us obtain a consistent and promising estimate at the very beginning of Stage 2, while other explore-then-commit-type algorithms still need some warmup when switching to Stage 2. This advantage is also validated in our experiments. Specifically, from Figure 1 (c),(d) and Figure 2 (c), (d) (Appendix I.2) we can see that G-ESTT could yield more robust performance when switching to Stage 2.
>
>
> **"Why numerically, the algorithm is only compared with LowESTR?"**
>
> This is because LowESTR is the only computationally feasible algorithm for the (generalized) low-rank matrix bandit problem. Specifically, LowLOC[2] and LowGLOC[2] are computationally prohibitive since they need to calculate the weights of a self-constructed covering of some low-rank matrix space at each iteration, and their authors also pointed out this impractical issue (Section 6 first paragraph, https://arxiv.org/pdf/2006.02948.pdf) on their paper, so they didn’t really implement these methods. ESTR and $\epsilon$-FALB are only designed for the bilinear bandit setting, and hence they are not applicable under our generalized low-rank matrix bandit problems. These facts also indicate the significance of our work not only in theory but also in practice.
> Thanks for pointing this out, and we'll make a remark in the revision.
>
> Thanks again for your valuable suggestions which definitely help improve the quality of our paper. Please feel free to let us know if you have any further questions.
>
> [1] Improved regret bounds of bilinear bandits using action space analysis. Jang et al. ICML 2021
>
> [2] Low-rank generalized linear bandit problems. Lu et al. AISTATS 2021

---

> > ### Comment · Reviewer_APoi · 2022-08-09
> > **reply**
> >
> > Thanks for your detailed comments.
> > My question is whether $D_{rr}$ is kept constant under Assumption 3.4.
> >
> > If look at eq. (6) of
> > https://arxiv.org/pdf/2102.01229v1.pdf for example,
> > the minimum eigenvalue is a function of $d$.
> > In many cases $D_{rr}$ can depend on $d$ then the regret order can be different in terms of $d$.
> >
> > Best,
> >
> > Reviewer APoi

---

> > > ### Author Response · Authors · 2022-08-09
> > > **Response to your question**
> > >
> > > Thank you very much for the reply. Since we assume the parameter matrix $\Theta^* (d_1 \times d_2)$ has a low-rank structure $r < < d_1, d_2$, then we believe the minimum non-zero eigenvalue $D_{rr}$ should at most be suppressed by an order of $r$ (i.e. $1/\sqrt{r}$) instead of $d$ given the matrix norm bound $S_0$. And even if $D_{rr}$ is in the order of $1/\sqrt{r}$, the final regret bound of G-ESTT and G-ESTS is still $\tilde O ((d_1+d_2)r \sqrt{T})$. Specifically, according to our Theorem 4.2 and 4.3, the accumulative regret could be upper bounded by
> > > $$\tilde O ((\frac{\sqrt{d_1d_2r}}{D_{rr}} + (d_1+d_2)r )\sqrt{T})$$
> > >
> > > And both terms are in the order of $\tilde O ((d_1+d_2)r \sqrt{T})$ given $D_{rr}$ is lower bounded by $1/\sqrt{r}$. Thank you very much for raising this question, and we added a description regarding this issue in the revision.
> > >
> > > Note that the term $D_{rr}$ is inevitable in the regret analysis for all explore-then-commit-type methods. (e.g. ESTR, LowESTR) And our regret bound is strictly better than any existing method by an order of $\sqrt{d}$ regarding the dependence on $d$.

---

> ### Author Response · Authors · 2022-08-02
> **Thanks a lot for your insightful comments, and please see our responses to the weakness you pointed out:**
>
> **"Weakness: Explore then commit type algorithm is not practical"**
>
> We believe the explore-then-commit-type frameworks still have great practical potential for the high-dimensional bandit community. Since both the sparse high-dimensional estimation problems (e.g. low-rank matrix recovery, sparse linear regression) and low-dimensional contextual linear bandit problem have been extensively studied with comprehensive empirical results, it would be easy and efficient to combine those state-of-the-art methods based on the explore-then-commit-type idea to solve high-dimensional bandit problems with sparsity. Also, a very big advantage of the explore-then-commit-type framework is that we could easily replace the algorithms in two stages with any other popular methods, so that the framework could always incorporate state-of-the-art algorithms for high-dimensional estimation. In our work, both of our proposed frameworks are verified on a suite of numerical experiments.
>
> The bottleneck of the low-rank matrix bandit problem is still how to find an algorithm attaining the theoretical lower bound deduced in [2]. In this work, we successfully addressed this open problem.
>
>
> [1] Improved regret bounds of bilinear bandits using action space analysis. Jang et al. ICML 2021
>
> [2] Low-rank generalized linear bandit problems. Lu et al. AISTATS 2021

---

### Official Review · Reviewer_Bwj4 · 2022-07-26

**Rating:** 7
**Confidence:** 2
**Soundness:** 3 good
**Presentation:** 2 fair
**Contribution:** 4 excellent

**Summary:**

This paper proposes efficient algorithms for matrix GLM bandit with regret and runtime guarantees additive in the matrix dimensions a significant improvement from previous work on low-rank bandits with linear and non-linear rewards (generalized linear model). Both the algorithms are verified on a suite of numerical experiments.

**Questions:**

--
Did the authors consider other methods to detect low-rank especially those based on random projections, thresholding and PCA, at least int the numerical section?

**Limitations:**

--
Writing can be significantly improved.

**Strengths And Weaknesses:**

-- Strengths:
1. The extension to GLM reward using a quadratic optimization problem is non-trivial. This forms the key novel step in the first algorithm, while the second step relies on a standard GLM-UCB algorithm. The novelty for the reduction to low-dimensional space lies in construction of an oracle inequality. Although, this reduces the regret incurred by the bandit algorithm making its implementation computationally expensive.

2. In order to improve the computational complexity of the proposed algorithm, the authors suggest another algorithm by excluding inclusions into a properly determined subspace thereby allowing to invoke any low-dimensional GLM bandit algorithm. Surprisingly, this inclusion doesn't lead to large estimation error as compared to the previous algorithm.

--
Weakness:

1. Writing can be significantly improved. While the paper is technical, the description of the results and the intuition behind them can be elucidated using a clear language.

---

> ### Author Response · Authors · 2022-08-02
> **Thank you for your insightful suggestions. Please see our responses to your question below:**
>
> Thank you very much for your time and valuable suggestions. And we are pleased to learn that you find our proposed frameworks are theoretically solid with improved regret bound over all previous work, and practically efficient. Regarding the writing issue, we will definitely elucidate our description and intuition more clearly in the revision. Please see our response to your main question:
>
> **"Did the authors consider other methods to detect low-rank......?"**
>
> 1. We have also considered the popular nuclear-norm regularized log-likelihood maximization problem as shown in Eqn.(23) in appendix J. We originally proved that the bound of loss $||\hat\Theta - \Theta^*||_F^2$ is of order $d^3r/T_1$ in Theorem J.4 for this method. The proof is based on restricted strong convexity, as stated in Appendix J. However, this bound is worse than the one in our Theorem 4.1 so we did not include the method in the main paper. This comparison firmly shows the advantage of our Stein's lemma-based method in theory.
>
> 2. To pre-check the efficiency of our Stein's lemma-based method, we also tried the above nuclear-norm regularized log-likelihood maximization in stage 1 for two cases of our simulations: 480 arms, d=10, r=1 (Figure 1(a) case) and 480 arms, d=10, r=2 (Figure 1(c) case). Specifically, we used the same setting as described in Appendix I ($T_1=1800, T = 45000$), and implemented proximal gradient descent with the backtracking line search for optimization. The average cumulative regret along with the average transformed error $||\theta^*_{k+1:p}||_2$ defined in Eqn.(8) are reported below:
>
> | Case       | Low-rank detection method   | Regret                        | Transformed error |
> |------------|-----------------------------|-------------------------------|-------------------|
> | Figure1(a) | Stein's lemma-based method  | G-ESTT:723.27, G-ESTS:510.80   | 0.086             |
> | Figure1(a) | Log-likelihood maximization | G-ESTT:724.96, G-ESTS:515.25   | 0.089             |
> | Figure1(c) | Stein's lemma-based method  | G-ESTT:1088.26, G-ESTS:1106.71 | 0.542             |
> | Figure1(c) | Log-likelihood maximization | G-ESTT:1136.54, G-ESTS:1198.39 | 0.583             |
>
>
> Therefore, we can see that our low-rank matrix detection method outperforms the log-likelihood maximization method, especially when the underlying parameter matrix is complicated (Figure 1(c) case). This is also consistent with our theoretical analysis.
>
> Thanks again for the suggestions. Please feel free to let us know if you have any further questions.

---

### Meta-Review · Area_Chair_uhWq · 2022-08-27

**Recommendation:** Accept
**Confidence:** Certain

**Metareview:**

This paper proposes computationally efficient algorithms for low-rank generalized linear bandit with regret and runtime guarantees that improves over state of the art results. All reviewers think that the technical contributions of this paper is solid, especially the insight that one can directly apply Stein's method for accurate subspace estimation without matrix estimation. Accept.

The authors are encouraged to include the additional experiments in the rebuttal in the final version.

**Award:**

No

---

### Decision · Program_Chairs · 2022-09-14

Accept